# ARTICLES

## OPEN

# Pan-cancer pervasive upregulation of 3′ UTR splicing drives tumourigenesis

Jia Jia Chan[1,32], Bin Zhang[1,2,3,32], Xiao Hong Chew[1], Adil Salhi[2,3], Zhi Hao Kwok[1,29], Chun You Lim[1], Ng Desi[1], Nagavidya Subramaniam[4], Angela Siemens[1,30], Tyas Kinanti[1], Shane Ong[1], Avencia Sanchez-Mejias[1,31], Phuong Thao Ly[5], Omer An[1], Raghav Sundar[6,7,8,9], Xiaonan Fan[1], Shi Wang[10], Bei En Siew[11], Kuok Chung Lee[11,12], Choon Seng Chong[11,12], Bettina Lieske[11,12], Wai-Kit Cheong[11,12], Yufen Goh[1], Wee Nih Fam[1], Melissa G. Ooi[6,13], Bryan T. H. Koh[14], Shridhar Ganpathi Iyer[11,15], Wen Huan Ling[16], Jianbin Chen[17], Boon-Koon Yoong[18], Rawisak Chanwat[19], Glenn Kunnath Bonney[11,15], Brian K. P. Goh[20], Weiwei Zhai[16,21], Melissa J. Fullwood[1,5,22], Wilson Wang[14], Ker-Kan Tan[11,12], Wee Joo Chng[1,6,13], Yock Young Dan[13], Jason J. Pitt[1,23], Xavier Roca[5], Ernesto Guccione[24], Leah A. Vardy[4], Leilei Chen[1,25], Xin Gao[2,3,26], Pierce K. H. Chow[16,20,27], Henry Yang[1] and Yvonne Tay ●[1,23,28] ✉

**Most mammalian genes generate messenger RNAs with variable untranslated regions (UTRs) that are important post-transcriptional regulators. In cancer, shortening at 3′ UTR ends via alternative polyadenylation can activate oncogenes. However, internal 3′ UTR splicing remains poorly understood as splicing studies have traditionally focused on protein-coding alterations. Here we systematically map the pan-cancer landscape of 3′ UTR splicing and present this in SpUR (http://www.cbrc.kaust.edu.sa/spur/home/). 3′ UTR splicing is widespread, upregulated in cancers, correlated with poor prognosis and more prevalent in oncogenes. We show that antisense oligonucleotide-mediated inhibition of 3′ UTR splicing efficiently reduces oncogene expression and impedes tumour progression. Notably, *CTNNB1* 3′ UTR splicing is the most consistently dysregulated event across cancers. We validate its upregulation in hepatocellular carcinoma and colon adenocarcinoma, and show that the spliced 3′ UTR variant is the predominant contributor to its oncogenic functions. Overall, our study highlights the importance of 3′ UTR splicing in cancer and may launch new avenues for RNA-based anti-cancer therapeutics.**

Messenger RNAs (mRNAs) comprise protein-coding regions flanked by 5′ and 3′ untranslated regions (UTRs) that play important roles in post-transcriptional regulation. UTRs harbour many regulatory sequences and structures, such as AU-rich elements, G-rich elements and microRNA response elements, through which RNA-binding proteins (RBPs) and microRNAs modulate mRNA metabolism[1]. This includes processes such as mRNA localization, stability and export, which are tightly controlled to ensure correct gene expression and function under physiological conditions.

Most mammalian genes generate alternative 3′ UTRs via various mechanisms including alternative polyadenylation (APA) and alternative splicing. In addition to key physiological functions, frequent observation of mutations and other structural variations in 3′ UTRs in various disease states suggests that 3′ UTR processing may play critical roles in pathogenesis[2,3]. For example, structural variations disrupting the 3′ UTR of *PD-L1* led to its overexpression and evasion of anti-tumour immunity[2]. In cancer, research on alternative 3′ UTRs has almost exclusively focused on APA-derived shorter 3′ UTRs[4,5]. Critically, studies have shown that 3′ UTR shortening by APA disrupted microRNA binding and associated competing endogenous RNA networks in which transcripts compete for shared microRNAs, resulting in the aberrant expression of key oncogenes and tumour suppressors in cancer[4,6]. Additionally, these shortened 3′ UTRs could differentially regulate protein function, localization

and protein–protein interactions to confer oncogenic advantages to cancer cells[6–9].

More than 95% of human genes undergo alternative splicing to dramatically increase transcriptome and proteome diversity. Recent large-scale transcriptomic analyses have revealed a high frequency of aberrant splicing in cancer[10]. Although 3′ UTR splicing events (3USPs) are annotated in databases, only a handful have been characterized so far as the vast majority of splicing studies have concentrated on the protein-coding regions of mRNAs[11]. In this Article, we systematically map the pan-cancer landscape of 3′UTR splicing and investigate its potential impact on oncogene expression and cancer progression.

## Results

**Global analysis reveals widespread 3′ UTR splicing.** Using splice junctions specifically located within 3′ UTRs (Fig. 1a), we identified and quantified 3USPs in 7,917 RNA sequencing (RNA-seq) samples across ten cancer types from The Cancer Genome Atlas (TCGA) and the corresponding tissues from The Genotype-Tissue Expression (GTEx) (Supplementary Table 1). In total, 45,815 and 18,253 3USPs were identified from TCGA tumours and their adjacent normal samples (TCGA-tumour and TCGA-normal), while 68,668 events were identified in healthy tissues from GTEx (Fig. 1b).

For each cancer and tissue type, we defined common 3USPs (c3USPs) as events detected in more than half the samples. This enabled us to obtain a robust list of events for further analysis.

Compared with the 3USPs, the total number of identified c3USPs was more consistent across different datasets, cancer and tissue types (Fig. 1b and Supplementary Table 1). Moreover, c3USPs were highly reproducible between the TCGA and GTEx datasets (Extended Data Fig. 1a,b and Supplementary Note), while the majority were ubiquitously detected in different cancers and tissues (Fig. 1c and Extended Data Fig. 1c,d). As both datasets were generated from Illumina short-read sequencing, we could not exclude the possibility that the junction reads were from independent 3′ UTRs[12]. To address this, we investigated whether these events were supported by long-read Pacific Biosciences (PacBio) sequencing and found that ~50% of the c3USPs were supported (Fig. 1d). More importantly, compared with the GENCODE annotations and the published TCGASpliceSeq[13], ~20% of the c3USPs identified from our analysis are unannotated (Extended Data Fig. 1e). Furthermore, we analysed the distances from stop codons to 3′ UTR splice sites and the sequence features of the removed introns, and found that 3′ UTR splicing is unlikely to trigger nonsense-mediated mRNA decay (NMD) and may additionally mitigate Staufen-mediated mRNA decay (SMD) (Fig. 1e,f, Extended Data Fig. 1f–l and Supplementary Note). These data suggest that not all 3USPs have been annotated, and they could be biologically functional.

**3′ UTR splicing is upregulated in cancer.** To identify spliced 3′ UTRs that were dysregulated in cancers, we compared the splicing levels (SPLs) of each c3USP in tumours with their adjacent normal in each TCGA cancer type and the corresponding GTEx normal tissue (Extended Data Fig. 2a). In total, 671 of 1,490 c3USPs showed significant differences in at least one cancer type (Supplementary Table 2). Eight of the ten cancer types analysed had more significantly upregulated than downregulated c3USPs (Fig. 1g). Intriguingly, significantly more ($P < 0.01$, hypergeometric test) oncogenes were represented in genes displaying upregulated rather than downregulated 3′ UTR splicing, whereas the difference in tumour suppressors was insignificant (Extended Data Fig. 2b,c).

We found that c3USPs identified in different cancer types are significantly overlapped ($P < 1 \times 10^{-100}$, hypergeometric test; Extended Data Fig. 2d). To further examine common versus tissue-specific c3USPs, we extended our analysis to a haematological malignancy, acute myeloid leukaemia (AML) (34 patients versus 21 healthy controls). Among the 1,431 c3USPs identified from these samples, ~46% overlapped with events identified from the ten solid tumours, a lower proportion compared with the 82% overlap among the solid tumours (Fig. 1h and Extended Data Fig. 2d). However, in line with our observations from the solid tumours, 160 c3USPs were significantly upregulated, while only 46 were downregulated in AML (Fig. 1h).

Next, we investigated the association between the SPLs of each c3USP and overall patient survival (OS) in each solid tumour type. We defined two types of prognosis-associated c3USPs based on clinical outcomes: unfavourable and favourable events, for which higher SPLs were correlated with poorer and better OS, respectively. The ratios of unfavourable and favourable events varied among the different cancer types (Fig. 1i, Extended Data Fig. 2e, Supplementary Table 3 and Supplementary Note). We overlapped them with the significantly dysregulated c3USPs across different cancers and found that the upregulated c3USPs were significantly more unfavourable (Extended Data Fig. 2f). Ninety of these were identified as unfavourable prognostic markers, while only 16 downregulated c3USPs were favourable (Fig. 1i). Taken together, these results suggest that 3′ UTR splicing is preferentially upregulated in cancers and may be linked to cancer outcomes.

**Top dysregulated 3USPs across cancers.** To gain a pan-cancer overview of each significantly dysregulated c3USP, we first measured the number of tumour samples in which the event was over- and under-spliced in each cancer type (Fig. 2a). By combining ten solid tumour types and AML, we showed that 3′ UTRs were preferentially over-spliced (median 149) than under-spliced (median 10) in tumours across 671 significantly dysregulated c3USPs. Among the ten c3USPs with the highest number of over-spliced tumour samples (Fig. 2b and Supplementary Table 4), the top candidate, *CTNNB1* c3USP (3′ SP), was over-spliced in ~40% of tumour samples (2,251/5,577) in 10 of the 11 cancer types analysed (Fig. 2b,c and Extended Data Fig. 2g–i). Notably, *CTNNB1* is a well-known oncogene that is also the second most frequently mutated gene after *TP53* in hepatocellular carcinoma (HCC) (26% versus 30.8% of TCGA-liver HCC (LIHC) tumour samples). Additionally, c3USPs from other annotated oncogenes, including *TCF3* (1,335/5,577) and *HRAS* (359/5,577), were also highly over-spliced (Fig. 2d).

**Targeted inhibition of 3′ UTR splicing impedes HCC carcinogenesis.** Among the ten cancer types studied, we found that the TCGA-LIHC samples had significantly higher numbers of 3USPs than their adjacent normal samples, and this was still true after their normalization to the number of all splicing events, including those from 5′ UTRs, 3′ UTRs, coding sequences (CDS) and non-coding RNAs (Fig. 3a and Extended Data Fig. 2j). Critically, a high number of 3USPs, but not the number of all splicing events, was significantly correlated with poorer OS (Fig. 3b and Extended Data Fig. 2k). We further analysed an additional RNA-seq dataset containing 211 samples from the Precision Medicine in Liver Cancer Asia-Pacific Network (PLANet) consortium[14], and an in-house dataset of four paired HCC-adjacent normal samples, which were sequenced to greater depths (Supplementary Table 5). The 3USPs identified from both datasets were highly consistent with those from the TCGA-LIHC data (Fig. 3c and Extended Data Fig. 3a). Moreover, the dysregulation of c3USPs in the TCGA and PLANet datasets was highly correlated (Extended Data Fig. 3b). Among the 31 shared significantly upregulated events, we selected the top 5 candidates not annotated as NMD targets (*CTNNB1*, *CHEK1*, *MAPK1*, *THUMPD1* and *WDR55*) for further experimental validation.

Following the Sanger sequencing validation of the candidate 3′ UTR splice junctions (Fig. 3d), we designed antisense oligonucleotides (ASOs) to block the 3′ UTR splice sites (Supplementary Note). These ASOs significantly reduced their respective 3′ SP expression with a concomitant increase in the unspliced, full-length transcript (3′ FL) for *CHEK1*, *CTNNB1* and *THUMPD1*, while the CDS transcripts were unaffected in the HCC cell lines, Hep3B and HepG2 (Fig. 3e,f, Extended Data Fig. 3c,d and Supplementary Note). These effects were accompanied by a decrease in the protein expression of the respective genes and the repression of tumour growth, likely due to cell cycle inhibition as evident from the downregulated expression of cell cycle genes, including CCNE1, CDK2, CDK4 and CDK6 (Fig. 3g,h and Extended Data Figs. 3e,f and 4a). This was further confirmed with additional ASOs (Extended Data Fig. 4). Thus, these ASOs could specifically inhibit 3′ UTR splicing, which potentially plays an important role in regulating the protein expression and tumourigenic functions of oncogenes.

**3′ UTR splicing of *CTNNB1* promotes tumourigenesis.** We further examined the most consistently dysregulated c3USP, *CTNNB1* 3′ SP in HCC and found that only over-splicing of the *CTNNB1* 3′ UTR, but not its somatic mutational status or total transcript expression, was significantly correlated with poorer OS (Fig. 4a), suggesting that its 3′ UTR splicing could be a robust prognosticator for HCC. 3′ UTR splicing of *CTNNB1* generates two 3′ UTR variants, 3′ SP (NM_001098210) and 3′ SP2 (NM_001330729). However, as 3′ SP2 is not significantly upregulated in HCC and has minimal effects on *CTNNB1* expression and tumourigenesis, we focused only on 3′ SP for further experimental validation (Fig. 4b and Extended Data Fig. 5a–d). As the *CTNNB1* 3′ UTR is spliced 11/12 nt downstream of the stop codon, we ruled out splicing-induced NMD by knocking

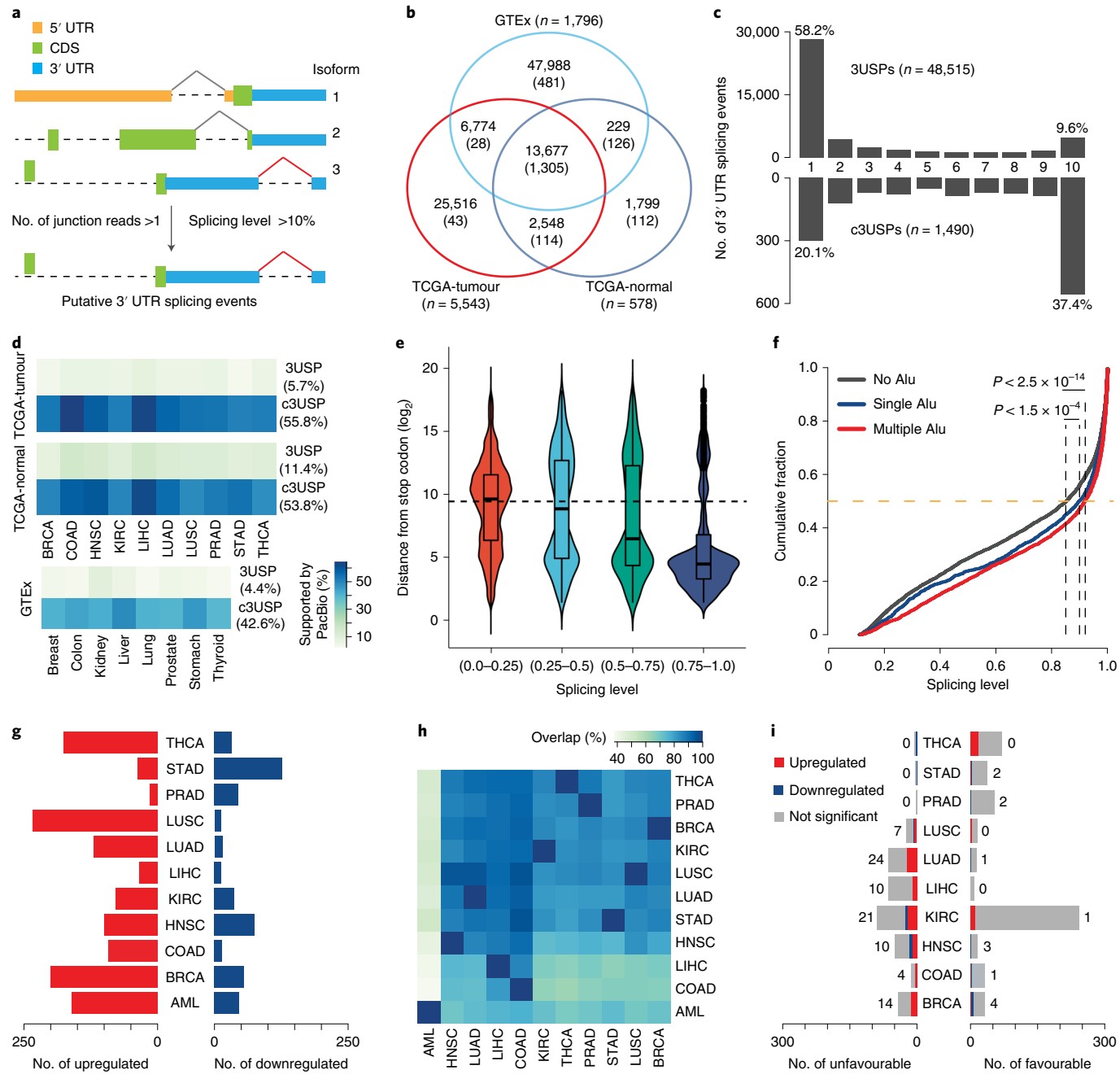

**Fig. 1 | Transcriptome-wide analysis reveals widespread 3′ UTR splicing. a**, Schematic for the identification of 3USPs. **b**, Venn diagram showing the overlap between 3USPs and c3USPs (in brackets) identified in TCGA-normal, TCGA-tumour and GTEx datasets. **c**, Bar plots showing the number of 3USPs and c3USPs detected in different numbers of TCGA cancer types. **d**, Heat map illustrating the percentages of 3USPs and c3USPs supported by PacBio data. **e**, Distribution of the distance from the stop codon for multiple groups of 3USPs classified by SPLs in TCGA-tumour. **f**, Cumulative distribution of SPLs of c3USPs in TCGA-tumour with or without Alu elements. *P* values: Wilcoxon test. **g**, Bar plots showing the number of significantly upregulated and downregulated c3USPs. **h**, Heat map illustrating percentages of the identified c3USPs overlapped across ten TCGA cancer types and AML. **i**, Bar plots showing the number of favourable (hazard ratio <1, *P* < 0.05) and unfavourable (hazard ratio >1, *P* < 0.05) c3USPs across ten cancers. Hazard ratios and *P* values: univariate Cox proportional hazards regression analysis.

down a key NMD regulator, *UPF1*, which did not alter CTNNB1 transcript and protein expression (Fig. 4c–e). This was also observed for other 3USP candidates: *CHEK1*, *MAPK1*, *THUMPD1* and *WDR55* (Fig. 4d,e). Consistent with its expression in patient samples, *CTNNB1* 3′ SP was upregulated in HCC cell lines, Hep3B and SNU398, compared with THLE-2 (Fig. 4f). We also verified that *CTNNB1* 3′ SP is conserved in mouse and is more highly expressed in the mouse tumour relative to the adjacent normal

tissue (Extended Data Fig. 5e). Collectively, these data underscore the functional relevance of the spliced 3′ UTR across different species and in a disease setting.

As *CTNNB1* plays critical roles in adherens junction formation and WNT signalling to regulate cell proliferation and migration[15], we first performed gene set enrichment analysis (GSEA) by comparing two groups of the tumour samples: (1) tumour samples with over-spliced *CTNNB1* 3′ SP and (2) the remaining tumour samples.

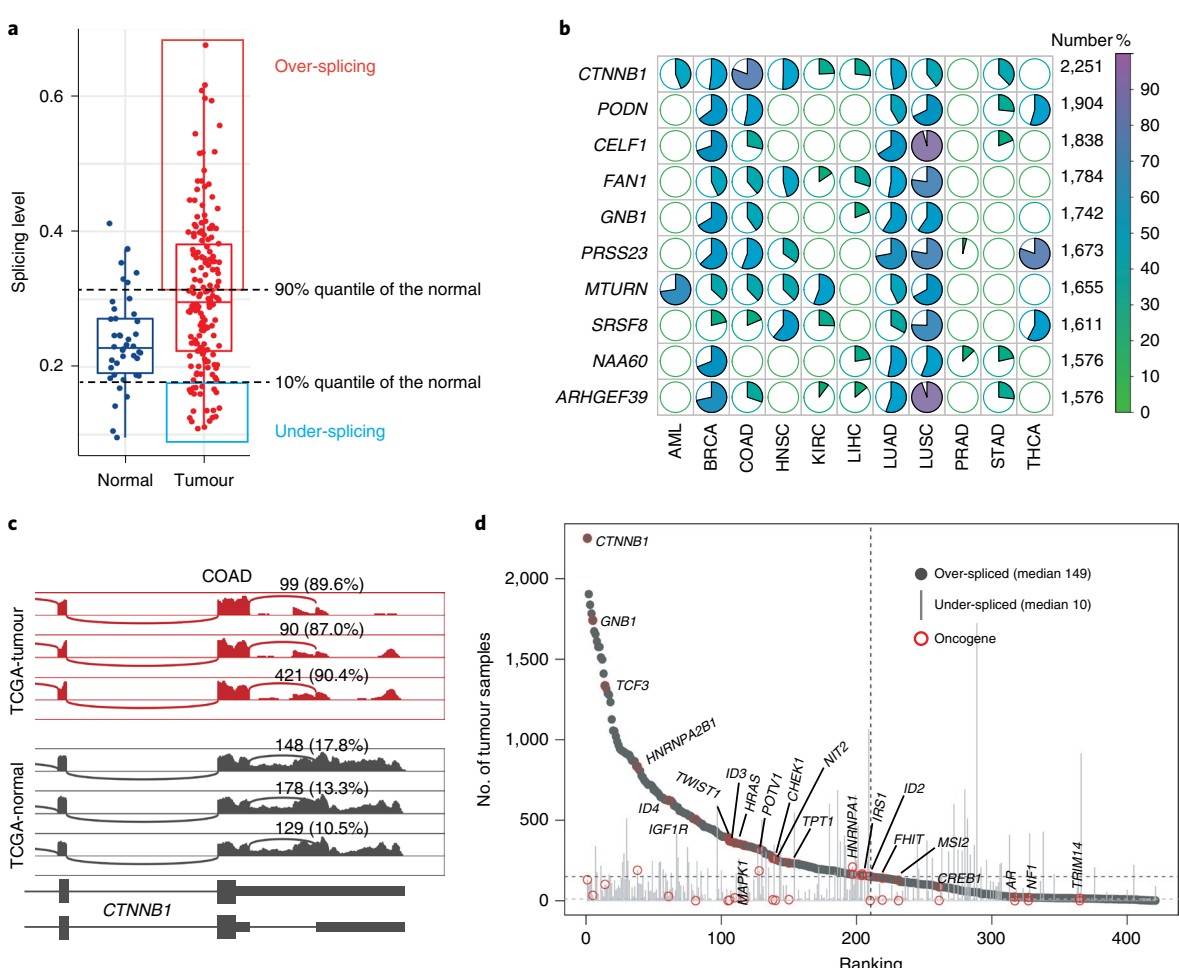

**Fig. 2 | Pan-cancer analysis identifies *CTNNB1* as the top dysregulated c3USP across 11 cancers. a**, Box plot illustrating the definition of over- and under-splicing for each significantly dysregulated c3USP event in each cancer type. Each dot represents one sample. The dashed lines indicate the 10% and 90% quantile of SPLs in the normal samples. **b**, The top ten 3USPs detected in the highest number of total tumour samples across 11 cancers. #, total number of over-spliced tumour samples; %, percentages in the colour key. **c**, Sashimi plot illustrating 3′ UTR splicing of the top dysregulated splicing event of *CTNNB1* in the normal and tumour samples from COAD. **d**, Number of tumour samples with over- and under-spliced 3′ UTRs for genes with c3USP.

We showed that over-splicing of the *CTNNB1* 3′ UTR was significantly associated with the upregulation of WNT signalling and cell cycle genes in both TCGA-LIHC and PLANet tumour samples (Fig. 5a and Extended Data Fig. 6a). To investigate the functional effects of *CTNNB1* 3′ UTR splicing, we employed a complementary approach to the ASOs using custom-designed short interfering RNAs (siRNAs) that specifically targeted the CDS, the intron of 3′ FL and the unique splice junction of 3′ SP. These siRNAs efficiently and specifically downregulated the expression of their respective transcripts (Fig. 5b, Extended Data Fig. 6b, Supplementary Table 6 and Supplementary Note). Additionally, 3′ SP depletion significantly reduced *CTNNB1* CDS transcript and protein expression compared with 3′ FL knockdown (Fig. 5b,c and Extended Data Fig. 6b,c), suggesting that the CTNNB1 protein is primarily expressed from the 3′ SP transcript. Consistent with the effect of the splice site-targeting ASO, we observed si-CDS and si-3′ SP-mediated reduction in cell growth and migration (Fig. 5d and Extended Data Fig. 6d,e), and a lack of tumour growth in mouse xenografts (Fig. 5e). These could partially be due to the diminished expression of WNT target genes, such as *AXIN2*, *MYC* and *TCF7*, as well as cell cycle markers CDK2, CDK4, CDK6 and CCNE1, upon the siRNA- and ASO-mediated downregulation of *CTNNB1* 3′ SP, in line with the GSEA results (Fig. 5f,g and Extended Data Fig. 6f,g). We further employed the

clustered regularly interspaced short palindromic repeats (CRISPR)–Cas9 system to mutate the *CTNNB1* 3′ UTR splice site (GT > GG) at the genomic level in Hep3B cells (Extended Data Fig. 6h). This led to a significant downregulation of the 3′ SP transcript expression compared with the CDS and 3′ FL transcripts, and a simultaneous decrease in CTNNB1 protein expression and cell proliferation (Fig. 5h,i and Extended Data Fig. 6i). Additionally, we verified these knockdown effects with additional siRNAs (Extended Data Fig. 7) and further tested the same ASOs and siRNAs in the COAD cell line DLD-1 since global 3′ UTR splicing is significantly increased in COAD and associated with poorer OS, while *CTNNB1* 3′ UTR splicing is also upregulated in COAD (Extended Data Fig. 8a–e). These resulted in similar phenotypic effects to those in the HCC cells (Extended Data Fig. 8f–m). Collectively, these findings suggest that the *CTNNB1* 3′ SP transcript is the predominantly translated isoform and highlight the critical role that the 3′ SP variant plays in the regulation of *CTNNB1* expression and oncogenic function.

**3′ UTR splicing may enhance CTNNB1 expression to promote tumourigenesis.** Next, we analysed ENCODE RNA-seq data and identified hundreds of 3USPs that were up- or downregulated upon knockdown of different RBPs (Extended Data Fig. 9a). We integrated this with crosslinking and immunoprecipitation (CLIP)-seq

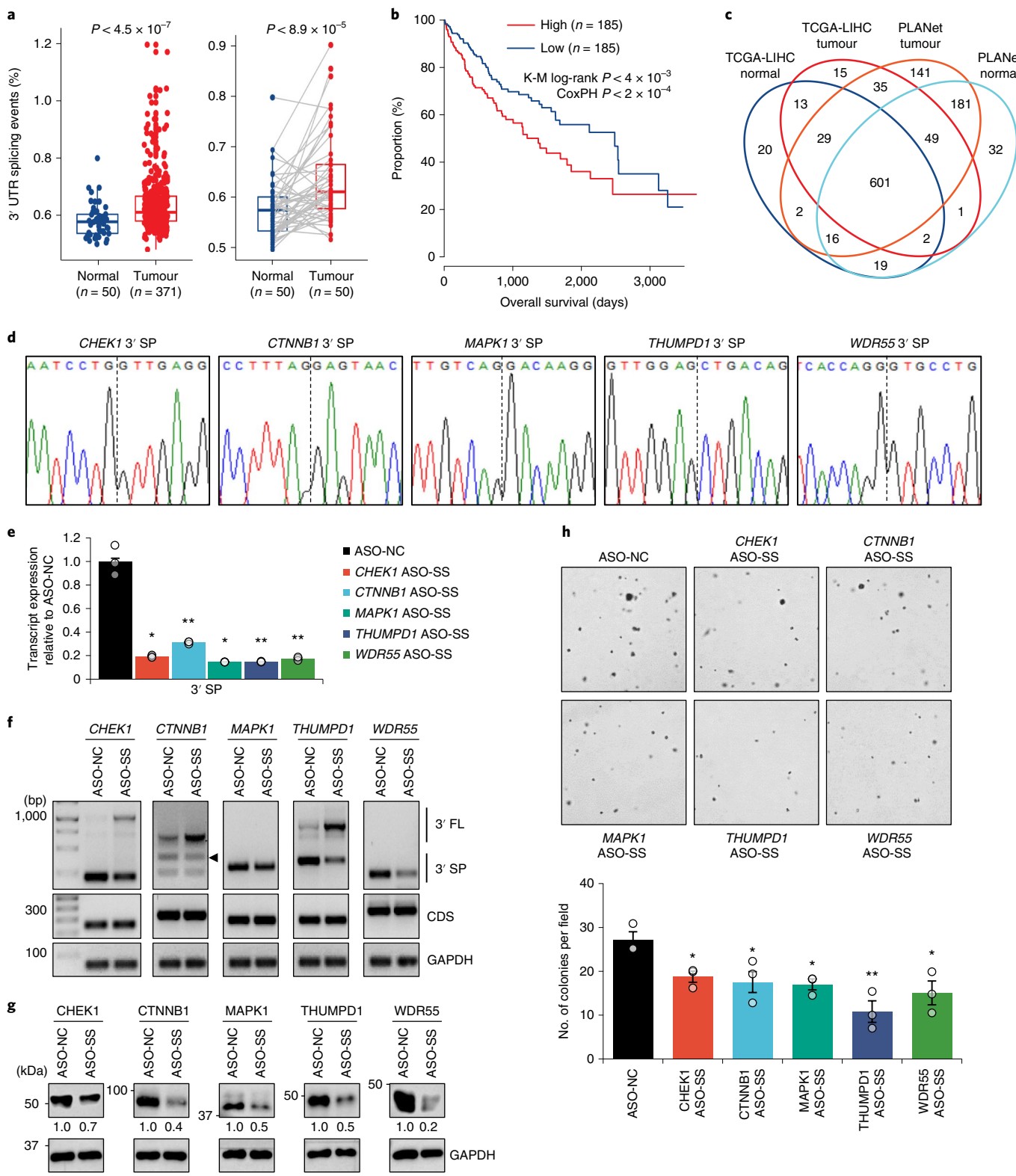

**Fig. 3 | Targeted inhibition of 3′ UTR splicing impedes HCC carcinogenesis. a**, Percentage of 3USPs relative to the total splicing events in all (left) and 50 matched normal-tumour samples (right) from TCGA-LIHC dataset. *P* values: Mann–Whitney *U* test. **b**, Kaplan–Meier (K-M) survival analysis of TCGA-LIHC samples (top and bottom half ranked by numbers of 3USPs). CoxPH, Cox proportional-hazards regression model. **c**, Venn diagram showing the overlap between c3USPs identified in TCGA and PLANet normal and tumour samples. **d**, Sanger sequencing validation of the 3′ UTR splice junctions. **e–h**, Effect of ASO-mediated blocking of the 3′ UTR splice site on candidate transcript expression by qPCR (*n* = 3 independent experiments) (**e**), PCR (arrowhead, 3′ SP2; see Supplementary Note) (**f**), protein expression (**g**) and anchorage-independent growth (*n* = 3 independent experiments) (**h**) in Hep3B. ASO-NC, non-targeting control; ASO-SS, splice site ASO. In **e** and **h**: mean ± standard error of the mean (s.e.m.); unpaired Student's *t*-test *\*P* < 0.05, *\*\*P* < 0.01 and *\*\*\*P* < 0.001. In **f** and **g**: data represent three independent experiments.

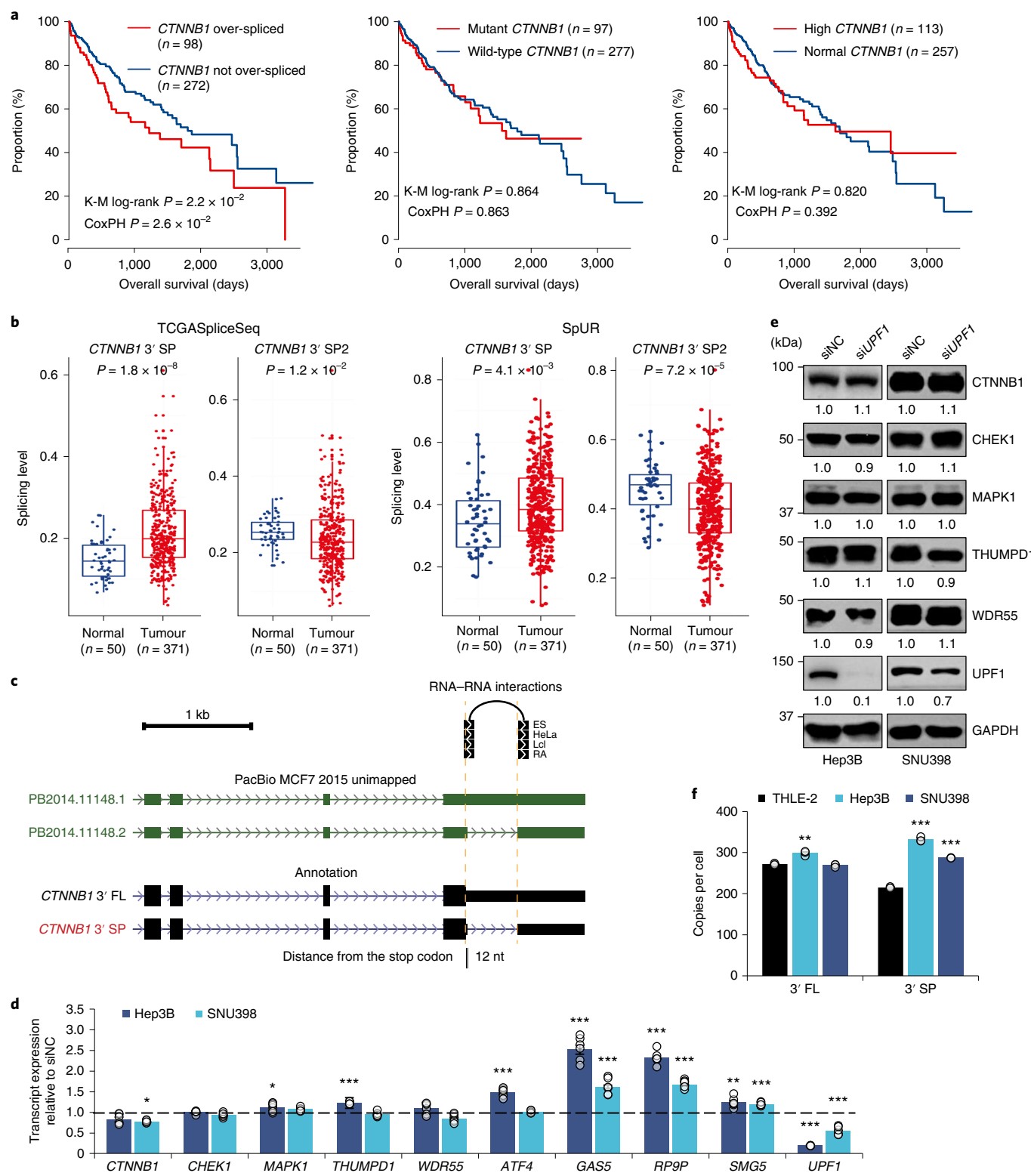

**Fig. 4 | Analysis of the *CTNNB1* spliced 3′ UTR in HCC. a**, Survival analysis of TCGA-LIHC samples by *CTNNB1* 3′ UTR SPLs (left), somatic mutation status (middle) and transcript expression (right). *n*, number of patients analysed. K-M, Kaplan-Meier; CoxPH, Cox proportional-hazards regression model. **b**, SPLs of *CTNNB1* 3′ SP and 3′ SP2 in normal and HCC tumour samples from the TCGASpliceSeq database (left) and SpUR (right). *n*, number of RNA-seq samples analysed. **c**, Genome browser tracks depicting splicing in the *CTNNB1* 3′ UTR, PacBio long-read sequencing data, RNA–RNA interactions and distance between the splice junction and stop codon. **d,e**, Effect of *UPF1* knockdown on candidate 3′ SP and positive controls (*ATF4*, *GAS5*, *RP9P* and *SMG5*) transcript (*n* = 3 independent experiments) (**d**) and candidate protein (data represent three independent experiments) (**e**) expression. **f**, *CTNNB1* 3′ FL and 3′ SP copy number quantification in THLE-2 and HCC cell lines, Hep3B and SNU398 (*n* = 3 independent experiments). siNC, siRNA non-targeting control. In **d** and **f**: mean ± s.e.m.; unpaired Student's *t*-test *$P < 0.05$, **$P < 0.01$ and ***$P < 0.001$.

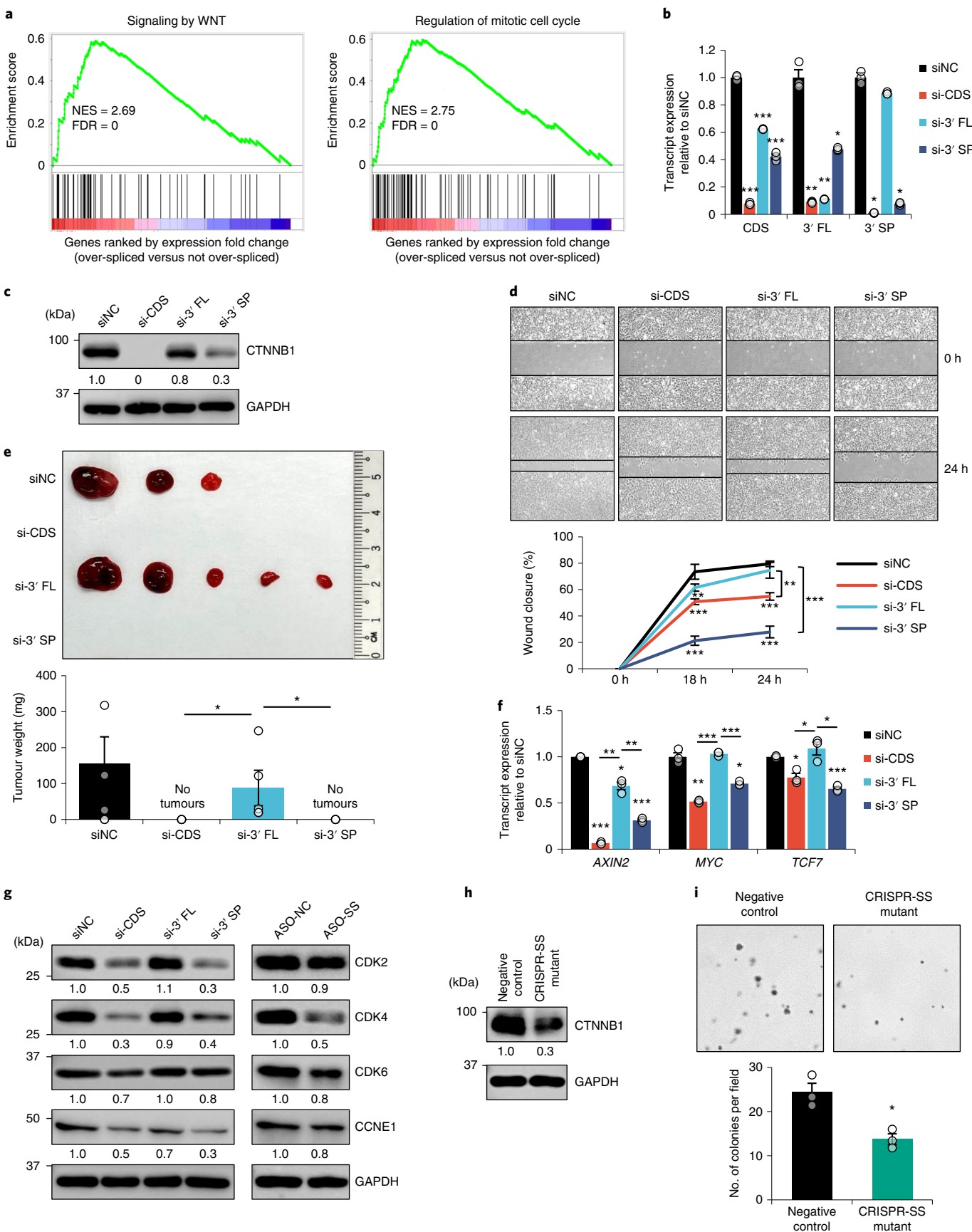

**Fig. 5 | 3′ UTR splicing of *CTNNB1* promotes tumourigenesis. a**, GSEA comparing two groups of tumours showing the enrichment of upregulated genes from the gene sets of WNT signalling and mitotic cell cycle in the TCGA-LIHC tumour samples with *CTNNB1* 3′ UTR over-splicing. **b–f**, Effect of siRNA-mediated knockdown of *CTNNB1* CDS, 3′ FL and 3′ SP on *CTNNB1* transcript (*n* = 3 independent experiments) (**b**) and protein (**c**) expression, cell migration (*n* = 3 independent experiments) (**d**), xenograft tumour growth (*n* = 5 mice) (**e**) and WNT target transcript expression (*n* = 3 independent experiments) (**f**) in Hep3B. **g**, Effect of siRNA-mediated *CTNNB1* knockdown and ASO-mediated 3′ UTR splicing inhibition on the protein expression of cell cycle markers. **h,i**, Effect of the CRISPR–Cas9-mediated genomic mutation of the *CTNNB1* 3′ UTR splice site (CRISPR-SS mutant) on CTNNB1 protein expression (**h**) and anchorage-independent growth (*n* = 3 independent experiments) (**i**) in Hep3B. In **b**, **d–f** and **i**: mean ± s.e.m.; unpaired Student's *t*-test *P < 0.05, **P < 0.01 and ***P < 0.001. In **c**, **g** and **h**: data represent three independent experiments.

data[16], and selected RBPs with putative binding sites within the terminal exon of *CTNNB1* (including both the CDS and unspliced 3′ UTR) and/or whose knockdown resulted in significant changes in *CTNNB1* 3′ SP for further validation (Fig. 6a and Supplementary Table 7). Only siRNA-mediated knockdown of *SF3B1*, *SRSF1* and *U2AF2* consistently downregulated the 3′ SP transcript, concomitantly increased the 3′ FL transcript and reduced CTNNB1 protein expression without affecting the four CDS exon–exon junctions tested (Fig. 6b,c and Extended Data Fig. 9b–e). We also showed that RNA immunoprecipitation (RIP) of SRSF1 and U2AF2 significantly enriched for both *CTNNB1* 3′ UTR variants, while SF3B1 RIP enriched for only *CTNNB1* 3′ FL (Fig. 6d). We further verified these associations by pulling down the *CTNNB1* transcripts whereby SRSF1 and U2AF2 were enriched by the antisense 3′ FL and 3′ SP probes, and consistent with the RIP results, enrichment of SF3B1 was observed only for the 3′ FL pulldown (Fig. 6e). These observations suggest that these RBPs may associate with the *CTNNB1* 3′ UTR and modulate its splicing.

To investigate whether the 3′ UTR variants may exhibit varying phenotypic effects due to their differential regulation of *CTNNB1* expression, we first assessed the protein expression of CTNNB1 overexpressed from constructs containing *CTNNB1* CDS tagged to each 3′ UTR variant (Extended Data Fig. 9f). Despite similar transcript levels, we observed higher CTNNB1 protein expression from the 3′ SP variant (Fig. 6f and Extended Data Fig. 9g). Next, we performed a luciferase reporter assay using reporter constructs containing the different 3′ UTR variants. 3′ SP significantly increased luciferase activity compared with 3′ FL (Fig. 6g). Similar results were also observed in COAD cells (Extended Data Fig. 10a–c). We first tested whether 3′ UTR splicing regulated *CTNNB1* expression at the transcript or protein level by inhibiting transcription or translation following the overexpression of HA-tagged *CTNNB1* variants. The transcript and protein expression of both 3′ FL and 3′ SP variants were similarly changed (Extended Data Fig. 10d,e), contrary to a previous study that demonstrated a longer mRNA half-life for *CTNNB1* 3′ SP in HeLa cells, which could be due to tissue-specific regulation[17]. We further inhibited proteasomal degradation and did not observe differential CTNNB1 protein stability (Extended Data Fig. 10f).

Next, we performed the translation reporter assay to investigate the effect of the *CTNNB1* 3′ UTR variants on translation efficiency and observed a significant increase in the 3′ SP luciferase signal compared with that of 3′ FL (Fig. 6h), implying that the 3′ SP variant may be preferentially translated. As this assay relies on exogenously expressed constructs, we also performed polysome profiling to detect translation efficiency of the endogenous *CTNNB1* transcript

variants. In contrast to the translation reporter assay, polysome profiling for Hep3B and SNU398 cells showed that the 3′ FL and 3′ SP transcripts are similarly distributed across the polysome fractions, suggesting the transcript variants present in the cytoplasm are equally translated (Fig. 6i). This discrepancy could be due to several factors: (1) the luciferase ORF (~1 kb) is much smaller than that of *CTNNB1* (~3 kb), which could carry additional components that influence its splicing, folding and/or translation, and (2) the luciferase reporters are exogenously expressed, whereas the polysome profiles measure endogenous levels of *CTNNB1* and may be more representative of physiological conditions.

Previous studies have demonstrated nuclear retention of intron-containing transcripts by the U1 small nuclear ribonucleoprotein (snRNP), a component of the RNA spliceosome, to regulate the efficient expression of protein-coding mRNAs[18–20]. We postulated that the variation in protein expression from the *CTNNB1* 3′ UTR variants may be attributed to their different transcript localization. To investigate this, we performed U1 RIP. Only the 3′ FL transcript was significantly enriched, while 3′ SP was undetected (Fig. 7a). Consistently, both nuclear–cytoplasmic fractionation and RNA fluorescence in situ hybridization (FISH) data showed that the 3′ SP transcripts were predominantly cytoplasmic, whereas the 3′ FL transcripts were mainly nuclear (Fig. 7b,c), which we also observed for *CHEK1* (Extended Data Fig. 10g,h). These findings suggest that nuclear retention of the intron-containing 3′ FL transcripts may contribute to their reduced availability for the cytoplasmic translational machinery, resulting in decreased protein expression.

To further interrogate the importance of 3′ UTR splicing for *CTNNB1* expression, we mutated the 5′ splice site (5′ SSmut) of the *CTNNB1* 3′ FL plasmid constructs. Overexpression of *CTNNB1* 5′ SSmut resulted in CTNNB1 protein levels higher than that of wild-type 3′ FL (3′ FL-WT) and comparable to 3′ SP (Fig. 8a). It also significantly increased luciferase activity compared with 3′ FL-WT in the luciferase reporter assay, but only in SNU398, and not in the translation reporter assay (Fig. 8b,c), suggesting that the 5′ SS mutation does not confer translational advantage. This is supported by the polysome profile of cells treated with splice site-blocking ASOs showing comparable distributions of the 3′ FL and 3′ SP variants across the polysome fractions compared with the control (Fig. 8d). Taken together, these findings indicate that differential cellular localization of the *CTNNB1* 3′ UTR variants could be the predominant factor impacting CTNNB1 protein expression. *CTNNB1* may be primarily translated from the 3′ SP transcripts that are exported to the cytoplasm upon splicing, highlighting the importance of 3′ UTR splicing in driving oncogene expression and cancer progression (Fig. 8e).

**Fig. 6 | 3′ UTR splicing may enhance CTNNB1 expression to promote tumourigenesis. a**, Volcano plot showing changes in *CTNNB1* 3′ UTR splicing upon RBP knockdown. Putative binding was predicted using eCLIP-seq data. **b,c**, Effect of splicing RBP knockdowns on *CTNNB1* transcript (*n* = 3 independent experiments) (**b**) and protein (**c**) expression in Hep3B. **d**, Enrichment of *CTNNB1* transcripts by RBP RIP in Hep3B and SNU398 (*n* = 4 independent experiments). **e**, Enrichment of RBPs upon *CTNNB1* 3′ FL and 3′ SP pulldown using biotinylated probes. **f**, Effect of overexpressing *CTNNB1* CDS, CDS + 3′ FL and CDS + 3′ SP on exogenous CTNNB1 protein expression. **g,h**, Luciferase activity of plasmid-transfected (*n* = 4 independent experiments) (**g**) and RNA-transfected (*n* = 5 independent experiments) (**h**) *CTNNB1* reporter constructs. **i**, Polysome profiles for *CTNNB1* 3′ FL and 3′ SP in Hep3B and SNU398 (*n* = 2 independent experiments). *HSP90* and *TBP* are housekeeping controls. S, sense; AS, antisense; EV, empty vector. In **b**, **d**, **g** and **h**: mean ± s.e.m.; unpaired Student's *t*-test *P < 0.05, **P < 0.01 and ***P < 0.001. In **c**, **e** and **f**: data represent three independent experiments.

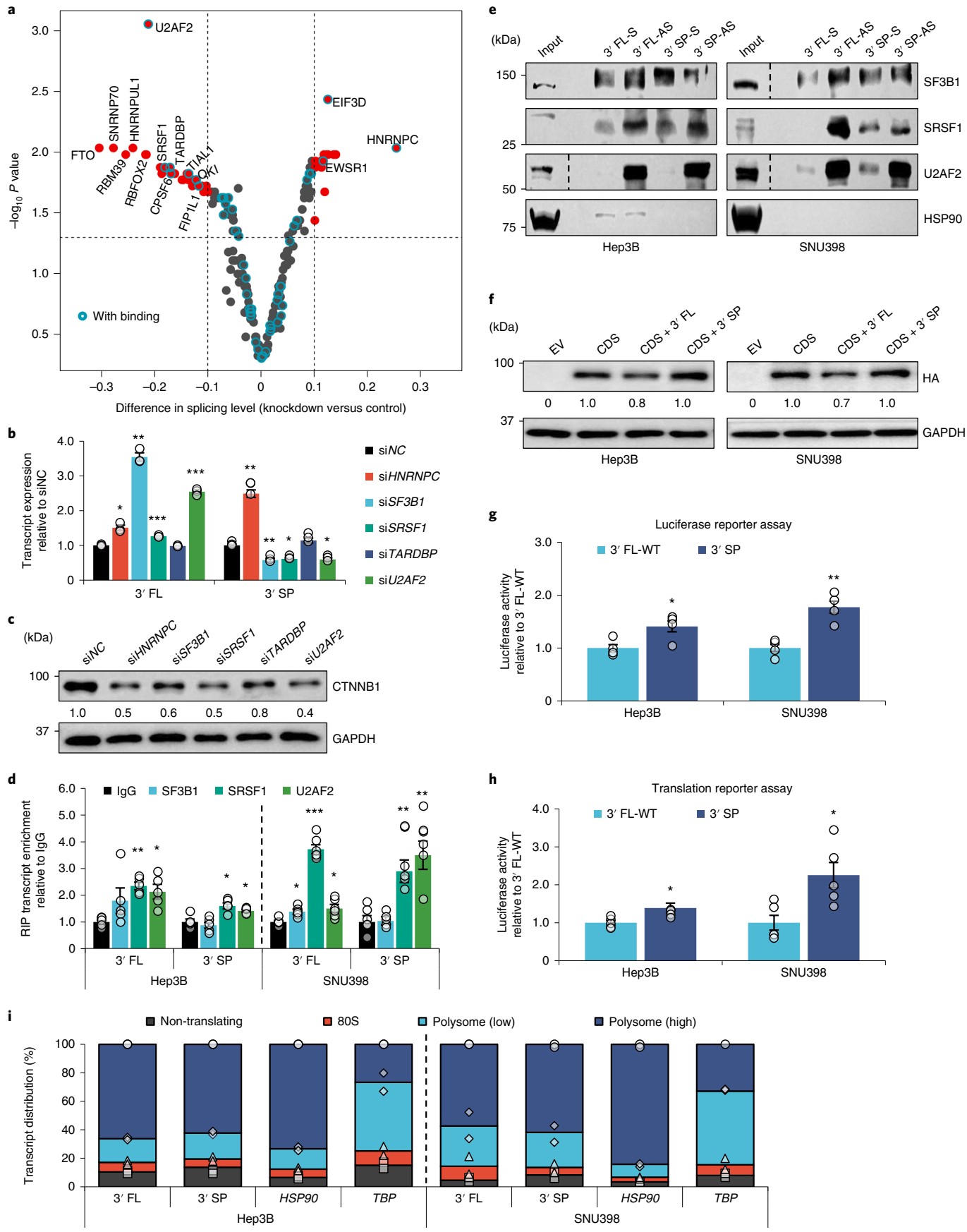

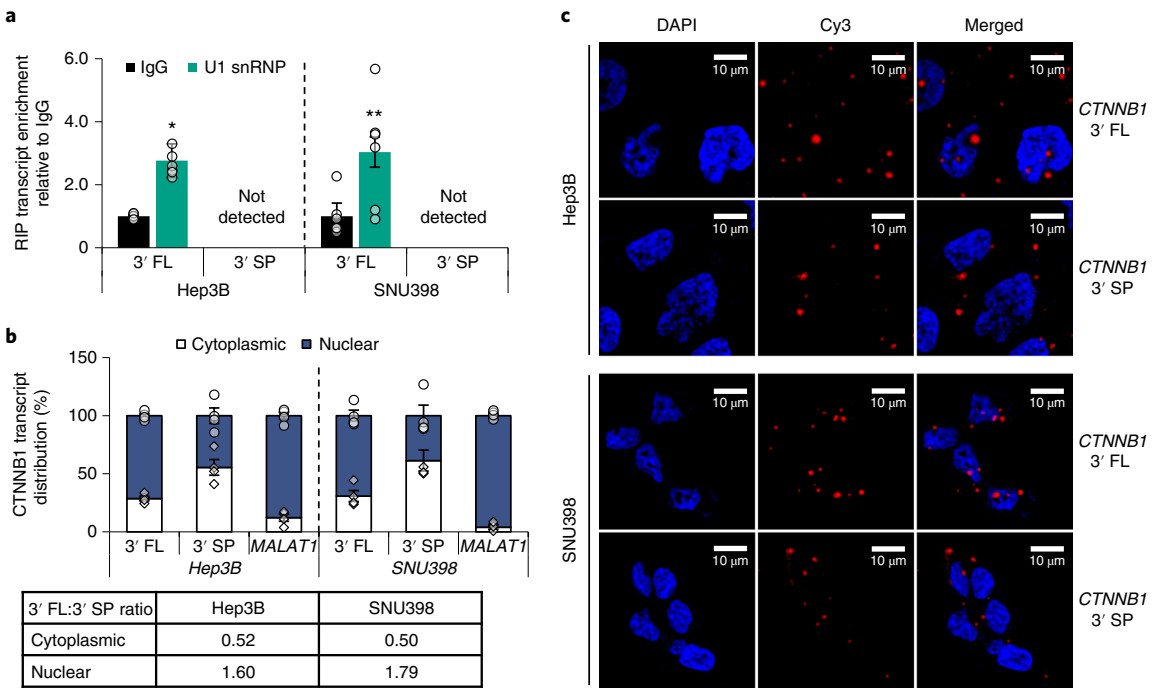

**Fig. 7 | 3′ UTR splicing-mediated cytoplasmic localization may enhance CTNNB1 expression. a**, Enrichment of the *CTNNB1* 3′ FL transcript by U1 snRNP RIP (*n* = 3 independent experiments). **b**, Subcellular distribution of *CTNNB1* 3′ FL and 3′ SP transcripts following nuclear–cytoplasmic fractionation of Hep3B and SNU398 cells. *MALAT1* was used as a nuclear control. The 3′ FL:3′ SP transcript ratios in each cellular compartment are presented in the table below (*n* = 4 independent experiments). **c**, RNA–FISH showing transcript localization of *CTNNB1* 3′ FL and 3′ SP in Hep3B and SNU398. Data represent three independent experiments. In **a** and **b**: mean ± s.e.m.; unpaired Student's *t*-test *P* < 0.05, **P* < 0.01 and ***P* < 0.001.

## Discussion

Multiple studies have shown that aberrant splicing in cancer confers proliferation, migratory and drug resistance advantages to cancer cells[10,21]. However, these have mostly focused on splicing events in coding exons as 3′ UTR splicing was often thought to trigger NMD[22]. Here we build SpUR, a database to comprehensively characterize 3USPs in human cancers and their corresponding normal tissues (http://www.cbrc.kaust.edu.sa/spur/home). We reveal that 3′ UTR splicing is widespread, upregulated in cancer, correlated with poor prognosis and more prevalent in oncogenes. We demonstrate the physiological, functional and clinical relevance of the spliced 3′ UTR of the key oncogene *CTNNB1*. We show that *CTNNB1* is over-spliced in ~40% of tumour samples in ten cancer types, and its spliced 3′ UTR (1) is a more robust prognostic indicator compared with its transcript expression and somatic mutational status in HCC; (2) is not an NMD target; (3) promotes cell proliferation and migration; and (4) enhances protein expression potentially through its cytoplasmic localization (Fig. 8e). Furthermore, these properties extend to 3′ UTR variants of other genes, such as *CHEK1*. Critically, dysregulated 3USPs may also play key roles in other cancer types, including both solid tumours and haematological malignancies. The low overlap of c3USPs between AML and the solid tumours is noteworthy and may reflect potential intrinsic differences between blood and solid tumours at the genomic level. Further work on other haematological malignances will provide a better understanding of these variations.

As 3′ UTRs carry regulatory elements critical for modulating RNA metabolism and even protein activity[23,24], deregulated 3′ UTR splicing could have other mechanistic and functional consequences. The loss of 3′ UTR regulatory elements and binding sites through splicing coupled with possible splicing-mediated changes in RNA secondary structures could significantly disrupt molecular interactions, such as those with microRNAs and RBPs, and their regulatory effects. Additionally, we show that 3′ UTRs that undergo splicing

contain introns enriched in Alu elements, which are known to facilitate splicing and RNA editing[25,26]. Potential crosstalk between these two RNA processing steps that are highly dysregulated in cancer could further disrupt gene expression to drive tumourigenesis[10,21,27].

In recent years, the use of RNA-based therapeutics has been gaining momentum. In particular, many ASOs are undergoing clinical trials for the treatment of various medical conditions. A handful of anti-cancer ASOs, such as Danvatirsen (AZD1950) and Travedersen (OT-101), which target *STAT3* and *TGF-β2*, respectively, have had varying levels of success[28]. Here we demonstrate the use of ASOs to manipulate 3′ UTR splicing to repress oncogene expression and cancer cell proliferation. Chemically modified ASOs have been successfully delivered via different routes of administration and shown to be active in various tissues, making them an attractive treatment option for different cancers[29], Moreover, the specific upregulation of 3′ UTR splicing in cancer suggests that these ASOs could possess therapeutic potential with minimal effects in normal cells, which are beneficial properties for the development of ASO-based anti-cancer drugs.

Multiple studies have identified a large repertoire of RBPs and demonstrated their essential roles in a diverse range of regulatory processes[30]. A recent ENCODE study has gone a step further to construct their binding and functional maps from multiple eCLIP datasets[16], which could facilitate easier identification of 3′ UTR-specific splicing factors that can be targeted for cancer treatment. For example, SRSF1, which can potentially modulate *CTNNB1* 3′ UTR splicing, was targeted using decoy RNA oligonucleotides to dampen its activity in a recent study[31]. With the relative simplicity of designing RNA-based therapies using base-pairing complementarity and the rapid advancement in drug delivery strategies, oligonucleotide-based drugs that can efficiently target cancer-specific 3′ UTR splicing, 3′ UTR spliced variants or the splicing factors involved could be a game changer in the field of cancer therapeutics.

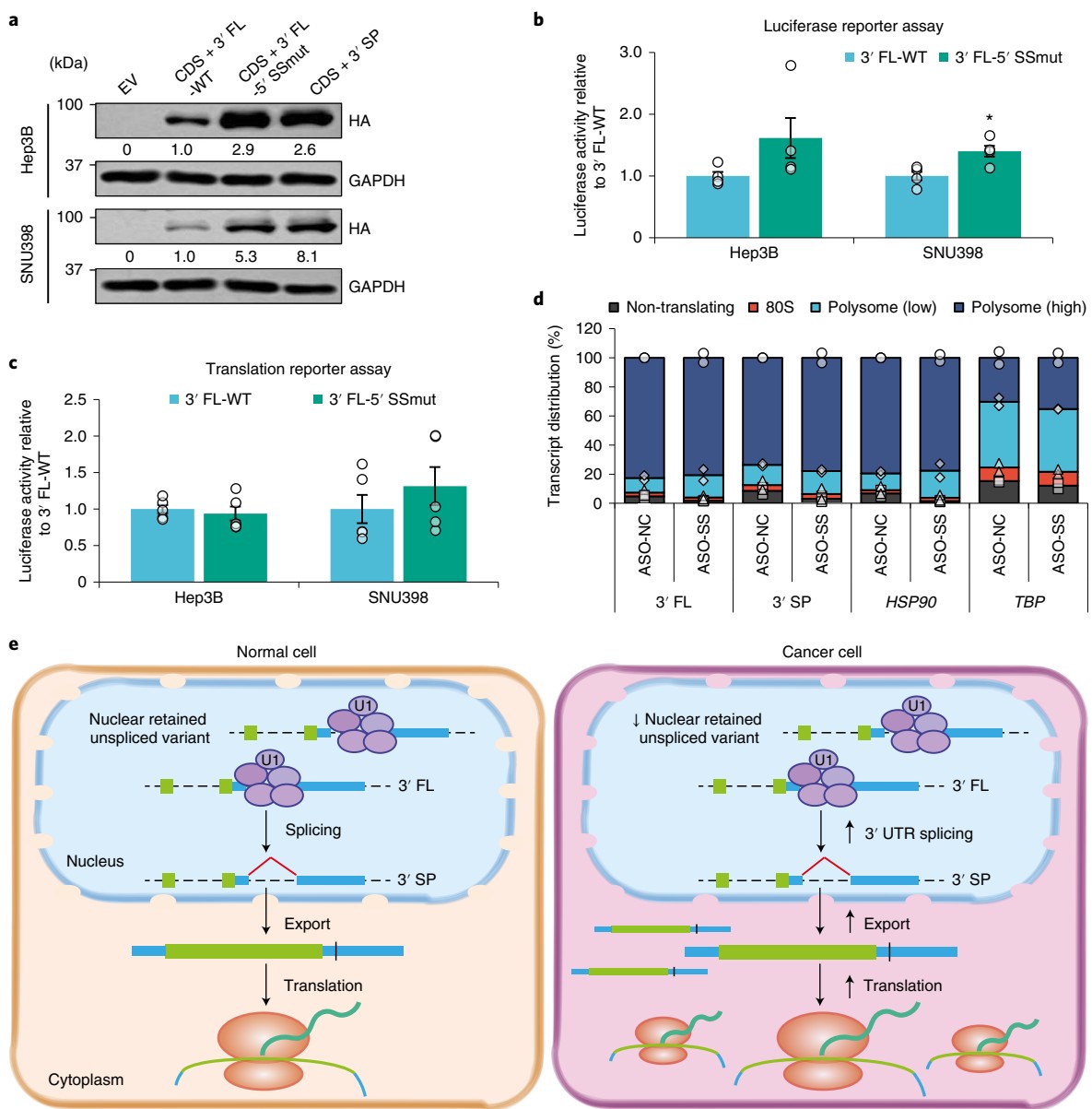

**Fig. 8 | Upregulated 3′ UTR splicing may promote carcinogenesis. a**, Effect of overexpressing *CTNNB1* CDS + 3′ FL-WT, CDS + 3′ FL-5′ SSmut and CDS + 3′ SP on exogenous CTNNB1 protein expression. Data represent three independent experiments. **b,c**, Luciferase activity of plasmid-transfected (*n* = 4 independent experiments) (**b**) and RNA-transfected (*n* = 5 independent experiments) (**c**) *CTNNB1* 3′ FL-WT and 3′ FL-5′ SSmut reporter constructs. **d**, Effect of ASO-mediated blocking of the *CTNNB1* 3′ UTR splice site on the polysome profiles for *CTNNB1* (*n* = 2 independent experiments). HSP90 and TBP are housekeeping controls. **e**, Schematic depicting how increased 3′ UTR splicing may promote carcinogenesis. In **b** and **c**: mean ± s.e.m.; unpaired Student's *t*-test *$P < 0.05$, **$P < 0.01$ and ***$P < 0.001$.

3′ UTR splicing could be a widespread mechanism that cancer cells exploit to generate NMD- and SMD-insensitive and intronless transcripts that are effectively exported to the cytoplasm to promote oncogene expression and tumourigenesis. These findings provide key insights into our understanding of this poorly characterized facet of RNA processing and its contribution to transcriptome heterogeneity and carcinogenesis. In particular, we provide evidence that specific targeting of 3′ UTR splicing could effectively attenuate the tumourigenic phenotype of key oncogenes. Furthermore, the upregulation of 3′ UTR splicing in various cancers and its significant correlation with prognosis suggest that its detection and targeting may represent new avenues for the development of more targeted diagnostics and therapeutics.

## Online content

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

¹Cancer Science Institute of Singapore, National University of Singapore, Singapore, Singapore. ²Computer Science Program, Computer, Electrical and Mathematical Sciences and Engineering Division, King Abdullah University of Science and Technology (KAUST), Thuwal, Saudi Arabia. ³KAUST Computational Bioscience Research Center, King Abdullah University of Science and Technology, Thuwal, Saudi Arabia. ⁴A*STAR Skin Research Labs and Skin Research Institute of Singapore, A*STAR, Immunos, Singapore, Singapore. ⁵School of Biological Sciences, Nanyang Technological University, Singapore, Singapore. ⁶Department of Haematology-Oncology, National University Cancer Institute of Singapore, National University Health System, Singapore, Singapore. ⁷Cancer and Stem Cell Biology Program, Duke-NUS Medical School, Singapore, Singapore. ⁸Yong Loo Lin School of Medicine, National University of Singapore, Singapore, Singapore. ⁹The N.1 Institute for Health, National University of Singapore, Singapore, Singapore. ¹⁰Department of Pathology, National University Health System, Singapore, Singapore. ¹¹Department of Surgery, Yong Loo Lin School of Medicine, National University of Singapore, Singapore, Singapore. ¹²Division of Colorectal Surgery, University Surgical Cluster, National University Health System, Singapore, Singapore. ¹³Department of Medicine, Yong Loo Lin School of Medicine, National University of Singapore, Singapore, Singapore. ¹⁴Department of Orthopaedic Surgery, Yong Loo Lin School of Medicine, National University of Singapore, Singapore, Singapore. ¹⁵Division of Hepatobiliary & Pancreatic Surgery, University Surgical Cluster, National University Health System, Singapore, Singapore. ¹⁶Program in Clinical and Translational Liver Cancer Research, National Cancer Center Singapore, Singapore, Singapore. ¹⁷Genome Institute of Singapore, Agency for Science, Technology and Research (A*STAR), Singapore, Singapore. ¹⁸Department of Surgery, Faculty of Medicine, University of Malaya, Kuala Lumpur, Malaysia. ¹⁹Hepato-Pancreato-Biliary Surgery Unit, Department of Surgery, National Cancer Institute, Bangkok, Thailand. ²⁰Department of Hepatopancreatobiliary and Transplantation Surgery, Singapore General Hospital and National Cancer Center Singapore, Singapore, Singapore. ²¹Key Laboratory of Zoological Systematics and Evolution, Institute of Zoology, Chinese Academy of Sciences, Beijing, China. ²²Institute of Molecular and Cell Biology, Agency for Science, Technology and Research (A*STAR), Singapore, Singapore. ²³NUS Centre for Cancer Research, Yong Loo Lin School of Medicine, National University of Singapore, Singapore, Singapore. ²⁴Center for Therapeutics Discovery, Department of Oncological Sciences and Pharmacological Sciences, Tisch Cancer Institute, Icahn School of Medicine at Mount Sinai, New York City, NY, USA. ²⁵Department of Anatomy, Yong Loo Lin School of Medicine, National University of Singapore, Singapore, Singapore. ²⁶BioMap, Beijing, China. ²⁷Academic Clinical Programme for Surgery, SingHealth Duke-NUS Academic Medical Centre (AMC), Singapore, Singapore. ²⁸Department of Biochemistry, Yong Loo Lin School of Medicine, National University of Singapore, Singapore, Singapore. ²⁹Present address: Division of Pulmonary and Critical Care Medicine, Department of Medicine, Boston University, Boston, MA, USA. ³⁰Present address: The University of British Columbia, Vancouver, British Columbia, Canada. ³¹Present address: Integra Therapeutics S.L., Barcelona, Spain. ³²These authors contributed equally: Jia Jia Chan, Bin Zhang. ✉e-mail: yvonnetay@nus.edu.sg

## Methods

This study complies with all relevant ethical regulations. The human studies were approved by the following institutional review boards (IRBs): the Domain Specific Review Board under the National Healthcare Group in Singapore, the Central Institution Review Board (CIRB) of SingHealth, of which all National Cancer Center Singapore, Singapore General Hospital and National University Hospital were constituent members (CIRB Ref: 2016/2626 and 2018/2112), Medical Research Ethics Committee of UMMC (MREC ID number 201713-4729) and Research Committee of National Cancer Institute Thailand (project number 174_2017C_OUT504). Animal protocols were approved by the National University of Singapore (NUS) Institutional Animal Care and Use Committee (IACUC).

**Human studies.** The protocols for the human studies are approved by the IRBs listed above. Each patient gave informed written consent. The AML study includes 18 AML patients (10 females and 8 males) in the age range of 22–75 years and 15 patients undergoing total knee replacements (11 females and 4 males) in the age range of 47–84 years. The PLANet study includes 46 patients (13 females and 33 males) in the age range of 47–87 years. There is no patient information available for the in-house HCC dataset as all patients in this study have been de-identified.

**Identification of 3USPs.** RNA-seq data (fastq files) of 7,917 samples from TCGA and GTEx were downloaded from dbGaP repository (2016) and aligned to the reference human genome (hg19) using STAR (v2.5.2a)[32]. In total, ten TCGA cancer types with more than 30 adjacent normal samples were selected for analysis. The derived splicing junction (splicing-out) reads were filtered and merged to identify all introns. For each defined intron, the splicing-in reads (covering the splice site by at least 6 nt) were counted by featureCounts (v1.6.1) and SPLs were calculated using the following formula:

$$\text{Splicing level (SPL)} = \frac{\text{splicing} - \text{out reads}}{\text{splicing} - \text{out reads} + \text{splicing} - \text{in reads}/2}$$

To identify putative splicing events in 3′ UTRs, each intron was overlapped with annotations from GENCODE (v23) (ref. [33]). Only introns specifically located within 3′ UTRs that did not overlap with any annotated CDS or 5′ UTR were retained. Within these introns, those supported by at least two junction reads and exhibited SPLs >10% in at least one sample, were selected as putative 3USPs. The same approach was applied to identify and quantify the 3USPs in the AML, PLANet and in-house HCC RNA-seq samples. All clinical samples obtained from human research participants were done in accordance with the protocols approved by the relevant IRBs in Singapore.

For each RNA-seq sample, we counted both the total number of 3USPs and all splicing events including those in the 5′ UTR, 3′ UTR, CDS and non-coding RNAs. To exclude the influence of transcriptional activity in quantifying the number of 3USPs, the number of 3USPs in each sample was normalized by dividing it with the number of all splicing events. To account for the variability caused by different sample sizes, sequencing depths and read lengths, we further defined common 3′ UTR splicing events (c3USPs) as the 3USPs exhibiting SPLs >10% in more than half of the samples in each TCGA cancer and GTEx tissue type. This was also applied to the AML and PLANet datasets, but not to the in-house data owing to its small sample size ($n = 4$).

**Analysis of dysregulated 3′ UTR splicing in cancer.** To identify differentially spliced 3′ UTRs between tumour and normal samples, we used two approaches to analyse the TCGA, AML, PLANet and in-house datasets owing to the different numbers of samples (5,543 TCGA-tumour samples in 10 cancers, 34 AML, 165 PLANet HCC and 4 pairs of in-house HCC tumour samples). For each TCGA cancer cohort, we compared the SPL of each c3USP between the tumour and adjacent normal samples (TCGA-tumour and TCGA-normal) using the Mann–Whitney U test. The Benjamini–Hochberg method was used for multiple test adjustment (false discovery rate (FDR) <0.1). The same approach was applied to the AML and PLANet HCC samples with the following cut-offs: FDR <0.1 and median SPL difference between tumour and normal >5%. Significant candidates in each TCGA cancer type were filtered on the basis of the criteria: (1) unidirectional median SPL changes between TCGA-tumour and TCGA-normal, and TCGA-tumour and GTEx, and (2) median SPL difference of >5% between TCGA-tumour and TCGA-normal or TCGA-tumour and GTEx. For the in-house dataset, we applied a method optimized for the detection and quantification of splicing differences between tumour and normal samples for a small sample size as previously described[34]. Significance was determined using permutation-derived FDR (<0.1) and the median SPL difference between tumour and normal (>5%).

To identify tumour samples that exhibit over-splicing for a given significantly dysregulated c3USP in each cancer type, the SPLs in each tumour sample were compared with the SPLs from the corresponding TCGA-normal and GTEx samples. The dashed line in the bottom left panel of Fig. 2a indicates the 90% quantile cut-off of SPLs in the normal samples (TCGA-normal and GTEx samples were analysed separately, and the higher value was used for enhanced stringency). Tumour samples with SPLs higher than the 90% quantile cut-off were considered over-spliced for a given c3USP. Under-splicing was similarly defined as SPLs lower than the 10% quantile of the normal samples.

**Survival analysis.** SPLs of each c3USP in each cancer type were correlated with patient survival using the Cox proportional hazards regression model. The Kaplan–Meier method was applied by splitting patients into high and low groups according to their SPLs (top and bottom halves ranked by SPL). The same approach was applied to investigate the prognostic effect of general splicing in each cancer type using the total number of 3USPs and all splicing events. For the Kaplan–Meier curves of *CTNNB1* in liver cancer, tumour samples in which *CTNNB1* was mutated, overexpressed or the 3′ UTR of *CTNNB1* was over-spliced were compared with the samples in which these phenomena are absent, respectively. Therein, overexpression and over-splicing were defined by transcript expression or SPLs greater than 90% quantile of that in the normal samples as described in the section 'Analysis of dysregulated 3′ UTR splicing in cancer'. The mutant *CTNNB1* samples included only missense mutations while the wild-type samples did not contain any genetic alterations for *CTNNB1*.

**Processed public datasets.** The processed GFF files derived from the PacBio long-read sequencing data from liver, heart, brain and the MCF7 cancer cell line were downloaded from the PacBio IsoSeq Human Tissue and MCF7 datasets (http://datasets.pacb.com.s3.amazonaws.com/2014/). Owing to the shallow sequencing depth, we combined all the identified transcripts from these four cell types. The 3USPs that overlapped with the PacBio-identified isoforms with identical 5′ and 3′ splice sites were considered as being supported by PacBio. The genomic coordinates of repeat elements were downloaded from the UCSC genome browser[35] and overlapped with introns in 3′ UTRs using BEDTools (v2.29) (ref. [36]). The putative binding sites of RBPs were downloaded from the POSTAR database[37], and significant peaks identified from PAR-CLIP, HITS-CLIP, iCLIP and eCLIP were merged. RBPs related to splicing were selected on the basis of merging the annotations from GO terms and KEGG pathways as described in a previous study[38]. In total, 519 HepG2 RNA-seq samples from ENCODE, including short-hairpin-RNA-mediated knockdown of 227 RBPs and 51 control samples, were analysed. For each RBP, we compared the SPL of *CTNNB1* 3′ SP between the knockdown and control samples using the Mann–Whitney U test.

Annotations of oncogenes and tumour suppressors were derived by combining the resources from CancerMine[39] and OncoKB (Precision Oncology Knowledge Base)[40]. In total, 889 oncogenes and 878 tumour suppressors were obtained. Next, they were overlapped with genes that contained significantly upregulated and downregulated c3USPs (FDR <0.1) and the hypergeometric test was used to measure the significance of the overlap.

**Reagents.** Reagents are as follows: antibody reagents (Supplementary Table 8); TRIzol, Lipofectamine 3000, Lipofectamine RNAiMAX, Dulbecco's modified Eagle medium (DMEM), Roswell Park Memorial Institute 1640 medium (RPMI), Opti-MEM reduced serum medium, foetal bovine serum (FBS), 10× transcription buffer, NTPs (Thermo Fisher); Dharmafect 1, siGENOME and On-targetPLUS siRNA reagents (Dharmacon) (Supplementary Table 8); ASOs, Alt-R S.p. HiFi Cas9 Nuclease V3, single guide RNA, homology-directed repair template (Integrated DNA Technologies) (Supplementary Table 8); pcDNA3.1+ vector (Addgene); psiCHECK-2 vector (Promega).

**Plasmids and mutagenesis.** The *CTNNB1* 3′ UTR variants were cloned into psiCHECK-2, and the HA tag, *CTNNB1* CDS and 3′ UTR variants were cloned into pcDNA3.1+ using the primers and restriction sites listed in Supplementary Table 9. Restriction sites or linkers between the CDS and 3′ UTRs were removed using the Quikchange Lightning Multi Site-Directed Mutagenesis Kit (Agilent) as per the manufacturer's protocol. All constructs were verified by Sanger sequencing.

**Cell culture, transfection and treatments.** Human HCC cell lines Hep3B (ATCC: HB-8064) and HepG2 (ATCC: HB-8065) were cultured in DMEM and SNU398 (ATCC: CRL-2233) in RPMI. The colon adenocarcinoma (COAD) cell line DLD-1 (Horizon Discovery: HD PAR-086) was cultured in RPMI. Both DMEM and RPMI were supplemented with 10% FBS, penicillin–streptomycin and glutamine. Hep3B and DLD-1 cells express wild-type *CTNNB1*, while HepG2, SNU398 and HCT116 cells express constitutively active *CTNNB1* mutants[41,42]. The normal liver cell line, THLE-2 (ATCC: CRL-2706), was grown in BEGM Bronchial Epithelial Cell Growth Basal Medium (Lonza) supplemented with 10% FBS, 5 ng ml⁻¹ human epidermal growth factor, 70 ng ml⁻¹ phosphoethanolamine and the additives from the BEGM Bronchial Epithelial Cell Growth Medium BulletKit (except gentamycin/amphotericin and epinephrine). The normal colon cell line CCD 841 CoN and colorectal carcinoma cell line HCT116 were cultured in DMEM. The cells were maintained at 37 °C in a humidified atmosphere with 5% $CO_2$. For knockdown experiments, 150,000 cells were transfected with 50 nM of each siRNA per well in 12-well plates using Dharmafect 1 following the manufacturer's instructions. For overexpression and ASO experiments, cells were seeded at 120,000 cells per well in 12-well plates 24 h before transfecting 500–1,000 ng of each plasmid with Lipofectamine 3000 or 100 nM of each ASO with Lipofectamine RNAiMAX as per the manufacturer's protocol. For treatments, 5 μm of actinomycin D (Sigma), 355 nM of cycloheximide (Sigma) or 20 μm of MG132 (Santa Cruz) was added to cells 48 h post-transfection. Post-treatment, the cells were collected at the specified timepoints for downstream analysis.

**CRISPR–Cas9 gene editing.** For CRISPR–Cas9 gene editing experiments, cells were seeded in 12-well plates 24 h before transfecting 13.6 nM of ribonucleoprotein complexes (consisting of Cas9 and single guide RNA) and 7.8 nM of homology-directed repair template with Lipofectamine CRISPRMAX following the manufacturer's instructions. Cells were collected 48 h post-transfection and genomic DNA was extracted using the DNeasy Blood & Tissue Kit (Qiagen) as per the manufacturer's protocol for subsequent PCR and Sanger sequencing validation. Soft agar assays and RNA and protein extractions were also performed as described below.

**Soft agar assay.** Cells were transfected as described above 18–24 h before seeding. On the day of seeding, a 0.6% base agarose was prepared in 12-well plates. Transfected cells were trypsinized, resuspended and counted. A seeding density of 7,000 cells per well was used for Hep3B and SNU398, and 6,000 cells per well for HepG2. The cells were mixed in their respective growth medium and agarose to a final agarose concentration of 0.3% and added to the prepared base. Once the agarose solidified, 0.5 ml of growth medium was added to each well. The cells were maintained at 37 °C in a humidified atmosphere with 5% $CO_2$ and the growth medium was changed every 2 days. The colonies were imaged after 10–14 days under 4× magnification using the Olympus IX71 microscope and quantified using ImageJ (v1.51j8).

**Migration assay.** Cell migration assays were performed with Hep3B and SNU398 cells in wound-healing culture-insert dishes (Ibidi) and transwell chambers (Corning). The cells were transfected with siRNAs as described above and collected 48 h post-transfection. For the wound healing assay, 70,000 cells were seeded on each side of the chamber. Twenty-four hours after seeding, the insert was removed and the cells were washed three times with their corresponding medium. At the specified timepoints, imaging was performed and the percentage wound closure was measured using the CellSens software (v1.15). For the transwell experiment, 200,000 cells were seeded on the upper transwell chamber in serum-free medium as described[43]. The seeded cells were cultured for another 48 h followed by fixation, staining and imaging.

**Xenograft.** Hep3B cells were transfected in six-well plates. Forty-eight hours post-transfection, cells were collected, washed and counted. Two million cells per injection were prepared by mixing the cell suspension with Matrigel Matrix (Corning) in a 1:1 ratio. The cell mixture was injected subcutaneously into the lower flank on each side of five (per condition) 4- to 6-week-old, female, CrTac:NCr-Foxn1<nu> (NCr nude) mice (Invivos). Tumour sizes were measured every 3 days. The mice were killed after 35 days and the tumours were excised, weighed and measured. All mouse work was performed in accordance with the NUS IACUC guidelines under the protocol number R19-0852. The maximum tumour volume permitted is 2,000 mm³, which was not exceeded in all our xenograft experiments. The mice were housed in the following conditions: 23–24 °C, 44–58% humidity, 12 h/12 h dark/light cycle (19:00–7:00/7:00–19:00).

**Luciferase and translation reporter assays.** Cells were seeded at 50,000 cells per well in 24-well plates a day before transfection. Then, 25 ng of psiCHECK-2 plasmids were transfected per well using Lipofectamine 3000 as described above. The transfected cells were washed in PBS and lysed and luminescence was measured 72 h post-transfection following the manufacturer's protocol for the dual luciferase reporter assay kit (Promega).

For the translation reporter assay, PCR was performed using forward primers with T7 promoter and Kozak sequences, reverse primers with polyT (Supplementary Table 10) and psiCHECK-2 plasmids as templates. In vitro transcription was performed using 1 μg of purified PCR product, 1× transcription buffer, 4 mM NTP mix, 8 mM 3′-O-Me-m7G(5′)ppp(5′)G RNA Cap Structure Analog (New England Biolabs) and 200 U T7 RNA polymerase (Ambion), and incubated for 4 h at 37 °C. The transcription products were purified by ethanol precipitation followed by the Microspin G-50 columns (GE Healthcare). Cells were seeded as described above. Then, 20 ng of the Firefly luciferase control was co-transfected using Lipofectamine 3000 with the corresponding amount of Renilla luciferase control or 3′ UTR reporter (calculated on the basis of $5 \times 10^{11}$ copies per reporter). Luciferase activity was measured 48–72 h post-transfection as described above.

**RNA extraction and RT–qPCR.** Total RNA was extracted using TRIzol followed by column purification using the PureLink RNA Mini Kit (Thermo Fisher) with on-column DNase I treatment (Thermo Fisher). The High Capacity cDNA Reverse Transcription Kit (Thermo Fisher) was used to generate complementary DNA. Subsequently, real-time quantitative PCR (RT–qPCR) was performed using the PowerUp SYBR Green Master Mix (Applied Biosystems) on the QuantStudio 5 RT–PCR system (Applied Biosystems). PCR experiments were performed using EconoTaq PLUS GREEN 2× Master Mix (Lucigen) following the manufacturer's instructions. qPCR and PCR primers are listed in Supplementary Table 10.

**Protein extraction and western blot analysis.** Cells were collected and lysed as previously described[44]. For western blot analysis, 10–15 μg of lysates were

fractionated using 8% SDS–PAGE gels in running buffer (25 mM Tris, 192 mM glycine and 0.1% SDS) and transferred to PVDF membranes (Thermo Fisher) in transfer buffer (25 mM Tris, 192 mM glycine and 20% (v/v) methanol). The membranes were probed with specific primary and secondary antibodies in 5% BSA–TBST.

**RNA fluorescence in situ hybridization.** RNA–FISH was performed using custom BaseScope probes, RNAscope Pretreatment Reagents, Wash Buffer Reagents and BaseScope Detection Reagents v2-Red (ACD, Supplementary Table 11) following the manufacturer's protocol. Briefly, 200,000 Hep3B and 300,000 SNU398 cells were seeded on glass coverslips in six-well plates. The cells were grown to 50–70% confluency, fixed in 10% formalin for 30 min and subjected to a series of ethanol dehydration and rehydration steps, hydrogen peroxide and RNAscope Protease III treatment. They were subsequently hybridized with the custom ACD probes, followed by a series of signal amplification steps, signal detection using the BaseScope Fast RED dye and counterstaining with DAPI (Thermo Fisher). The glass coverslips were mounted using ProLong Gold Antifade Mountant (Thermo Fisher). Fluorescence images were acquired at 60× magnification using the Olympus FV1200 confocal microscope and Fluoview (v3.0) and processed using ImageJ (v1.51j8).

**RNA immunoprecipitation.** The protocol was adapted from the RIP–ChIP protocol described previously[45]. Briefly, protein A Sepharose beads (Sigma) were coated with 3 μg of U1 snRNP, SF3B1, SRSF1, U2AF2 or mouse IgG antibody (Santa Cruz), followed by incubation with 2 mg of Hep3B or SNU398 total cell lysates overnight. The RNA–protein–bead complexes were washed once with NT2 crowders (25 mg Ficoll PM400 (GE Healthcare), 75 mg Ficoll PM70 (GE Healthcare) and 2.5 mg dextran sulfate (Fluka) in 10 ml of NT2 buffer) and five times with NT2 buffer (50 mM Tris pH 7.0, 150 mM NaCl, 1 mM $MgCl_2$ and 0.05% (v/v) NP-40). Protein–RNA complexes were collected in 100 μl of NET2 buffer (1 mM DTT, 16.7 mM EDTA, 200 U RNaseOUT (Thermo Fisher) and 100 U SUPERase In (Ambion) in 1× NT2 crowder), supplemented with 100 μl of 2× SDS–TE (100 mM Tris pH 7.5, 10 mM EDTA pH 8.0 and 1% SDS). RNA was isolated using TRIzol reagent and subsequently purified with phenol:chloroform:isoamyl alcohol (25:24:1) and chloroform:isoamyl alcohol (24:1).

**RNA pulldown.** RNA pulldown using biotinylated probes was performed as described previously with slight modifications[46]. In brief, cell lysates were prepared in lysis buffer (25 mM Tris pH 7.5, 150 mM NaCl, 1 mM EDTA, 1% (v/v) NP-40, 5% (v/v) glycerol and 100 U ml⁻¹ SUPERase In protease inhibitor). Then, 50 μl of Dynabeads MyOne Streptavidin C1 (per sample) were pre-washed and blocked in 500 μl of lysis buffer supplemented with 0.2 μg μl⁻¹ yeast tRNA (Thermo Fisher) for 2 h at 4 °C. Meanwhile, 4 μg of biotinylated probe (Supplementary Table 8) was incubated with 1 mg of lysate per reaction for 1 h at room temperature with rotation, after which blocked beads were washed and added to the lysate–probe mix and incubated for another 2 h at room temperature with rotation. The RNA–protein–bead complexes were washed six times with lysis buffer. Proteins were eluted in 1× reducing sample buffer and loaded on 8% SDS–PAGE gels for western blot analysis.

**Nuclear–cytoplasmic fractionation.** Nuclear–cytoplasmic fractionation was performed following the Abcam protocol with some modifications. Briefly, cell pellets were lysed in Buffer A (10 mM HEPES, 1.5 mM $MgCl_2$, 10 mM KCl, 0.5 mM DTT and 0.05% (v/v) NP-40, pH 7.5). Following a 10 min incubation on ice and centrifugation, supernatants were collected as the cytoplasmic fractions. The remaining pellets were resuspended in Buffer B (5 mM HEPES, 1.5 mM $MgCl_2$, 0.2 mM EDTA, 0.5 mM DTT and 26% (v/v) glycerol, pH 7.5) and 750 mM NaCl. The suspension was homogenized on ice using a handheld homogenizer. After a 30 min incubation on ice and centrifugation, the resulting supernatants were collected as the nuclear fractions. The fractions were divided and processed to collect RNA and proteins as described above.

**Polysome extraction and fractionation.** Polysome extraction, fractionation and RNA extraction were performed as previously described with some modifications[47]. Gradient centrifugation was performed for 1.5 h, followed by polysome fraction collection and RNA extraction. cDNA was prepared using the High Capacity cDNA Reverse Transcription Kit (Thermo Fisher) and the same volume of RNA across all fractions (calculated from 1 μg of RNA based on the highest concentration among all fractions). RT–qPCR was performed as described above.

**Statistics and reproducibility.** All statistical analyses for experimental data were performed using Excel from Microsoft 365 v16 and statistical significance was considered at test level $P < 0.05$. $P$ values were calculated using unpaired two-tailed Student's $t$-test unless otherwise stated. All experiments were performed independently at least twice by two or more investigators with reproducible results. No statistical method was used to pre-determine sample size. Eighty RNA-seq samples from GTEx were excluded because of low sequencing depth (total number of splicing junctions <1,000). Sample numbers before and after exclusion are presented in Supplementary Table 1. No data were excluded from the analyses

of biological experiments. Randomization was applied to all in vivo experiments but not in vitro experiments as it was not necessary. Blinding was not applied to computational analyses as these were performed using unbiased software programs or algorithms. Blinding was applied to the data collection of at least one set of each experiment except for RNA–FISH owing to the experimental technicality and licence requirement for confocal microscopy.

**Reporting Summary.** Further information on research design is available in the Nature Research Reporting Summary linked to this article.

## Data Availability

The PLANet RNA-seq dataset of 211 samples was generated in the PLANet study and deposited in the European Genome-phenome Archive (EGA, http://www.ebi.ac.uk/ega/) under the accession code EGAS00001003813 (ref. [14]). All c3USPs across ten TCGA cancer types and their corresponding normal tissues, as well as the patterns of SPL, can be found on the SpUR database: http://www.cbrc.kaust.edu.sa/spur/home. It also provides a function to query the association between 3′ UTR SPLs and prognosis in each cancer type. RNA-seq data of the in-house four HCC matched pairs and 55 AML and healthy control samples analysed in this study have been deposited in Sequence Read Archive (SRA, https://www.ncbi.nlm.nih.gov/sra) with the accession number PRJNA602213. RNA-seq data from PacBio used in this study are released on its official website and can be downloaded from the links below: http://datasets.pacb.com.s3.amazonaws.com/2014/Iso-seq_Human_Tissues/list.html; http://datasets.pacb.com.s3.amazonaws.com/2013/IsoSeqHumanMCF7Transcriptome/list.html The genome and gene annotation used for the alignment were downloaded from GENCODE (https://www.gencodegenes.org). Source data are provided with this paper. All other data supporting the findings of this study are available from the corresponding author on reasonable request.

## Code availability

The custom code to extract, filter and analyse 3USPs is publicly available at https://github.com/christear/RNASeq3USP, which requires the use of BEDTools (v2.29), featureCounts (v1.6.1), SAMtools (v1.8), Perl (v5.26) and R (v4.1.2).

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

## Acknowledgements

We thank all past and present Y.T. lab members for their constructive feedback on this project, V. Teh for providing the mouse RNA samples, the NUHS Leukemia Cell Bank for the AML samples and S. J. Tang for help with ASO design. The computational analysis in this study is supported by National Supercomputing Centre Singapore (NSCC). Y.T. is funded by NMRC OF-IRGs (NMRC/OFIRG/MOH-000380, MOH-000923), the National Research Foundation Singapore and the Singapore Ministry of Education under its Research Centres of Excellence initiative, and the RNA Biology Center at the Cancer Science Institute of Singapore, NUS, as part of funding under the Singapore Ministry of Education's AcRF Tier 3 grants (MOE2014-T3-1-006). Singapore National Medical Research Council grants (TCR/015-NCC/2016 and NMRC/CSA-SI/0018/2017): P.K.H.C. King Abdullah University of Science and Technology (KAUST) Office of Sponsored Research (OSR) (BAS/1/1624-01, FCC/1/1976-23-01, FCC/1/1976-26-01, REI/1/0018-01-01, REI/1/4216-01-01, REI/1/4437-01-01, REI/1/4473-01-01 and URF/1/4098-01-01): X.G.

## Author contributions

J.J.C. performed experiments and analysed data. B.Z. performed all computational analyses. X.H.C., Z.H.K., C.Y.L., N.D., A. Siemens, T.K. and S.O. performed experiments. O.A. and H.Y. provided the TCGA data. A. Salhi and X.G. built the SpUR database. N.S. performed polysome fractionation experiments. A.S.-M., P.T.L., R.S., X.F., J.J.P., X.R., E.G., L.A.V., L.C. and H.Y. provided input for the project. S.W., B.E.S., K.C.L., C.S.C., B.L., W.-K.C. and K.-K.T. provided COAD clinical samples; Y.G., W.N.F., M.G.O., B.T.H.K., M.J.F., W.W. and W.J.C. provided the clinical AML samples and RNA-seq dataset; S.G.I. and Y.Y.D. provided HCC clinical samples for the in-house RNA-seq dataset; W.H.L., J.C., B.-K.Y., R.C., G.K.B., B.K.P.G., W.Z. and P.K.H.C. provided the HCC clinical samples and RNA-seq dataset from the PLANet study. J.J.C., B.Z. and Y.T. designed the study and prepared the manuscript. All authors reviewed and commented on the manuscript.

## Competing interests

The authors declare no competing interests.

## Additional information

**Extended data** is available for this paper at https://doi.org/10.1038/s41556-022-00913-z.

**Correspondence and requests for materials** should be addressed to Yvonne Tay.

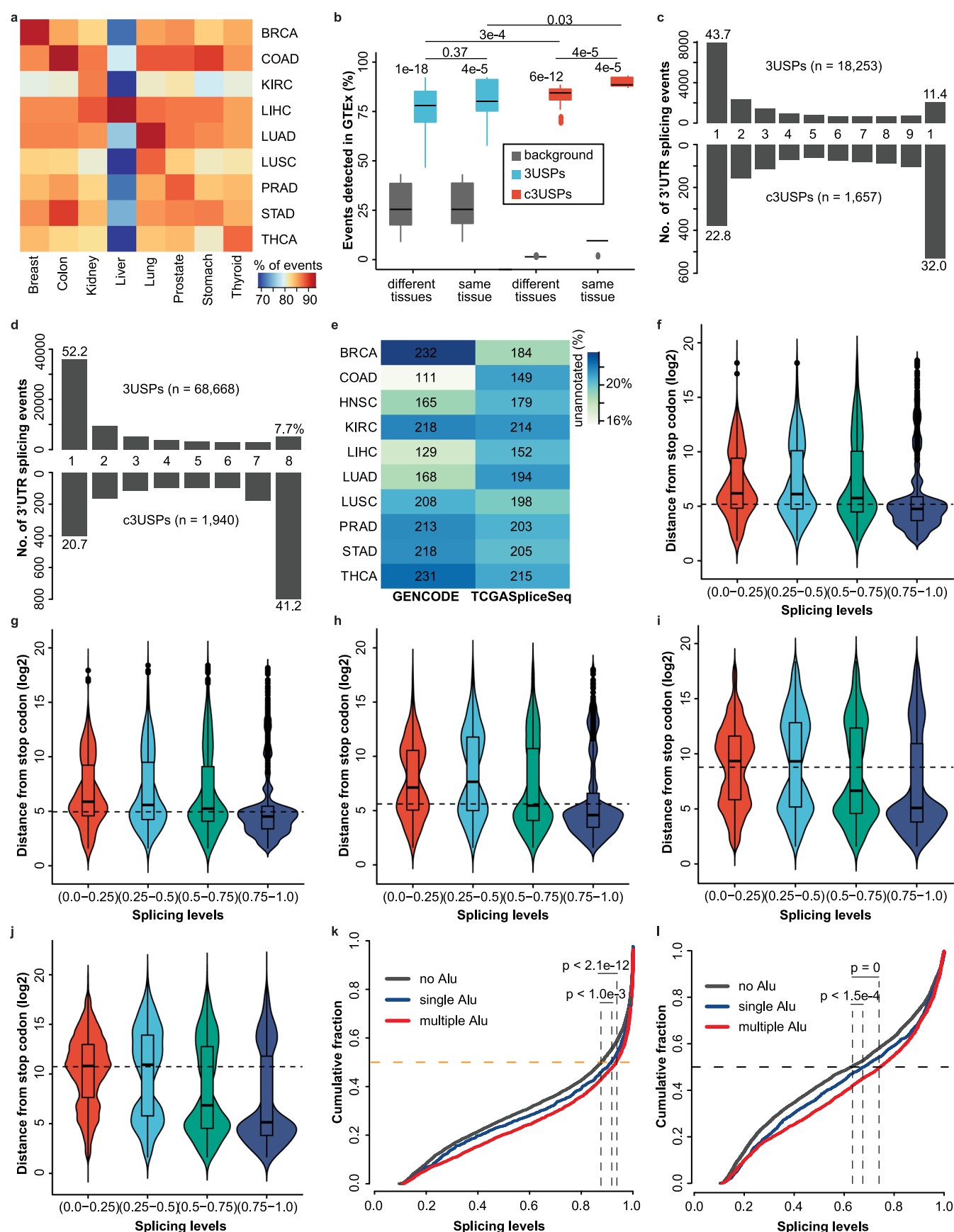

**Extended Data Fig. 1 | See next page for caption.**

**Extended Data Fig. 1 | Global analysis of 3'UTR splicing events with TCGA and GTEx datasets. a**, Heatmap illustrating the percentage overlap of c3USPs identified from TCGA-normal and GTEx samples. The rows and columns indicate the TCGA-normal and GTEx tissues, respectively. **b**, Boxplot showing the distribution of the percentage of 3'UTR splicing events identified in TCGA-normal that overlapped with those from GTEx, including 3USPs, c3USPs and their corresponding backgrounds. **c-d**, Bar plots showing the number of 3USPs and c3USPs detected in the TCGA-normal samples of different numbers of cancer types (**c**) and different tissues from GTEx (**d**). The x-axis indicate the number of cancer types (Supplementary Table 1). **e**, Proportion of novel c3USPs that are not annotated or reported in previous studies. **f-j**, Distribution of the distance from the stop codon for multiple groups of c3USPs (**f-h**) and 3USPs (**i,j**) classified by splicing levels in TCGA-tumor (**f**), TCGA-normal (**g,i**) and GTEx (**h,j**). **k-l**, Cumulative distribution of splicing levels of c3USPs in TCGA-normal (**k**) and GTEx (**l**) with or without Alu elements. P-values: Wilcoxon test.

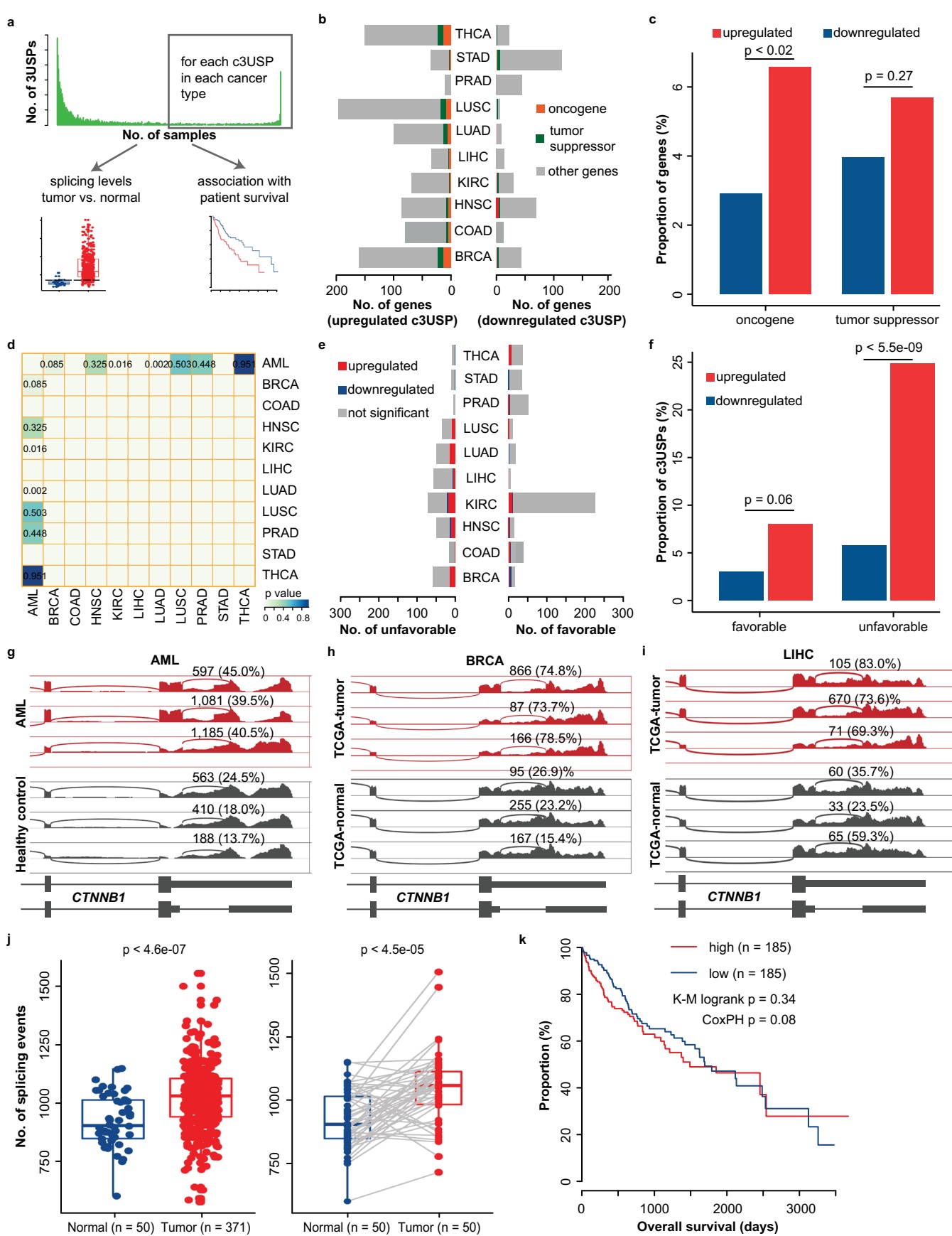

**Extended Data Fig. 2 | See next page for caption.**

**Extended Data Fig. 2 | 3'UTR splicing is upregulated in cancer and correlated with poor prognosis. a**, Workflow of the analysis for each c3USP in each cancer type. **b**, Bar plot showing the number of genes with up and downregulated c3USPs. **c**, Bar plot illustrating the proportion of oncogenes and tumor suppressors in two groups of genes with either upregulated or downregulated c3USPs. P-values: hypergeometric test. **d**, Heatmap illustrating p-values derived from overlapping c3USPs between each two cancer types (hypergeometric test). P-values >0.001 are labeled. **e**, Bar plot showing the number of favorable and unfavorable prognostic c3USPs across different cancer types derived from the Kaplan-Meier method with a p < 0.05 cutoff. **f**, Bar plot illustrating the proportion of favorable and unfavorable markers in upregulated and downregulated c3USPs. **g-i**, Sashimi plots illustrating 3'UTR splicing of the top dysregulated splicing event of *CTNNB1* in the normal and tumor samples from AML (**g**), TCGA-BRCA (**h**) and -LIHC (**i**). **j**, Number of 3'UTR splicing events in all (left), and 50 matched tumor-normal samples (right) from the TCGA-LIHC dataset. n: number of RNA-seq samples analyzed; P-values: Mann-Whitney U test. **k**, Kaplan-Meier survival analysis of TCGA-LIHC samples based on the segmentation numbers of total splicing events (top and bottom half of samples ranked by numbers). n: number of patients analyzed.

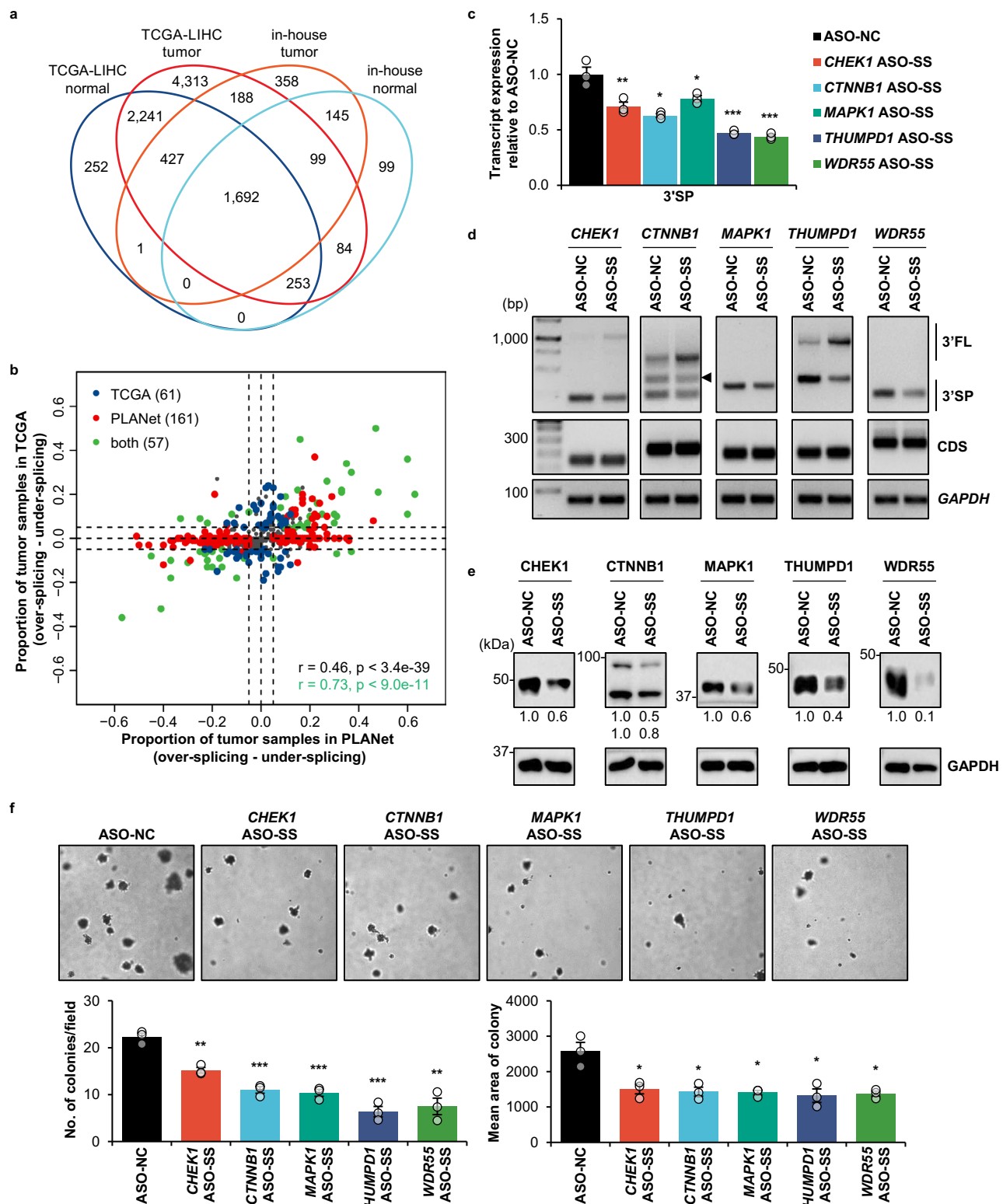

**Extended Data Fig. 3 | Targeted inhibition of 3'UTR splicing impedes HCC carcinogenesis. a**, Overlap of 3USPs identified from in-house and TCGA-LIHC data. **b**, Scatterplot illustrating the dysregulation of c3USPs from the PLANet and TCGA-LIHC datasets. **c-f**, Effect of ASO-mediated blocking of the 3'UTR splice site on candidate 3'SP transcript expression by qPCR (n = 3 independent experiments) (**c**), candidate 3'FL, 3'SP and CDS transcript expression by PCR (◄ 3'SP2, see Supplementary Note) (**d**), protein expression (**e**) and anchorage-independent growth (n = 3 independent experiments) (**f**) in HepG2. CDS: coding sequence; 3'FL: full length 3'UTR; 3'SP: spliced 3'UTR; ASO-NC: non-targeting control ASO; ASO-SS: splice site ASO. **c,f**, Mean ± SEM; unpaired Student's t-test *p < 0.05, **p < 0.01, ***p < 0.001. **d,e**, Data shown represent three independent experiments.

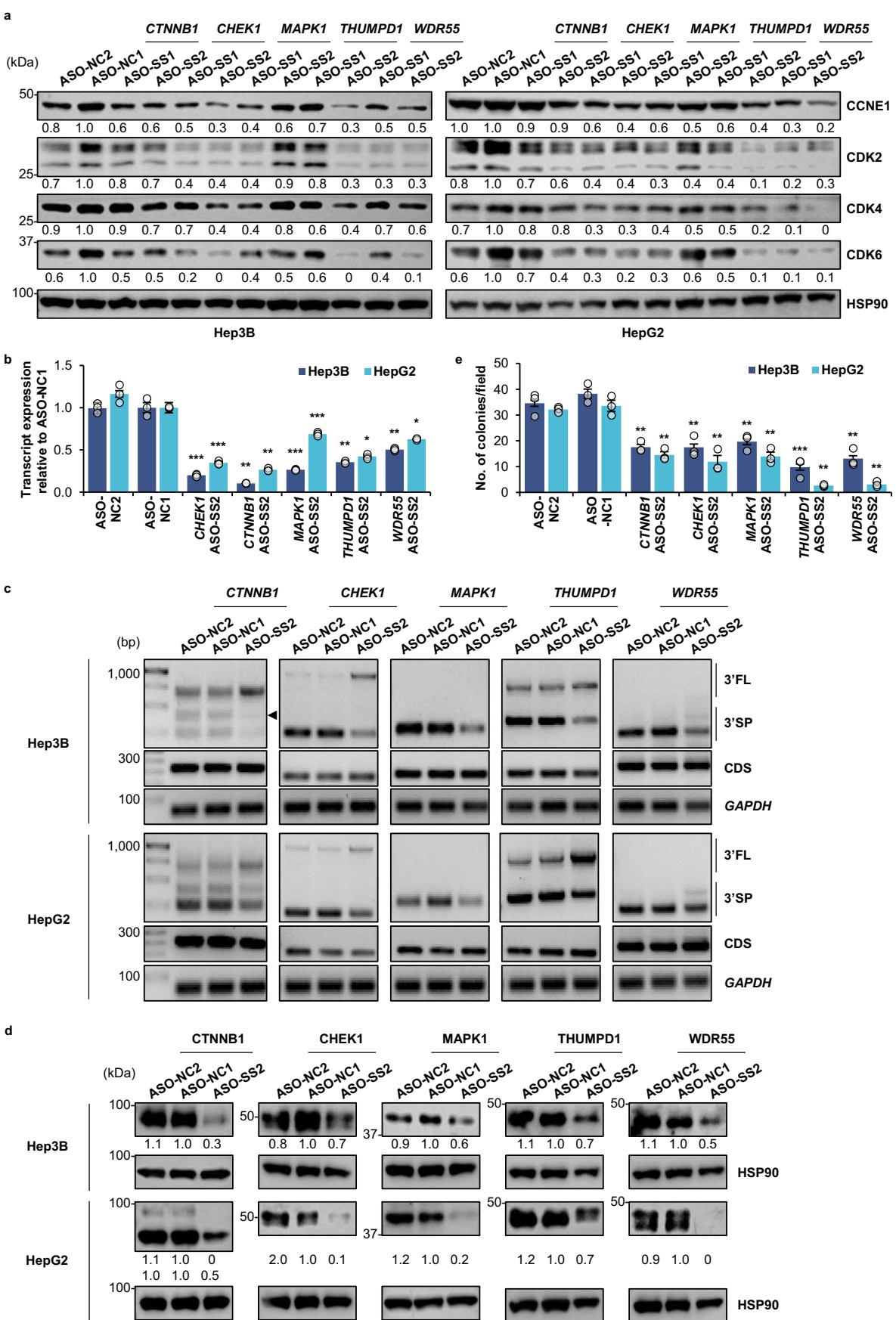

**Extended Data Fig. 4 | See next page for caption.**

**Extended Data Fig. 4 | Targeted inhibition of 3'UTR splicing impedes HCC cell proliferation. a-e**, Effect of ASO-mediated blocking of the 3'UTR splice site on the protein expression of cell cycle markers (**a**), candidate transcript expression by qPCR (n = 3 independent experiments) (**b**), and PCR (◄ 3'SP2, see Supplementary Note) (**c**), candidate protein expression (**d**) and anchorage-independent growth (n = 3 independent experiments) (**e**) in Hep3B and HepG2. CDS: coding sequence; 3'FL: full length 3'UTR; 3'SP: spliced 3'UTR; ASO-NC: non-targeting control ASO; ASO-SS: splice site ASO. **b,e**, Mean ± SEM; unpaired Student's t-test *p < 0.05, **p < 0.01, ***p < 0.001. **a,c,d**, Data shown represent three independent experiments.

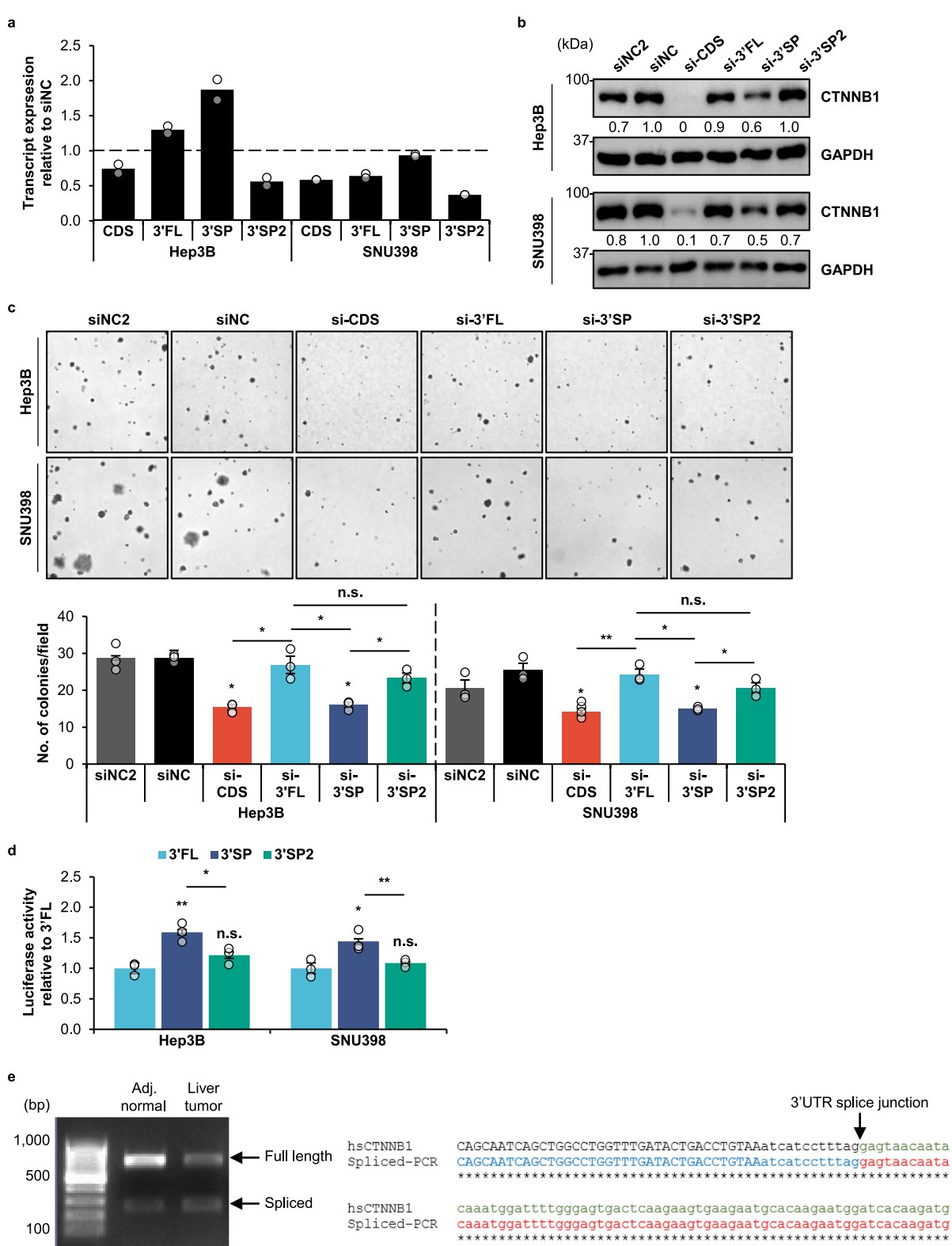

**Extended Data Fig. 5 | *CTNNB1* 3'SP2 does not affect CTNNB1 expression and HCC tumorigenesis. a-c**, Effect of siRNA-mediated knockdown of *CTNNB1* 3'SP2 on *CTNNB1* transcript (n = 2 independent experiments) (**a**) and protein (**b**) expression, and anchorage-independent growth (n = 3 independent experiments) (**c**) in Hep3B and SNU398. **d**, Luciferase activity of reporter constructs with *CTNNB1* 3'FL, 3'SP and 3'SP2 (n = 3 independent experiments). **e**, PCR analysis of 3'FL and 3'SP expression in mouse adjacent normal and liver tumor samples (left). Alignment of the 3'UTR splice junctions and flanking regions of human *CTNNB1* (hsCTNNB1 NCBI RefSeq) and the mouse *CTNNB1* 3'UTR splice variant (spliced-PCR, detected via Sanger sequencing of the PCR product) (right). CDS and 3'UTR in upper- and lowercase, human 5' and 3' exons in black and green, and mouse in blue and red, respectively. siNC: siRNA non-targeting control; CDS: coding sequence; 3'FL: full length; 3'SP: spliced 3'UTR. **c,d**, Mean ± SEM; unpaired Student's t-test *p < 0.05, **p < 0.01, ***p < 0.001. **b,e**, Data shown represent three independent experiments.

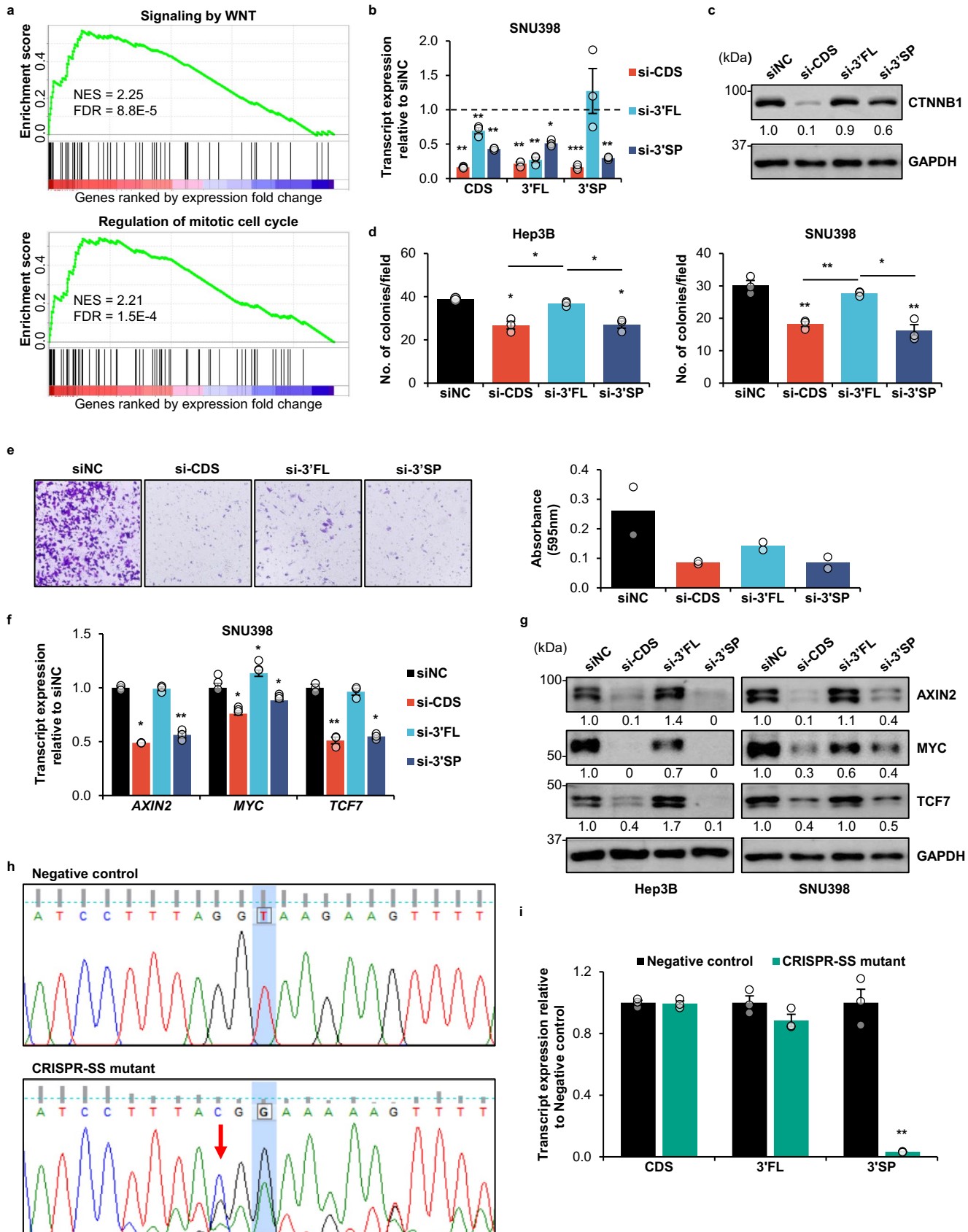

**Extended Data Fig. 6 | See next page for caption.**

**Extended Data Fig. 6 | 3'UTR splicing of *CTNNB1* promotes HCC tumorigenesis. a**, Gene set enrichment analysis (GSEA) showing the enrichment of upregulated genes from the WNT signaling and mitotic cell cycle gene sets in the PLANet tumor samples with *CTNNB1* 3'UTR over-splicing. **b-g**, Effect of siRNA-mediated knockdown of *CTNNB1* CDS, 3'FL and 3'SP on *CTNNB1* transcript (n = 3 independent experiments) (**b**) and protein expression (**c**) in SNU398, anchorage-independent growth in Hep3B and SNU398 (n = 3 independent experiments) (**d**), cell migration (n = 2 independent experiments) (**e**), WNT target transcript (n = 3 independent experiments) (**f**) and protein expression (**g**) in SNU398. **h**, Chromatograms depicting Sanger sequencing validation of the negative control and CRISPR-Cas9-mediated T > G mutation (highlighted) of the *CTNNB1* 3'UTR splice site (CRISPR-SS mutant) at the genomic level. The red arrow indicates the G > C mutation introduced to the PAM sequence to prevent further Cas9 cleavage. **i**, Effect of the CRISPR-SS mutation on *CTNNB1* transcript expression (n = 3 independent experiments). siNC: siRNA non-targeting control; CDS: coding sequence; 3'FL: full length; 3'SP: spliced 3'UTR. **b,d,f,i**, Mean ± SEM; unpaired Student's t-test *p < 0.05, **p < 0.01, ***p < 0.001. **c,g**, Data shown represent three independent experiments.

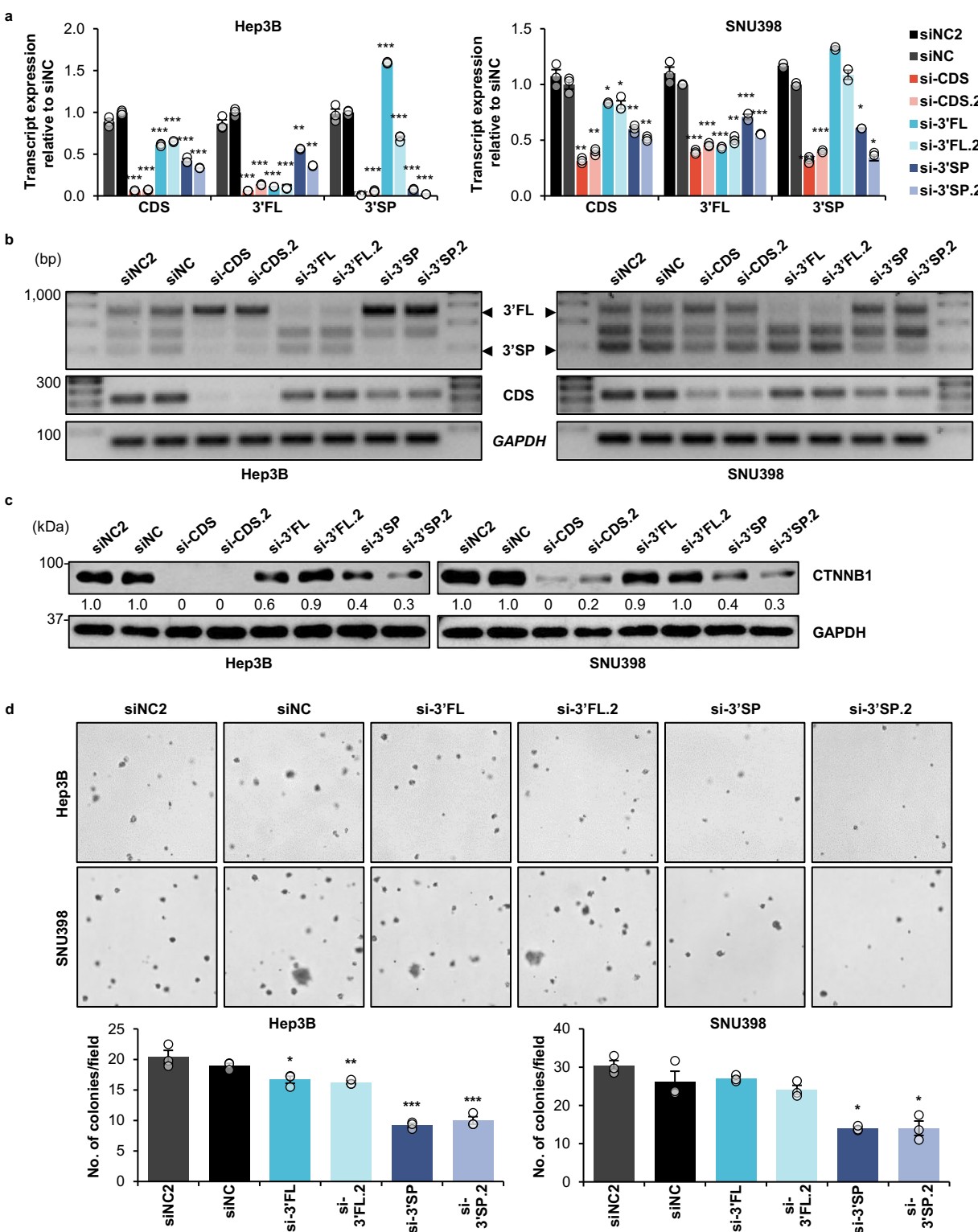

**Extended Data Fig. 7 | 3'UTR splicing of *CTNNB1* promotes HCC tumorigenesis. a-d**, Effect of siRNA-mediated knockdown of *CTNNB1* CDS, 3'FL and 3'SP on CTNNB1 transcript expression by qPCR (n = 3 independent experiments) (**a**) and PCR (**b**), protein expression (**c**) and anchorage-independent growth (n = 3 independent experiments) (**d**). siNC: siRNA non-targeting control; CDS: coding sequence; 3'FL: full length 3'UTR; 3'SP: spliced 3'UTR. **a,d**, Mean ± SEM; unpaired Student's t-test *p < 0.05, **p < 0.01, ***p < 0.001. **b,c**, Data shown represent three independent experiments.

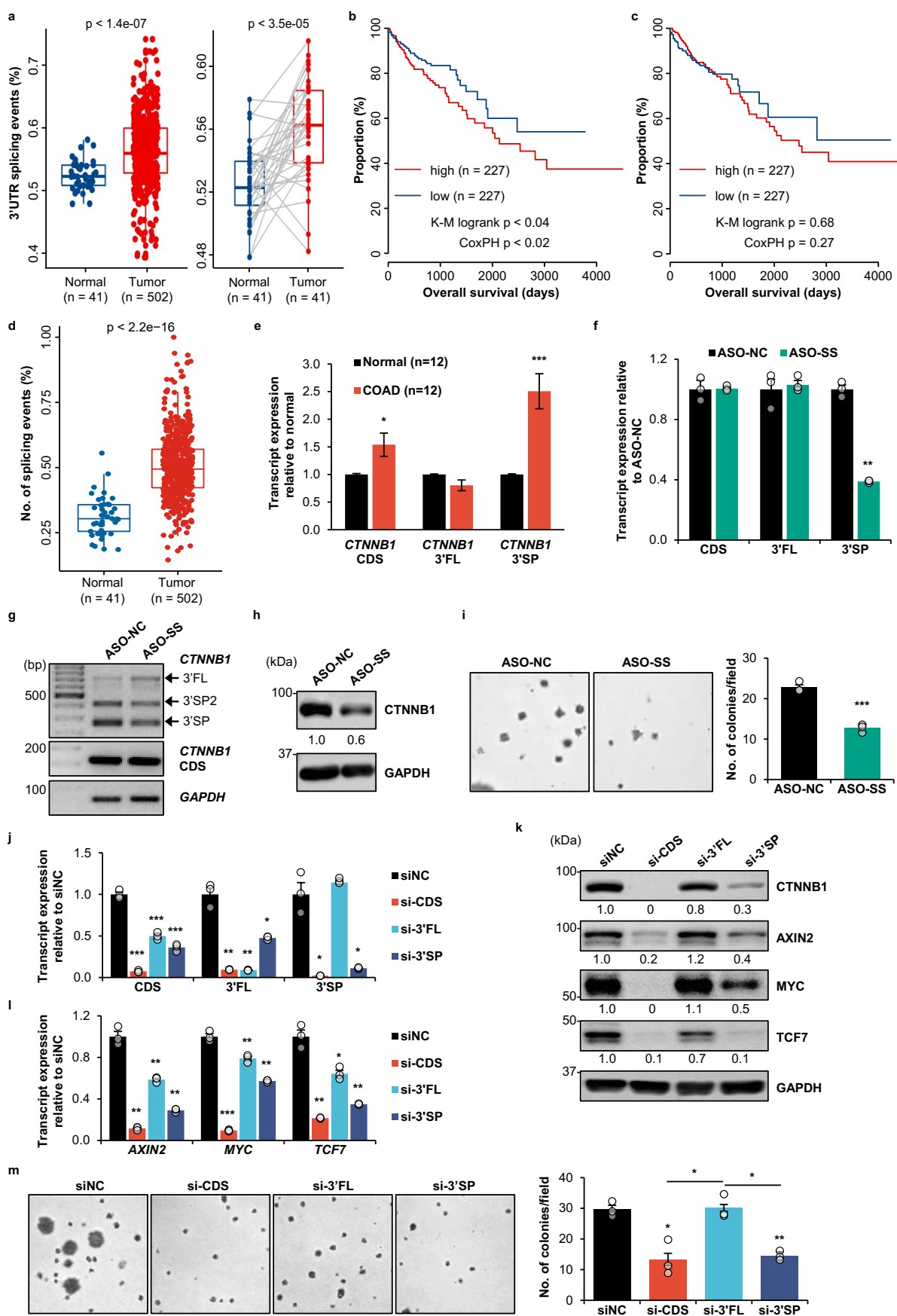

**Extended Data Fig. 8 | See next page for caption.**

**Extended Data Fig. 8 | 3′UTR splicing of *CTNNB1* promotes COAD tumorigenesis. a**, Proportion of 3′UTR splicing events in all (left), and 41 matched tumor-normal samples (right) from the TCGA-COAD dataset. P-values: Mann-Whitney U test; n: number of RNA-seq samples analyzed. **b,c**, Kaplan-Meier survival analysis of TCGA-COAD samples based on the segmentation numbers of 3USPs (**b**) and total splicing events (**c**) (top and bottom half of samples ranked by numbers). n: number of patients analyzed. **d**, Proportion of *CTNNB1* 3′UTR splicing events in all normal and tumor samples from the TCGA-COAD dataset. n: number of RNA-seq samples analyzed. **e**, Comparison of the *CTNNB1* 3′SP transcript expression between paired normal and COAD clinical samples (n=12 patient samples). **f-i**, Effect of ASO-mediated blocking of the 3′UTR splice site on *CTNNB1* transcript expression by qPCR (n=3 independent experiments) (**f**), and PCR (**g**), protein expression (**h**) and anchorage-independent growth (n=3 independent experiments) (**i**) in DLD-1. **j-m**, Effect of siRNA-mediated knockdown of *CTNNB1* on *CTNNB1* transcript (n=3 independent experiments) (**j**), CTNNB1 and WNT target proteins (**k**) and WNT target transcript (n=3 independent experiments) (**l**) expression, and anchorage-independent growth (n=3 independent experiments) (**m**) in DLD-1. ASO-NC: non-targeting control ASO; ASO-SS: splice site ASO; CDS: coding sequence; 3′FL: full length 3′UTR; 3′SP spliced 3′UTR. **e,f,i,j,l,m**, Mean ± SEM; unpaired Student's t-test *p < 0.05, **p < 0.01, ***p < 0.001. **g,h,k**, Data shown represent three independent experiments.

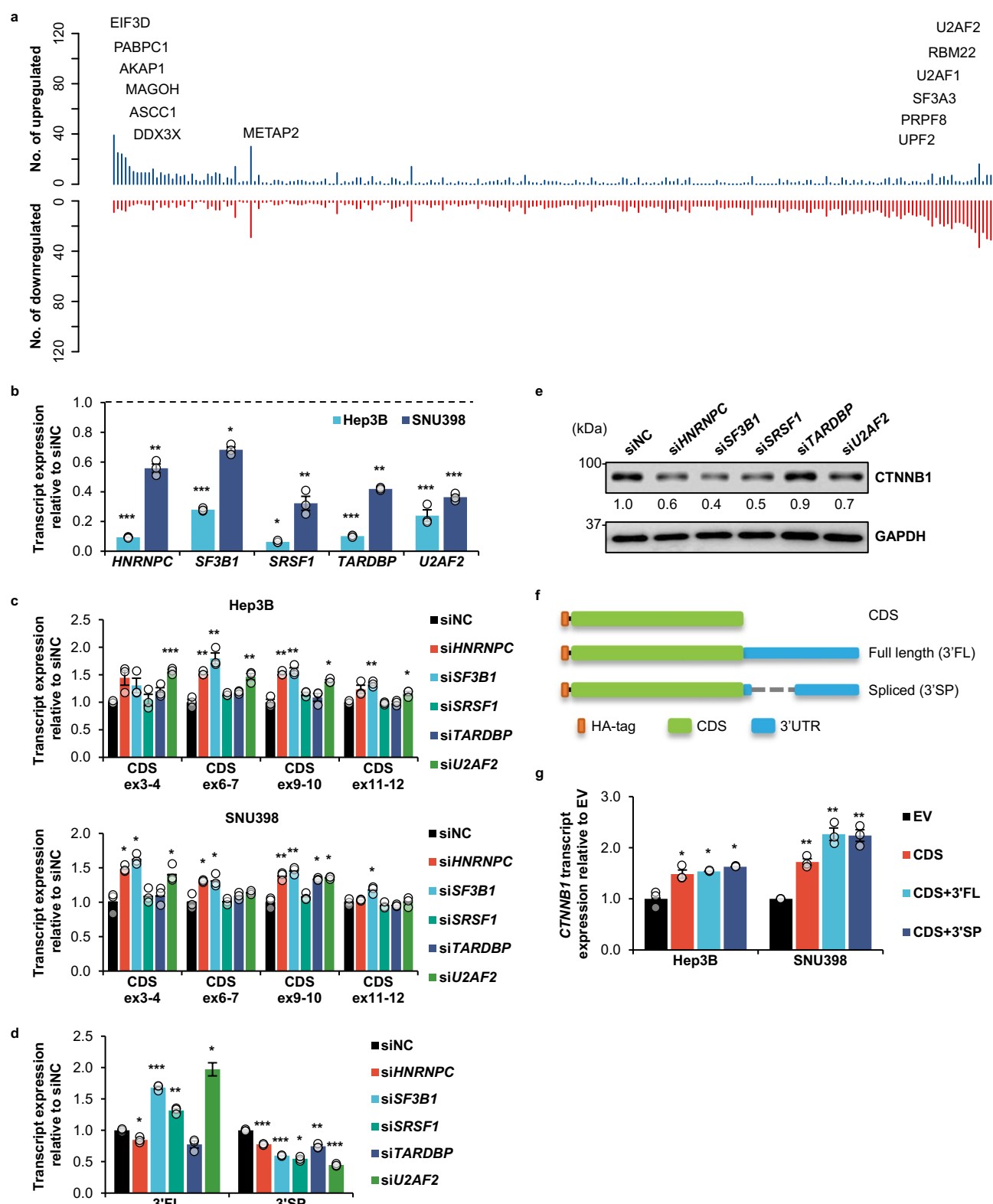

**Extended Data Fig. 9 | 3'UTR splicing may enhance CTNNB1 protein expression in HCC. a**, Number of c3USPs that are significantly up/downregulated upon the knockdown of RBPs compared to controls. **b-e**, Effect of the knockdown of splicing factors on the individual RBP transcript expression (n = 3 independent experiments) (**b**) and CDS exon-exon junctions (n = 3 independent experiments) (**c**) in Hep3B and SNU398, *CTNNB1* transcript (n = 3 independent experiments) (**d**) and CTNNB1 protein (data shown represent three independent experiments) (**e**) expression in SNU398. **f**, Schematic representation of the *CTNNB1* CDS, CDS + 3'FL and CDS + 3'SP overexpression constructs. **g**, Effect of overexpressing *CTNNB1* CDS, CDS + 3'FL and CDS + 3'SP on *CTNNB1* transcript expression in Hep3B and SNU398 (n = 3 independent experiments). siNC: siRNA non-targeting control; CDS: coding sequence; 3'FL: full length 3'UTR; 3'SP: spliced 3'UTR. **b-d,g**, Mean ± SEM; unpaired Student's t-test *p < 0.05, **p < 0.01, ***p < 0.001.

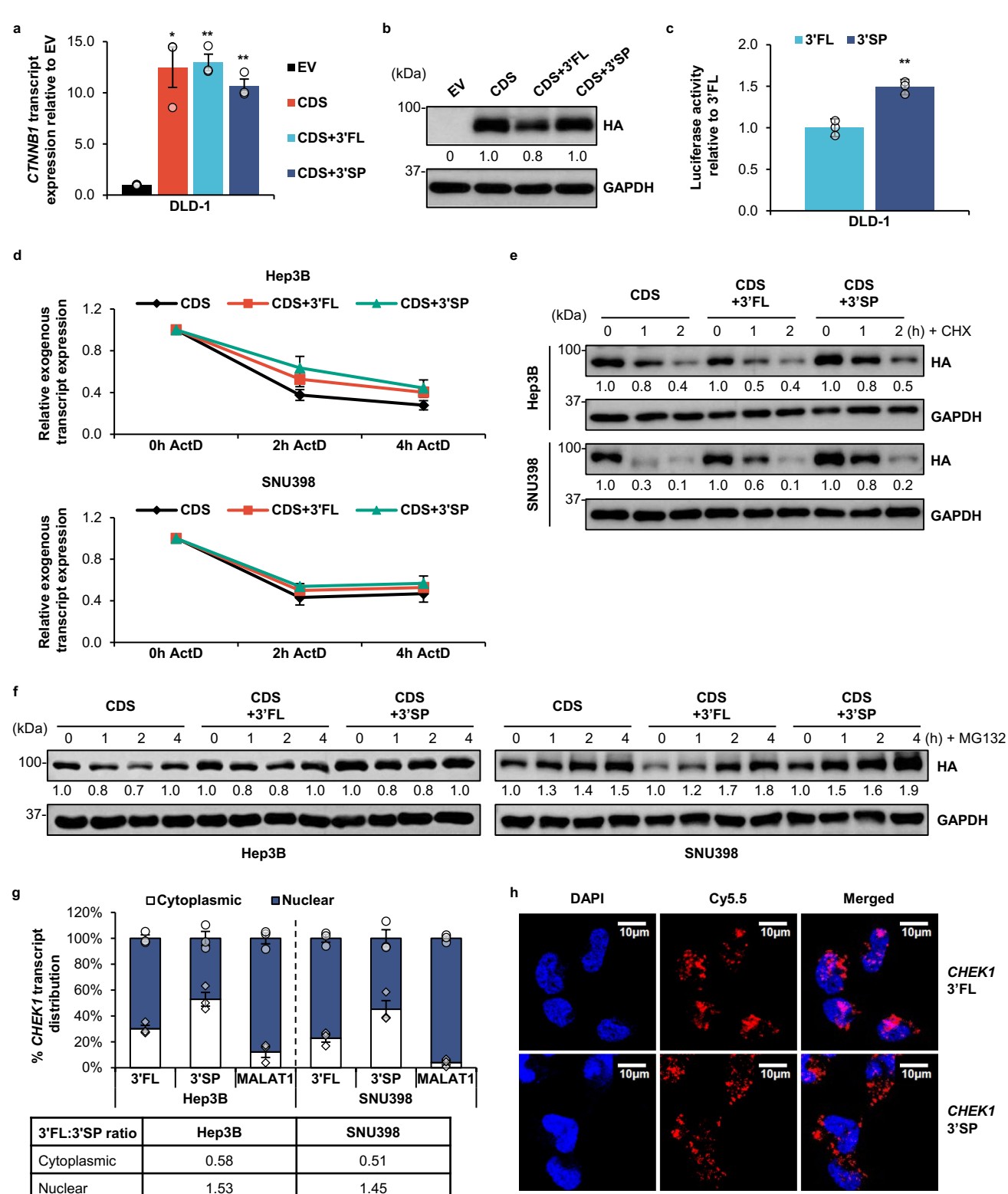

**Extended Data Fig. 10 | 3′UTR splicing-mediated cytoplasmic localization enhances CTNNB1 expression. a,b,** Effect of overexpressing *CTNNB1* CDS, CDS + 3′FL and CDS + 3′SP on endogenous *CTNNB1* transcript (n = 3 independent experiments) (**a**) and exogenous CTNNB1 protein (**b**) expression in DLD-1. **c**, Luciferase activity of reporter constructs with *CTNNB1* 3′FL and 3′SP in DLD-1 (n = 3 independent experiments). **d**, Effect of actinomycin D (ActD) treatment on the transcript levels of exogenously expressed *CTNNB1* CDS, CDS + 3′FL and CDS + 3′SP in Hep3B and SNU398 (n = 3 independent experiments). **e,f,** Effect of cycloheximide (CHX) (**e**) or MG132 (**f**) treatment on exogenously expressed CTNNB1 protein levels of in Hep3B and SNU398. **g**, Subcellular distribution of *CHEK1* 3′FL and 3′SP transcripts following nuclear-cytoplasmic fractionation of Hep3B and SNU398 cells (n = 3 independent experiments). *MALAT1* was used as a nuclear control. The 3′FL:3′SP transcript ratios in each cellular compartment are shown in the table below. **h**, RNA-FISH showing transcript localization of *CHEK1* 3′FL and 3′SP in SNU398. EV: empty vector; CDS: coding sequence; 3′FL: full length 3′UTR; 3′SP spliced 3′UTR. **a,c,d,g,** Mean ± SEM; unpaired Student's t-test *p < 0.05, **p < 0.01, ***p < 0.001. **b,e,f,h,** Data shown represent three independent experiments.

# Reporting Summary

## Statistics

For all statistical analyses, confirm that the following items are present in the figure legend, table legend, main text, or Methods section.

| n/a | Confirmed | |
|---|---|---|
| ☐ | ☒ | The exact sample size (*n*) for each experimental group/condition, given as a discrete number and unit of measurement |
| ☐ | ☒ | A statement on whether measurements were taken from distinct samples or whether the same sample was measured repeatedly |
| ☐ | ☒ | The statistical test(s) used AND whether they are one- or two-sided *Only common tests should be described solely by name; describe more complex techniques in the Methods section.* |
| ☐ | ☒ | A description of all covariates tested |
| ☐ | ☒ | A description of any assumptions or corrections, such as tests of normality and adjustment for multiple comparisons |
| ☐ | ☒ | A full description of the statistical parameters including central tendency (e.g. means) or other basic estimates (e.g. regression coefficient) AND variation (e.g. standard deviation) or associated estimates of uncertainty (e.g. confidence intervals) |
| ☐ | ☒ | For null hypothesis testing, the test statistic (e.g. *F*, *t*, *r*) with confidence intervals, effect sizes, degrees of freedom and *P* value noted *Give P values as exact values whenever suitable.* |
| ☒ | ☐ | For Bayesian analysis, information on the choice of priors and Markov chain Monte Carlo settings |
| ☒ | ☐ | For hierarchical and complex designs, identification of the appropriate level for tests and full reporting of outcomes |
| ☒ | ☐ | Estimates of effect sizes (e.g. Cohen's *d*, Pearson's *r*), indicating how they were calculated |

*Our web collection on statistics for biologists contains articles on many of the points above.*

## Software and code

Policy information about availability of computer code

| Data collection | CellSens (v1.15) was used for imaging soft agar assays. Fluoview (v3.0) was used for confocal microscope imaging for RNA-FISH. |
|---|---|
| Data analysis | The custom codes for 3'UTR splicing events identification, filtering, analysis of the features , have been made publicly available on GitHub at https://github.com/christear/RNASeq3USP Additionally, STAR (v2.52a), BEDTools (v2.29), featureCounts (v1.6.1), SAMtools (v1.8), Perl (v5.26) and R (v4.1.2) were used. ImageJ (v1.51j8 ) was used for anchorage-independent growth and RNA-FISH image processing and analysis. CellSens (v1.15) was used for cell migration analysis. |

For manuscripts utilizing custom algorithms or software that are central to the research but not yet described in published literature, software must be made available to editors and reviewers. We strongly encourage code deposition in a community repository (e.g. GitHub). See the Nature Portfolio guidelines for submitting code & software for further information.

## Data

Policy information about availability of data

All manuscripts must include a data availability statement. This statement should provide the following information, where applicable:
- Accession codes, unique identifiers, or web links for publicly available datasets
- A description of any restrictions on data availability
- For clinical datasets or third party data, please ensure that the statement adheres to our policy

All c3USPs across 10 TCGA cancer types and their corresponding normal tissues, as well as the patterns of splicing level, can be found on the SpUR database: http://www.cbrc.kaust.edu.sa/spur/home. It also provides a function to query the association between 3'UTR splicing levels and prognosis in each cancer type.

# Field-specific reporting

Please select the one below that is the best fit for your research. If you are not sure, read the appropriate sections before making your selection.

☒ Life sciences    ☐ Behavioural & social sciences    ☐ Ecological, evolutionary & environmental sciences

For a reference copy of the document with all sections, see nature.com/documents/nr-reporting-summary-flat.pdf

# Life sciences study design

All studies must disclose on these points even when the disclosure is negative.

| | |
|---|---|
| Sample size | No statistical methods were used to predetermine sample sizes. The 10 cancer cohorts used in this study were selected based on them having sufficient adjacent normal samples (>30). Eight tissues from GTEx were also included in this study since LUAD and LUSC shared the same normal control tissues, and HNSC did not have a good corresponding normal controls. As the study mainly focused on HCC, an additional two datasets of HCC samples, including one with a large sample size (PLANet dataset with 211 samples) and one with small sample size (in-house dataset with 4 matched pairs) were used to further confirm the findings based on the public dataset. To investigate the heterogeneities of 3'UTR splicing events across cancers, we also included a dataset from 55 AML patient samples (34 AML vs. 21 healthy controls). The datasets and number of samples selected were sufficiently large to show statistically significant differences. For biological experiments, sample sizes were selected based on similarly published research. |
| Data exclusions | 80 RNA samples from GTEx were excluded due to low sequencing depth (total number of splicing junctions < 1000). Sample numbers before and after exclusion are shown in Supplementary Table 1. No data were excluded for the biological experiments. |
| Replication | All experiments were replicated at least three times by two or more investigators. All repeats were reproducible. |
| Randomization | For computational analyses, samples were grouped by different tissues or cancer types and comparisons between tumors and their corresponding normal tissues were performed in each dataset.
Randomization was applied to all in vivo experiments but not in vitro experiments as it was not necessary. For biological experiments, at least two non-targeting/ scrambled control was used in all knockdown and ASO experiments for normalization and the identification of significant gene expression and splicing changes. The relevant empty vector was used as a within-batch control for all overexpression and luciferase studies. |
| Blinding | No blinding was performed for the computational analyses as these were performed using unbiased software programs or algorithms. Blinding was applied to the data collection of at least one set of each experiment except for RNA-FISH due to the experimental technicality and license requirement for confocal microscopy. |

# Reporting for specific materials, systems and methods

We require information from authors about some types of materials, experimental systems and methods used in many studies. Here, indicate whether each material, system or method listed is relevant to your study. If you are not sure if a list item applies to your research, read the appropriate section before selecting a response.

## Materials & experimental systems

| n/a | Involved in the study |
|---|---|
| ☐ | ☒ Antibodies |
| ☐ | ☒ Eukaryotic cell lines |
| ☒ | ☐ Palaeontology and archaeology |
| ☐ | ☒ Animals and other organisms |
| ☐ | ☒ Human research participants |
| ☒ | ☐ Clinical data |
| ☒ | ☐ Dual use research of concern |

## Methods

| n/a | Involved in the study |
|---|---|
| ☒ | ☐ ChIP-seq |
| ☒ | ☐ Flow cytometry |
| ☒ | ☐ MRI-based neuroimaging |

# Antibodies

| | |
|---|---|
| Antibodies used | Rabbit monoclonal Recombinant Anti-Cdk2 antibody [E304]; Abcam; Cat# ab32147; Lot# GR292523-12; 1:2,000 |

**Antibodies used**

Rabbit monoclonal Recombinant Anti-c-Myc antibody [Y69]; Abcam; Cat# ab32072; Lot# GR3232703-14; 1:2,000
Rabbit polyclonal Anti-THUMPD1 antibody; Abcam; Cat# ab199850; Lot# GR3176375-2; 1:2,000
Rabbit polyclonal Anti-U2AF65 antibody; Abcam; Cat# ab37530; Lot# GR3221592-4; 1:2,000
Rabbit monoclonal Axin2 (76G6); Cell Signaling; Cat# 2151; Lot# 10/2019-2; 1:1,000
Rabbit monoclonal Cyclin E1 (D7T3U); Cell Signaling; Cat# 20808; Lot# 04/2019-3; 1:2,000
Rabbit monoclonal CDK4 (D9G3E); Cell Signaling; Cat# 12790; Lot# 10/2017-4; 1:2,000
Mouse monoclonal CDK6 (DCS83); Cell Signaling; Cat# 3136; Lot# 06/2018-2; 1:2,000
Mouse monoclonal Chk1 (2G1D5); Cell Signaling; Cat# 2360; Lot# 09/2018-3; 1:2,000
Rabbit monoclonal β-Catenin (D10A8) XP®; Cell Signaling; Cat# 8480; Lot# 09/2018-5; 1:2,000
Rabbit monoclonal GAPDH (D16H11) XP®; Cell Signaling;  Cat# 5174; Lot# 10/2017-7; 1:10,000
Rabbit monoclonal HA-Tag (C29F4); Cell Signaling; Cat# 3724; Lot# 06/2019-9; 1:1,000
Rabbit monoclonal hnRNP C1/C2 (D6S3N); Cell Signaling; Cat# 91327; Lot# 07/2019-1; 1:1,000
Rabbit monoclonal p44/42 MAPK (Erk1/2) (137F5); Cell Signaling; Cat# 4695; Lot# 11/2018-21; 1:5,000
Rabbit monoclonal SF3B1 (D7L5T); Cell Signaling; Cat# 14434; Lot# 05/2019-1; 1:2,000
Rabbit monoclonal TDP43 (G400); Cell Signaling;  Cat# 3448; Lot#  01/2020-2; 1:1,000
Rabbit monoclonal TCF1/TCF7 (C63D9); Cell Signaling; Cat# 2203; Lot# 05/2019-8; 1:1,000
normal mouse IgG; Santa Cruz; Cat# sc-2025; Lot# F1818; 3μg for IP
Mouse monoclonal Anti-SAP 155 Antibody (B-3); Santa Cruz; Cat# sc-514655; Lot# B2621; 3μg for IP
Mouse monoclonal Anti-SF2/ASF Antibody (3G268); Santa Cruz; Cat# sc-73026; Lot# J2717; 1:5,000 for WB, 3μg for IP
Mouse monoclonal Anti-U2AF65 Antibody (MC3); Santa Cruz; Cat# sc-53942; Lot# C1521 ; 3μg for IP
Mouse monoclonal Anti-U1 snRNP 70 Antibody (C-3); Santa Cruz; Cat# sc-390899; Lot# G1219; 3μg for IP
Mouse monoclonal Anti-WDR55 Antibody (A-5);  Santa Cruz; Cat# sc-514225; Lot# D2518; 1:1,000

**Validation**

Rabbit monoclonal Recombinant Anti-Cdk2 antibody [E304]
Knockout validated in HAP1 cells for WB
https://www.abcam.com/cdk2-antibody-e304-ab32147.html

Rabbit monoclonal Recombinant Anti-c-Myc antibody [Y69]
Knockout validated in HEK-293T cells for WB
https://www.abcam.com/c-myc-antibody-y69-ab32072.html

Rabbit polyclonal Anti-THUMPD1 antibody
Validated using HeLa, 293T, Jurkat cell lysates for WB
https://www.abcam.com/thumpd1-antibody-ab199850.html

Rabbit polyclonal Anti-U2AF65 antibody
Validated using HeLa, Jurkat, A-431, HEK-293, HepG2, MCF-7, SHSY-5Y, U2OS cell lysates for WB
https://www.abcam.com/u2af65-antibody-ab37530.html

Rabbit monoclonal Axin2 (76G6)
Validated using HCT15 and SW620 cel lysates for WB
https://www.cellsignal.com/products/primary-antibodies/axin2-76g6-rabbit-mab/2151

Rabbit monoclonal Cyclin E1 (D7T3U)
Validated using HT-29 cell lysates +/- aphidicolin treatment
https://www.cellsignal.com/products/primary-antibodies/cyclin-e1-d7t3u-rabbit-mab/20808

Rabbit monoclonal CDK4 (D9G3E)
Validated using Jurkat, HeLa, MCF7 and COS-7 cell lysates for WB
https://www.cellsignal.com/products/primary-antibodies/cdk4-d9g3e-rabbit-mab/12790

Mouse monoclonal CDK6 (DCS83)
Validated using HeLa, IM-CD-3, C6 cell lysates for WB
https://www.cellsignal.com/products/primary-antibodies/cdk6-dcs83-mouse-mab/3136

Mouse monoclonal Chk1 (2G1D5)
Knockdown validated in HeLa cells for WB
https://www.cellsignal.com/products/primary-antibodies/chk1-2g1d5-mouse-mab/2360

Rabbit monoclonal β-Catenin (D10A8) XP®
Validated using HeLa, 293T, NIH3T3, C6 cell lysates for WB
https://www.cellsignal.com/products/primary-antibodies/b-catenin-d10a8-xp-rabbit-mab/8480

Rabbit monoclonal GAPDH (D16H11) XP®
Validated using HeLa, NIH3T3, C6, COS-7 cell lysates for WB
https://www.cellsignal.com/products/primary-antibodies/gapdh-d16h11-xp-rabbit-mab/5174

Rabbit monoclonal HA-Tag (C29F4)
Validated using untransfected vs. HA-FoxO4 or HA-Akt3 transfected HeLa cell lysates for WB
https://www.cellsignal.com/products/primary-antibodies/ha-tag-c29f4-rabbit-mab/3724

Rabbit monoclonal hnRNP C1/C2 (D6S3N)
Validated using HL60, MOLT4, IMR32, COS7 cell lysates for WB
https://www.cellsignal.com/products/primary-antibodies/hnrnp-c1-c2-d6s3n-rabbit-mab/91327

Rabbit monoclonal p44/42 MAPK (Erk1/2) (137F5)

Knockdown validated in HEK-293 cells for WB
https://www.cellsignal.com/products/primary-antibodies/p44-42-mapk-erk1-2-137f5-rabbit-mab/4695

Rabbit monoclonal SF3B1 (D7L5T)
Validated using PANC-1, HeLa, 3T3, H-e-II-E cell lysates for WB
https://www.cellsignal.com/products/primary-antibodies/sf3b1-d7l5t-rabbit-mab/14434

Rabbit monoclonal TDP43 (G400)
Validated using HeLa and rat brain cell lysates for WB
https://www.cellsignal.com/products/primary-antibodies/tdp43-g400-antibody/3448

Rabbit monoclonal TCF1/TCF7 (C63D9)
Validated using HT-29, Colo201, Jurkat and mouse thymocytes cell lysates for WB
https://www.cellsignal.com/products/primary-antibodies/tcf1-tcf7-c63d9-rabbit-mab/2203

normal mouse IgG
No validation information

Mouse monoclonal Anti-SAP 155 Antibody (B-3)
Validated using AMJ2-C8, A549, Raji, WEHI-231, Cak-1, BYDP cell lysates for WB
https://www.scbt.com/p/sap-155-antibody-b-3
Validated using Hep3B cell lysate for IP in lab

Mouse monoclonal Anti-SF2/ASF Antibody (3G268)
Validated using A-431, LADMAC, F9, C6, H19-7/IGF-IR cell lysates for WB
https://www.scbt.com/p/sf2-asf-antibody-3g268
Validated using Hep3B cell lysate for IP in lab

Mouse monoclonal Anti-U2AF65 Antibody (MC3)
Validated using HeLa, HEK293, Jurkat, SK-N-MC cell lysates for WB
https://www.scbt.com/p/u2af65-antibody-mc3
Validated using Hep3B cell lysate for IP in lab

Mouse monoclonal Anti-U1 snRNP 70 Antibody (C-3)
Validated using HepG2, Jurkat, RAW 264.7 cell lysates for WB
https://www.scbt.com/p/u1-snrnp-70-antibody-c-3
Validated using Hep3B cell lysate for IP in lab

Mouse monoclonal Anti-WDR55 Antibody (A-5)
Validated using K-562, SUP-T1, HL-60 cell lysates for WB
https://www.scbt.com/p/wdr55-antibody-a-5

# Eukaryotic cell lines

Policy information about cell lines

| Cell line source(s) | THLE-2 (ATCC), Hep3B (ATCC), HepG2 (ATCC), SNU398 (ATCC), DLD-1 (Horizon Discovery) |
|---|---|
| Authentication | All cell lines were authenticated by STR profiling by the suppliers when purchased (COAs available). |
| Mycoplasma contamination | All cell lines were routinely tested for mycoplasma contamination (every 3 months) and all tested negative. |
| Commonly misidentified lines (See ICLAC register) | No commonly misidentified cell lines were used in this study. |

# Animals and other organisms

Policy information about studies involving animals; ARRIVE guidelines recommended for reporting animal research

| Laboratory animals | CrTac:NCr-Foxn1<nu> (NCr nude) mice, female, 4-6 weeks old purchased from Invivos<br>The mice were housed in the following conditions: 23-24 °C, 44-58% humidity, 12 h/ 12 h dark/ light cycle (7 pm-7 am/ 7 am-7 pm). |
|---|---|
| Wild animals | No animals were used in this study. |
| Field-collected samples | No field-collected samples were used in this study. |
| Ethics oversight | All mouse work was performed in accordance to the NUS Institutional Animal Care and Use Committee (IACUC) guidelines. |

Note that full information on the approval of the study protocol must also be provided in the manuscript.

# Human research participants

Policy information about studies involving human research participants

| | |
|---|---|
| Population characteristics | Clinical information on the TCGA patients used in this study is available via the Genomic Data Commons (GDC) at https://portal.gdc.cancer.gov/. Information on population characteristics of the patients from which the in-house HCC, AML and PLANet samples were derived will be available before publication. |
| Recruitment | The patients in the PLANet study were recruited under a Translational and Clinical Research (TCR) Flagship Programme: Precision Medicine in Liver Cancer across an Asia Pacific NETwork (PLANet), funded by the Singapore National Medical Research Council (NMRC) programme. A total of 46 patients recruited from Singapore (National Cancer Centre Singapore, Singapore General Hospital, and National University Hospital), Thailand (National Cancer Institute Thailand) and Malaysia (University of Malaya Medical Centre) under the PLANet study were included in this study. PLANet recruited treatment-naïve patient with early stage liver cancer based on AASLD imaging criteria and required the patients to have no extra-hepatic metastasis (defined as lymph node <2 cm, lung modules < 1 cm, farther lymph nodes < 2 cm) with R0 or R1 resection and Child-Pugh ≤ 7 points without clinical ascites. Tumour (T), adjacent non-tumour liver tissue (N) and peripheral blood (P) were collected from these patients and subsequently processed for whole genome sequencing and RNAseq. There are no self-selection bias or other biases that may be present.

The AML patients recruited were consecutive AML patients presented to NUH who consented to have their samples stored for research. For normals, these are patient undergoing total knee replacements who consented. There is no specific inclusion criteria that will lead to bias. |
| Ethics oversight | The human studies were approved by the following Institutional Review Boards: the Domain Specific Review Board (DSRB) under the National Healthcare Group (NHG) in Singapore, the Central Institution Review Board (CIRB) of SingHealth, of which all National Cancer Center Singapore, Singapore General Hospital and National University Hospital were constituent members (CIRB Ref: 2016/2626 and 2018/2112), Medical Research Ethics Committee of UMMC (MREC ID NO: 201713-4729) and Research Committee of National Cancer Institute Thailand (Project Number: 174_2017C_OUT504). Each patient gave informed written consent. |

Note that full information on the approval of the study protocol must also be provided in the manuscript.

