## [Peer Review File · Nature Cell Biology]

Peer Review Information

Journal: Nature Cell Biology

Manuscript Title: Pan-cancer pervasive upregulation of 3'UTR splicing drives tumorigenesis

Corresponding author name(s): Yvonne Tay

Reviewer Comments & Decisions:

Decision Letter, initial version:

Subject: Decision on Nature Cell Biology submission NCB-T44994

Message:

*Please delete the link to your author homepage if you wish to forward this email to co-authors.

Dear Dr Tay,

Your manuscript, "Pan-cancer, pervasive upregulation of 3'UTR splicing drives tumorigenesis", has now been seen by 3 referees, who are experts in RNA splicing and cancer (referee 1); noncoding RNA and cancer (referee 2); and RNA splicing and omics analysis (referee 3). As you will see from their comments (attached below) they find this work of potential interest, but have raised substantial concerns, which in our view would need to be addressed with considerable revisions before we can consider publication in Nature Cell Biology.

Nature Cell Biology editors discuss the referee reports in detail within the editorial team, including the chief editor, to identify key referee points that should be addressed with priority, and requests that are overruled as being beyond the scope of the current study. To guide the scope of the revisions, I have listed these points below. I should stress that the referees' concerns point to a premature dataset and these points would need to be addressed with experiments and data, and reconsideration of the study for this journal and re-engagement of referees would depend on strength of these revisions.

In particular, it would be essential to:

a) provide more data to fully support that the 3' SP isoform is exported into the cytosol and gets translated, as noted by referee 3.

5. In figure 3a, they found that the splicing of CTNNB1 3'UTR was significantly correlated with patient survival, and focused on one of the variants, 3'SP, with fully spliced 3'UTR. However they did not define what is the "over-spliced" 3'SP, and did not show the direct evidence that the 3' SP isoform, not the FL isoform, is the one under active translation. Since this is one of the most important point of their paper, they need to use ribosome profile and translation reporter to validate such conclusion.

6. In figure 4, the authors claimed that CTNNB1 has oncogenic functions, and 3'SP variant is the one to be exported to cytoplasm and translated. They also claimed that U1SnRNP play a role in retaining FL variant of CTNNB1 in the nucleus. Although this model makes sense, I think they still need to provide more direct evidence to support this. For example, if the mutation in the 5' splice site of 3'UTR can increase the transport and translation of FL variant of CTNNB1, if the manipulation of 3' UTR splicing by ASO or SR proteins can affect CTNNB1 translation. In addition, they should use cell fractionation experiments (in addition to the microscopy in fig 4h) to quantify the cytoplasm export, as the relative levels of two isoforms in nucleus vs cytoplasm seems to be critical.

b) strengthen the relevance of 3'UTR splicing with tumorigenesis, as noted by referee 1:

-The data in Extended Figure 3g-i do not show clear differences in the 3 representative tumors versus normal tissues in LIHC, COAD, and BRCA. Given this, I am not clear on how robust the differences in CTNNB1 3' UTR isoforms actually are in tumor versus normal.

-While the authors use multiple orthogonal means to functionally manipulated 3' UTR splicing of CTNNB1 (e.g. ASOs, siRNAs, sgRNAs), only a single oligonucleotide is used in each experiment (e.g. Figure 2F) which is not sufficiently rigorous as there are well-defined off target effects of each of these modalities. This is particularly problematic for the ASO used for CTNNB1 where it is not clear that 3' UTR isoform usage has actually been specifically manipulated- the data in Figure 1f-g actually suggest that the SSO is merely suppressing CTNNB1 mRNA and protein expression rather than manipulating 3' UTR length. Similarly, I do not see primary data of the effects of the 3' UTR siRNA used in Figure 3d-e on 3' UTR splice isoform expression (the manner shown in Figure 2f).

c) All other referee concerns pertaining to strengthening existing data, providing controls, methodological details, clarifications and textual changes, should also be addressed.

d) Finally please pay close attention to our guidelines on statistical and methodological reporting (listed below) as failure to do so may delay the reconsideration of the revised manuscript. In particular please provide:

We would be happy to consider a revised manuscript that would satisfactorily address these points, unless a similar paper is published elsewhere, or is accepted for publication in Nature Cell Biology in the meantime.

- ensure that it conforms to our format instructions and publication policies (see below and www.nature.com/nature/authors/).

- provide a point-by-point rebuttal to the full referee reports verbatim, as provided at the end of this letter.

- provide the completed Editorial Policy Checklist (found here <https://www.nature.com/authors/policies/Policy.pdf>), and Reporting Summary (found here <https://www.nature.com/authors/policies/ReportingSummary.pdf>). This is essential for reconsideration of the manuscript and these documents will be available to editors and referees in the event of peer review. For more information see <http://www.nature.com/authors/policies/availability.html> or contact me.

Nature Cell Biology is committed to improving transparency in authorship. As part of our efforts in this direction, we are now requesting that all authors identified as 'corresponding author' on published papers create and link their Open Researcher and Contributor Identifier (ORCID) with their account on the Manuscript Tracking System (MTS), prior to acceptance. ORCID helps the scientific community achieve unambiguous attribution of all scholarly contributions. You can create and link your ORCID from the home page of the MTS by clicking on 'Modify my Springer Nature account'. For more information please visit www.springernature.com/orcid.

[REDACTED]

We would like to receive a revised submission within six months. We would be happy to consider a revision even after this timeframe, however if the resubmission deadline is missed and the paper is eventually published, the submission date will be the date when the revised manuscript was received.

We hope that you will find our referees' comments, and editorial guidance helpful. Please do not hesitate to contact me if there is anything you would like to discuss.

Best wishes,

Jie Wang

Jie Wang, PhD
Senior Editor
Nature Cell Biology

Tel: +44 (0) 207 843 4924
email: jie.wang@nature.com

Reviewers' Comments:

Reviewer #1:

Remarks to the Author:

This manuscript utilized public RNA-seq data to evaluate 3' UTR usage in cancer versus normal. The authors focus functional experiments on a specific 3' UTR splicing event in beta catenin 1 (CTNNB1). The concept behind this manuscript is interesting and the evaluation of 3' UTR usage in cancer is certainly an area in greater need of research. However, it is not clear how robust the claimed differences in 3' UTR isoform expression of CTNNB1 actually are. The experiments performed to manipulate 3' UTR isoform usage in CTNNB1 are not performed with sufficient rigor. Similarly, there are issues with the claimed mechanisms of 3' UTR usage and clinical prognostic relevance of this isoform. Finally, the manuscript writing needs to be greatly improved. The text contains unnecessary superlatives and claims of novelty which are not accurate in most instances (and not necessary to include). These and other points are noted below in comments for authors:

- The data in Extended Figure 3g-i do not show clear differences in the 3 representative tumors versus normal tissues in LIHC, COAD, and BRCA. Given this, I am not clear on how robust the differences in CTNNB1 3' UTR isoforms actually are in tumor versus normal.
- While the authors use multiple orthogonal means to functionally manipulated 3' UTR splicing of CTNNB1 (e.g. ASOs, siRNAs, sgRNAs), only a single oligonucleotide is used in each experiment (e.g. Figure 2F) which is not sufficiently rigorous as there are well-defined off target effects of each of these modalities. This is particularly problematic for the ASO used for CTNNB1 where it is not clear that 3' UTR isoform usage has actually been specifically manipulated- the data in Figure 1f-g actually suggest that the SSO is merely suppressing CTNNB1 mRNA and protein expression rather than manipulating 3' UTR length. Similarly, I do not see primary data of the effects of the 3' UTR siRNA used in Figure 3d-e on 3' UTR splice isoform expression (the manner shown in Figure 2f).
- The evaluation of RBP's regulating 3' UTR splicing of CTNNB1 is not well evaluated. SF3B1 and U2AF2 are required for nearly all splicing events so the claims of their specific role in CTNNB1 is not meaningful. SRSF1 regulates numerous aspects of mRNA metabolism in addition to splicing and there is no validation of SRSF1 regulation of CTNNB1 3' UTR splicing.
- It is not clear that the survival data in Figure 3a/c take into account known prognostic variables in HCC defined in PMID 28622513. Without this information, it is not clear how to evaluate the data in Figure 3a. Expression of distinct 3' UTR isoforms of CTNNB1 could be merely associated with more aggressive forms of HCC and this would not mean these isoforms drive adverse disease. Also, how were cutoffs for "over-spliced" and "under-spliced" defined?
- Despite the note in the Methods, it is not clear what exact material was used for the long-read RNA-seq data in Figure 1c and how these data relate to the TCGA tumors. There is insufficient description of these methods and analyses in the manuscript and it is not possible to ascertain the value of these data.
- There are almost certainly multiple 3' UTR isoforms and it is not clear how this is handled in Figure 1 which appears to illustrate only two potential isoforms if I understand correctly. It is also not clear "over-spliced" and "under-spliced" in Figure 1h refers to and I do not believe this terminology is appropriate or precise.
- I am not sure the claim that this manuscript provides the "first systematic and comprehensive analysis of splicing beyond coding region" is accurate. For example, there are numerous high profile publications of 3'UTR length (most of which are cited in the 2nd paragraph of the introduction) and intronic polyadenylation (PMID 30150773) in cancer in addition to multiple studies of RNA splicing that have been not restricted to protein coding transcripts. While there is definitely unique features to what is being performed in this manuscript, claims that this is the first paper to study altered splicing beyond protein-coding regions is disputable.
- The terms "over-spliced" and "under-spliced" are not helpful or precise as the comparator is not defined.
- The Abstract is too vaguely and imprecisely written. While the Abstract is obviously constrained by a strict word limit, the statements provided there are too vague. Here are examples:

---What does it mean “3’ UTR splicing is widespread, upregulated in cancers”? There is an abundant literature on 3’UTR shortening in cancer. Is that what the authors are referring to?
 ---It is not clear what CTNNB1 3’UTR isoform/event/impact is being referred to in the Abstract -----As noted above, there are multiple high-profile papers that have documented global 3’ UTR length in cancer and this point should be acknowledged (the statement that “3’ UTR splicing remains poorly understood” is not entirely accurate).
 ---How is targeted inhibition of 3’UTR shortening achieved?

Reviewer #2:

Remarks to the Author:

The manuscript by Chan et al deals with alternative 3’UTR splicing which is important in driving tumorigenesis. The paper is really well done and the initial part concerning the analysis of the 3’UTR splicing is full of control and details. I have few concerns in this manuscript:

- it would be interesting to perform a multivariate analysis to understand if higher 3’UTR splicing correlates with any specific clinical parameter in HCC.
- It’s not clear to me how you ruled out potential NMD: you picked only two proteins (which is the rationale?). Also, the real time in fig Extended 6E reports all values to 1, how do I know it’s not altering transcription? Moreover, where are the same data for CHEK1, mentioned in the manuscript? I would add them, at least in the extended data.
- Is the gene enrichment analysis done on tumor vs normal? How did you separate the over-spliced vs control counterpart? Because otherwise I won’t be sure that this enrichment is specifically correlated to the over-splicing, rather other potential variables.
- CTNNB1 protein levels in the human datasets? How are they? Are they correlated to the level of splicing?
- Not sure whether switching to colon cancer to confirm your data on liver makes a lot of sense. You chose a model after the first screening, stick to it.
- Why the siRNA against the 3SP leaves a little amount of protein but is way more efficient in decreasing cell growth respect to the one targeting CDS? Also, these cells are not even forming tumors *In vivo*. This section needs further explanations.
- In fig 4c looks like all the siRNAs are decreasing CTNNB1 protein levels. Why the mRNA levels are not consistent with protein ones?
- is protein stability somehow altered? Is the protein from the 3’UTR spliced more stable, other than more transcribed? It is discussed but no data are shown, it would be proper adding them.

Reviewer #3:

Remarks to the Author:

In this manuscript, the authors systematically analyzed the alternative splicing in 3'UTR, and identified lots of 3'UTR splicing events (3USPs) in tumor and normal samples which were presented in a database called SpUR. They found that 3'UTR splicing is widespread, upregulated in tumors, correlated with poor prognosis and occurred more frequently in oncogenes. The size reduction of 3'UTR in cancers was also reported previously, which is consistent with the findings in this work. Furthermore, they focused on the 3'UTR splicing of CTNNB1 that has an oncogenic function. They showed that this gene may be predominantly translated from the transcripts with spliced 3'UTR, and splicing in 3'UTR helps the transcripts to be exported into the cytoplasm. In summary, they discovered new regulation through splicing in the 3'UTR splicing, as well as the possible mechanisms by which the 3'UTR splicing of a specific gene promote carcinogenesis. This is a relatively comprehensive study but the clinical application is uncertain. In addition, their conclusion is a little overstated.

Comments:

1. In figure 1e, the overlap after clustering is confusing. For example, the overlap of AML and COAD is about 40% in the tenth row and first column while 80% in eleventh row and fifth column. It seems that these two number should be identical. In fact, I do not think the clustering is necessary here.
2. The figure legends should be more detailed. For example, the meaning of numbers in figure 1f should be mentioned in the legend, also the definition of favorable and unfavorable lacks details (e.g., how the correlation is defined as significant).
3. The figure 2f and extended figure 5d are two replicates, however, there are two bands in THUMPD1 3'FL in figure 2f, but only one band in Extended figure 5d. Please explain such data inconsistency.
4. The fourth section of the result entitled 'Targeted inhibition of 3'UTR splicing suppresses HCC carcinogenesis', however, this evidence is not enough to support this conclusion.
5. In figure 3a, they found that the splicing of CTNNB1 3'UTR was significantly correlated with patient survival, and focused on one of the variants, 3'SP, with fully spliced 3'UTR. However they did not define what is the "over-spliced" 3'SP, and did not show the direct evidence that the 3' SP isoform, not the FL isoform, is the one under active translation. Since this is one of the most important point of their paper, they need to use ribosome profile and translation reporter to validate such conclusion.
6. In figure 4, the authors claimed that CTNNB1 has oncogenic functions, and 3'SP variant is the one to be exported to cytoplasm and translated. They also claimed that U1SnRNP play a role in retaining FL variant of CTNNB1 in the nucleus. Although this model makes sense, I think they still need to provide more direct evidence to support this. For example, if the mutation in the 5' splice site of 3'UTR can increase the transport and translation of FL variant of CTNNB1, if the manipulation of 3' UTR splicing by ASO or SR proteins can affect CTNNB1 translation. In addition, they should use cell fractionation experiments (in addition to the microscopy in fig 4h) to quantify the cytoplasm export, as the relative levels of two isoforms in nucleus vs cytoplasm seems to be critical.
7. The database of SpUR should be more user-friendly.

Methods should be written concisely, but should contain all elements necessary to allow interpretation and replication of the results. As a guideline, Methods sections typically do not exceed 3,000 words. The Methods should be divided into subsections listing reagents and techniques. When citing previous methods, accurate references should be provided and any alterations should be noted. Information must be provided about: antibody dilutions, company names, catalogue numbers and clone numbers for monoclonal antibodies; sequences of RNAi and cDNA probes/primers or company names and catalogue numbers if reagents are commercial; cell line names, sources and information on cell line identity and authentication. Animal studies and experiments involving human subjects must be reported in detail, identifying the committees approving the protocols. For studies involving human subjects/samples, a statement must be included confirming that informed consent was obtained. Statistical analyses and

information on the reproducibility of experimental results should be provided in a section titled “Statistics and Reproducibility”.

All Nature Cell Biology manuscripts submitted on or after March 21 2016 must include a Data availability statement at the end of the Methods section. For Springer Nature policies on data availability see <http://www.nature.com/authors/policies/availability.html>; for more information on this particular policy see <http://www.nature.com/authors/policies/data/data-availability-statements-data-citations.pdf>. The Data availability statement should include:

- Accession codes for primary datasets (generated during the study under consideration and designated as "primary accessions") and secondary datasets (published datasets reanalysed during the study under consideration, designated as "referenced accessions"). For primary accessions data should be made public to coincide with publication of the manuscript. A list of data types for which submission to community-endorsed public repositories is mandated (including sequence, structure, microarray, deep sequencing data) can be found here <http://www.nature.com/authors/policies/availability.html#data>.
- Unique identifiers (accession codes, DOIs or other unique persistent identifier) and hyperlinks for datasets deposited in an approved repository, but for which data deposition is not mandated (see here for details <http://www.nature.com/sdata/data-policies/repositories>).
- At a minimum, please include a statement confirming that all relevant data are available from the authors, and/or are included with the manuscript (e.g. as source data or supplementary information), listing which data are included (e.g. by figure panels and data types) and mentioning any restrictions on availability.
- If a dataset has a Digital Object Identifier (DOI) as its unique identifier, we strongly encourage including this in the Reference list and citing the dataset in the Methods.

We recommend that you upload the step-by-step protocols used in this manuscript to the Protocol Exchange. More details can found at www.nature.com/protocolexchange/about.

FIGURES – Colour figure publication costs \$600 for the first, and \$300 for each subsequent colour figure. All panels of a multi-panel figure must be logically connected and arranged as they would appear in the

final version. Unnecessary figures and figure panels should be avoided (e.g. data presented in small tables could be stated briefly in the text instead).

All imaging data should be accompanied by scale bars, which should be defined in the legend. Cropped images of gels/blots are acceptable, but need to be accompanied by size markers, and to retain visible background signal within the linear range (i.e. should not be saturated). The boundaries of panels with low background have to be demarked with black lines. Splicing of panels should only be considered if unavoidable, and must be clearly marked on the figure, and noted in the legend with a statement on whether the samples were obtained and processed simultaneously. Quantitative comparisons between samples on different gels/blots are discouraged; if this is unavoidable, it should only be performed for samples derived from the same experiment with gels/blots were processed in parallel, which needs to be stated in the legend.

- For line art, graphs, charts and schematics we prefer Adobe Illustrator (.AI), Encapsulated PostScript (.EPS) or Portable Document Format (.PDF). Files should be saved or exported as such directly from the application in which they were made, to allow us to restyle them according to our journal house style.
- We accept PowerPoint (.PPT) files if they are fully editable. However, please refrain from adding PowerPoint graphical effects to objects, as this results in them outputting poor quality raster art. Text used for PowerPoint figures should be Helvetica (preferred) or Arial.
- We do not recommend using Adobe Photoshop for designing figures, but we can accept Photoshop generated (.PSD or .TIFF) files only if each element included in the figure (text, labels, pictures, graphs, arrows and scale bars) are on separate layers. All text should be editable in 'type layers' and line-art such as graphs and other simple schematics should be preserved and embedded within 'vector smart objects' - not flattened raster/bitmap graphics.

The total number of Supplementary Figures (not including the “unprocessed scans” Supplementary Figure) should not exceed the number of main display items (figures and/or tables (see our Guide to Authors and March 2012 editorial <http://www.nature.com/ncb/authors/submit/index.html#suppinfo>; <http://www.nature.com/ncb/journal/v14/n3/index.html#ed>). No restrictions apply to Supplementary Tables or Videos, but we advise authors to be selective in including supplemental data.

GUIDELINES FOR EXPERIMENTAL AND STATISTICAL REPORTING

REPORTING REQUIREMENTS – To improve the quality of methods and statistics reporting in our papers we have recently revised the reporting checklist we introduced in 2013. We are now asking all life sciences authors to complete two items: an Editorial Policy Checklist (found here <https://www.nature.com/authors/policies/Policy.pdf>) that verifies compliance with all required editorial policies and a reporting summary (found here <https://www.nature.com/authors/policies/ReportingSummary.pdf>) that collects information on experimental design and reagents. These documents are available to referees to aid the evaluation of the manuscript. Please note that these forms are dynamic ‘smart pdfs’ and must therefore be downloaded and completed in Adobe Reader. We will then flatten them for ease of use by the reviewers. If you would like to reference the guidance text as you complete the template, please access these flattened versions at <http://www.nature.com/authors/policies/availability.html>.

STATISTICS – Wherever statistics have been derived the legend needs to provide the n number (i.e. the sample size used to derive statistics) as a precise value (not a range), and define what this value represents. Error bars need to be defined in the legends (e.g. SD, SEM) together with a measure of centre (e.g. mean, median). Box plots need to be defined in terms of minima, maxima, centre, and percentiles. Ranges are more appropriate than standard errors for small data sets. Wherever statistical

significance has been derived, precise p values need to be provided and the statistical test used needs to be stated in the legend. Statistics such as error bars must not be derived from $n < 3$. For sample sizes of $n < 5$ please plot the individual data points rather than providing bar graphs. Deriving statistics from technical replicate samples, rather than biological replicates is strongly discouraged. Wherever statistical significance has been derived, precise p values need to be provided and the statistical test stated in the legend.

Author Rebuttal to Initial comments

Response to reviewers

We thank the editor and reviewers for their insightful comments, which we address below. We have performed additional experiments and revised the manuscript in light of this feedback (amendments are indicated in red font).

Reviewer #1:

Remarks to the Author:

This manuscript utilized public RNA-seq data to evaluate 3' UTR usage in cancer versus normal. The authors focus functional experiments on a specific 3' UTR splicing event in beta catenin 1 (CTNNB1). The concept behind this manuscript is interesting and the evaluation of 3' UTR usage in cancer is certainly an area in greater need of research. However, it is not clear how robust the claimed differences in 3' UTR isoform expression of CTNNB1 actually are. The experiments performed to manipulate 3' UTR isoform usage in CTNNB1 are not performed with sufficient rigor. Similarly, there are issues with the claimed mechanisms of 3' UTR usage and clinical prognostic relevance of this isoform. Finally, the

manuscript writing needs to be greatly improved. The text contains unnecessary superlatives and claims of novelty which are not accurate in most instances (and not necessary to include). These and other points are noted below in comments for authors:

1. The data in Extended Figure 3g-i do not show clear differences in the 3 representative tumors versus normal tissues in LIHC, COAD, and BRCA. Given this, I am not clear on how robust the differences in CTNNB1 3' UTR isoforms actually are in tumor versus normal.

>>We thank the reviewer for pointing out this problem, which has helped us improve the figure. To calculate splicing levels, we adapted a well-established metrics called percentage of splicing in (PSI) (Wang et al, 2008, Nature), which has also been used in previous TCGA pan-cancer splicing studies (Ryan et al, 2016, Nucleic Acids Research, Kahles et al, 2018, Cancer Cell). In brief, PSI was calculated using RNA-seq reads supporting splicing in, divided by the sum of reads supporting both splicing in and splicing out. Here, we define splicing levels by the reads supporting a spliced variant (splicing out) divided by reads supporting both spliced (splicing out) and unspliced (splicing in) variants, which is equivalent to $1 - \text{PSI}$ (**Rebuttal Fig. 1a**). We also compare the splicing levels of CTNNB1 3' UTR calculated by our method (SpUR) with results from TCGASpliceSeq (Ryan et al, 2016, Nucleic Acids Research), and observe a highly consistent trend (**Rebuttal Fig. 1b**). In our previous Extended Data Fig. 3g-i, we showed only reads supporting the spliced variant (splicing out). For improved clarity, we have added the percentage of splicing level in brackets (**Rebuttal Fig. 1c** and **Extended Data Fig. 3g-j**). Both the boxplot and IGV plot show very significant splicing level differences of CTNNB1 3' SP between the tumor and normal samples.

Rebuttal Fig. 1: Quantification of 3'UTR splicing levels using RNA-seq data. (a) Schematic showing the method we used to quantify 3'UTR splicing levels. (b) Distribution of CTNNB1 3'UTR splicing levels in TCGA-LIHC tumor and the adjacent normal samples derived from TCGASpliceSeq and quantified using our method, SpUR. (c) IGV plot illustrating the number of reads supporting 3'UTR splicing and splicing levels (percentages in brackets) in COAD tumor and normal samples.

2. While the authors use multiple orthogonal means to functionally manipulated 3' UTR splicing of CTNNB1 (e.g. ASOs, siRNAs, sgRNAs), only a single oligonucleotide is used in each experiment (e.g. Figure 2F) which is not sufficiently rigorous as there are well-defined off target effects of each of these modalities. This is particularly problematic for the ASO used for CTNNB1 where it is not clear that 3' UTR isoform usage has actually been specifically manipulated- the data in Figure 1f-g actually suggest that the SSO is merely suppressing CTNNB1 mRNA and protein expression rather than manipulating 3' UTR length. Similarly, I do not see primary data of the effects of the 3' UTR siRNA used in Figure 3d-e on 3' UTR splice isoform expression (the manner shown in Figure 2f).

>>We note the reviewer's concern here. To improve the rigor of our experiments and reinforce our conclusions, we have repeated the 3'UTR splice-blocking and CTNNB1 knockdown experiments with additional/new ASOs (**Extended Data Fig. 6**) and siRNAs (**Extended Data Fig. 9**). Due to space constraint, we showed only the quantification for the ASO anchorage-independent growth assay in the Extended Data Figures. The colony images are shown below in **Rebuttal Fig. 2a** and the migration data for siRNA-mediated CTNNB1 knockdown are shown in **Rebuttal Fig. 2b,c**. The data from the new ASOs and siRNAs are consistent with the original data on the manuscript, suggesting that the effects observed are not off-target effects but are due to specific 3'UTR splice-blocking or CTNNB1 knockdown. Amendments to the text in the relevant sections are shown below:

"These effects were accompanied by a decrease in the protein expression of the respective genes and the repression of tumor growth, which we confirmed with additional ASOs (Fig. 2g,h and Extended Data Fig. 5e,f,6)."

"These effects were further verified with additional siRNA-mediated knockdowns to reinforce our observations (Extended Data Fig. 9)."

For the ASO-mediated 3'UTR splice-blocking of CTNNB1, we have repeated the 3'UTR PCR and replaced the old images with the better quality repeats (**Fig. 2f, Extended Data Fig. 5d**). These and the PCR images for the new CTNNB1 ASO-SS (**Extended Data Fig. 6c**) highlight the opposite changes in the two CTNNB1 3'UTR transcript variants whereby ASO-SS decreases the spliced 3'UTR (3'SP) while increasing the full length (3'FL) 3'UTR. This indicates that the splice-inhibiting ASOs are manipulating the 3'UTR length rather than merely suppressing CTNNB1 and protein expression. These are also shown below in **Rebuttal Fig. 2d**.

Additionally, we have added the PCR gel images to show changes in the 3'UTR isoform and CDS expression upon siRNA-mediated CTNNB1 knockdown (**Extended Data Fig. 9b, Rebuttal Fig. 2e**). The gel image shows that both siRNAs targeting 3'FL and 3'SP respectively, specifically reduce their intended target (3'FL and 3'SP as indicated by the arrowheads).

Rebuttal Fig. 2: (a) Effect of ASO-mediated 3'UTR splicing inhibition on anchorage-independent growth

in Hep3B and HepG2. (b,c) Effect of siRNA-mediated CTNNB1 knockdown on cell migration in Hep3B (b) and SNU398 (c). (d,e) PCR gel images of CTNNB1 3'UTR, CDS and GAPDH for Hep3B and HepG2 cells treated with ASOs (d) or CTNNB1-targeting siRNAs (e). ASO: antisense oligonucleotide; NC: non-targeting control; SS: splice site; CDS: coding sequence; 3'FL: full length 3'UTR; 3'SP: spliced 3'UTR.

3. The evaluation of RBP's regulating 3' UTR splicing of CTNNB1 is not well evaluated. SF3B1 and U2AF2 are required for nearly all splicing events so the claims of their specific role in CTNNB1 is not meaningful. SRSF1 regulates numerous aspects of mRNA metabolism in addition to splicing and there is no validation of SRSF1 regulation of CTNNB1 3' UTR splicing.

>>We note the reviewer's comment and agree that these RBPs could also be involved in other splicing events while other splicing factors/ RBPs could be involved in the regulation of CTNNB1 and other 3'UTR splicing events. However, we would like to clarify that we do not claim that the three RBPs SF3B1, SRSF1 and U2AF2 specifically only regulate 3'UTR splicing in CTNNB1. In the context of our study, the RBPs were shortlisted based on CTNNB1-focused analysis of the ENCODE RBP knockdown and eCLIP data. Of the RBPs we validated, only the knockdown of SF3B1, SRSF1 and U2AF2 could simultaneously reduce CTNNB1 3'SP and increase 3'FL transcript expression without significantly affecting the CDS exon-exon junctions (**Extended Data Fig. 11c**), which are collectively indicative of the involvement of these RBPs in the regulation of CTNNB1 3'UTR rather than CDS splicing. Although these three RBPs are general splicing regulators, our knockdown data suggest that CTNNB1 3'UTR splicing is more sensitive to the perturbation of these splicing factors and could potentially be more selectively regulated by specific splicing factors compared to the CDS region where splicing may be more robust.

We have also added RNA immunoprecipitation and RNA pulldown data to show that these three RBPs are able to enrich for the CTNNB1 3'UTR variants and vice versa, suggesting potential associations and modulation of CTNNB1 3'UTR splicing (**Fig. 4d,e, Rebuttal Fig. 3**). Please see below for the new figures and addition to the main text.

“We also showed that RNA immunoprecipitation (RIP) of SRSF1 and U2AF2 significantly enriched for both CTNNB1 3'UTR variants, while SF3B1 RIP only enriched for CTNNB1 3'FL (Fig. 4d). We further verified these associations by pulling down the CTNNB1 transcripts whereby SRSF1 and U2AF2 were enriched by the antisense 3'FL and 3'SP probes, and consistent with the RIP results, enrichment of SF3B1 was only observed for the 3'FL pulldown (Fig. 4e). These observations suggest that these RBPs may associate with the CTNNB1 3'UTR and modulate its splicing.”

Rebuttal Fig. 3: (a) Enrichment of the CTNNB1 3'FL and 3'SP transcripts by RBPs SF3B1, SRSF1 and U2AF2 RNA immunoprecipitation (RIP) normalized to IgG in Hep3B and SNU398. (b) Enrichment of SF3B1, SRSF1 and U2AF2 RBPs upon CTNNB1 3'FL and 3'SP pulldown using biotinylated antisense (AS) probes. Sense (S) probes and HSP90 served as negative controls. 3'FL: full length 3'UTR; 3'SP: spliced 3'UTR.

4. It is not clear that the survival data in Figure 3a/c take into account known prognostic variables in HCC defined in PMID 28622513. Without this information, it is not clear how to evaluate the data in Figure 3a. Expression of distinct 3' UTR isoforms of CTNNB1 could be merely associated with more aggressive forms of HCC and this would not mean these isoforms drive adverse disease. Also, how were cutoffs for “over-spliced” and “under-spliced” defined?

>>We thank the reviewer for his/her questions. As suggested by the reviewer, we explored the possible compounding effect of factors from PMID 28622513 on our CTNNB1 survival analysis. The study classified TCGA-LIHC tumor samples into three clusters (iCluster1/2/3) by integrating molecular features from five platforms, including DNA copy number, DNA methylation, mRNA expression, miRNA expression and reverse phase protein array (RPPA) data. Similar to the study which did not observe a prognostic difference in the TCGA-LIHC cohort, we also do not observe any significant differences in the CTNNB1 3'UTR splicing levels across these three clusters (**Rebuttal Fig. 4a**). In our study, we adapted the approach used by Uhlen et al to build a pathology atlas of the human cancer transcriptome (Uhlen et al, Science, 2017) for our survival analysis. Instead of using gene expression, we studied the association between 3'UTR splicing and patients' overall survival. These molecular features are widely used in large-scale data analysis, and could include other features, such as promoter usage shown in a recent study (Demircioglu et al, Cell, 2019). Here, we observed that the overall survival is significantly correlated with CTNNB1 3'UTR splicing level, but not the mRNA expression. We also performed a multivariate analysis to correlate 3'UTR splicing levels with various clinical parameters and observed a positive correlation between 3'UTR splicing levels and patient neoplasm histologic grade (**Rebuttal Table 1**). We agree that it does not necessarily mean that increased CTNNB1 3'UTR splicing level drives cancer progression. Thus, we followed up with additional *in vitro* and *in vivo* experiments to demonstrate that the CTNNB1 spliced 3'UTR isoform could impact tumorigenesis.

Clinical/Molecular variable	Categories/Ranges	Correlation	p-value
-------------------	-------------	---------

Grade *	G1, G2, G3, G4	0.110	3.50E-02
Hoshida subclass *	S1, S2, S3	-0.23	9.70E-04

Rebuttal Table 1: Correlation between 3'UTR splicing levels and HCC neoplasm histologic grade. The p-values were calculated using the Wilcoxon test for factors that only have two categories and correlation test for factors with multiple categories or continuous values, respectively.

We apologize for the unclear definition of the terms “over-splicing” and “under-splicing”. These were adapted from the method in a well-known cancer genomics database COSMIC, Catalogue Of Somatic Mutations In Cancer (<https://cancer.sanger.ac.uk/cosmic>), in which the over- and under-expression of each gene in each cancer type are defined by comparing the expression of the gene in tumor samples to that from normal samples. In our study, for each significantly dysregulated c3USP (FDR < 0.1, difference in the median of splicing levels > 5% between tumor and normal) in each cancer type, we first extracted its splicing level distribution in the normal samples (left box of **Rebuttal Fig. 4b**). Next, we compared its splicing level in each tumor sample to that in the normal samples. An event with splicing levels higher than 90% quantile of the normal samples was defined as over-spliced. Conversely, an event with splicing levels below the 10% quantile of the normal samples was considered under-spliced. The advantage of this method was that it allowed us to obtain a standardized quantification (0-100%) of the dysregulation of each event in each cancer type, which could be applied for pan-cancer analysis. We have also added this figure to the manuscript to better illustrate these definitions (**Extended Data Fig. 3f**).

Rebuttal Fig. 4: (a) Distribution of CTNNB1 3'UTR splicing levels in three subclasses of the TCGA-LIHC iCluster1/2/3 (iC1/2/3) and adjacent normal samples. (b) Boxplot illustrating the definition of over- and under-splicing for each significantly dysregulated common 3'UTR splicing event in each cancer type. Each dot represents one sample and the dashed lines indicate the 10% and 90% quantile of splicing

levels in the normal samples.

5. Despite the note in the Methods, it is not clear what exact material was used for the long-read RNA-seq data in Figure 1c and how these data relate to the TCGA tumors. There is insufficient description of these methods and analyses in the manuscript and it is not possible to ascertain the value of these data.

>>The long-read RNA-seq data is released by PacBio, which performed third-generation sequencing on the brain, liver and heart tissues, as well as the MCF7 cell line (mentioned in the “Processed public datasets” Methods section). The PacBio data we used are indicated in the ‘Data availability’ section. We have also added the following explanation of our method of analysis to the Methods section:

“Due to the shallow sequencing depth, we combined all the identified transcripts from these four cell types. The 3’UTR splicing events that overlapped with the PacBio-identified isoforms with identical 5’ and 3’ splice sites were considered as being supported by PacBio.”

It is worth noting that we could still observe a ~50% overlap of the c3USPs with the PacBio-identified transcripts.

6. There are almost certainly multiple 3’ UTR isoforms and it is not clear how this is handled in Figure 1 which appears to illustrate only two potential isoforms if I understand correctly. It is also not clear “over-spliced” and “under-spliced” in Figure 1h refers to and I do not believe this terminology is appropriate or precise.

>>As shown in **Rebuttal Fig. 1a**, for each 3’UTR splicing event, we calculate the splicing level by comparing it with the corresponding unspliced isoform. For multiple splicing events within the same 3’UTR, we calculate them individually. This approach has also been used in many previous studies, such as the TCGA pan-cancer splicing study (Ryan et al, 2016, Nucleic Acids Research). Please refer to our response to comment 4 for the detailed explanation of the terms over-spliced and under-spliced, which is also illustrated in **Rebuttal Fig. 4b** and **Extended Data Fig. 3f**.

7. I am not sure the claim that this manuscript provides the “first systematic and comprehensive analysis of splicing beyond coding region” is accurate. For example, there are numerous high profile publications of 3’UTR length (most of which are cited in the 2nd paragraph of the introduction) and intronic polyadenylation (PMID 30150773) in cancer in addition to multiple studies of RNA splicing that have been not restricted to protein coding transcripts. While there are definitely unique features to what is being performed in this manuscript, claims that this is the first paper to study altered splicing beyond protein-coding regions is disputable.

>>We apologize for the unclear writing. We would like to clarify that what we wanted to convey was that this is the first systematic analysis of splicing in the 3' untranslated regions in cancer. We have amended the text accordingly (please see below).

“Here, we provide the first systematic and comprehensive analysis of 3'UTR splicing in cancers.”

8. The terms “over-spliced” and “under-spliced” are not helpful or precise as the comparator is not defined.

>>Please see our responses to comments 4 and 6 above regarding the derivation (including method of calculation) and definition of the terms over- and under-splicing.

9. The Abstract is too vaguely and imprecisely written. While the Abstract is obviously constrained by a strict word limit, the statements provided there are too vague. Here are examples:

---What does it mean “3' UTR splicing is widespread, upregulated in cancers”? There is an abundant literature on 3'UTR shortening in cancer. Is that what the authors are referring to?

>>We would like to clarify that our study focuses on splicing within the 3'UTR, which is distinct from APA and cleavage that leads to 3'UTR shortening from its 3' end. To address this confusion, we have amended the abstract to better distinguish the two events (please see below).

“In cancer, shortening at 3'UTR ends via alternative polyadenylation can activate oncogenes. However, internal 3'UTR splicing remains poorly understood as splicing studies have traditionally focused on protein-coding alterations.”

---It is not clear what CTNNB1 3'UTR isoform/event/impact is being referred to in the Abstract. As noted above, there are multiple high-profile papers that have documented global 3' UTR length in cancer and this point should be acknowledged (the statement that “3' UTR splicing remains poorly understood” is not entirely accurate).

>>Please refer to our response to the comment above for our clarification and amendments to the text.

---How is targeted inhibition of 3'UTR shortening achieved?

>>Again, we would like to clarify that we are not studying shortening from 3'UTR ends as described in literature. Instead, our study focuses on splicing within the 3'UTR. However, we have amended the text for better accuracy (please see below).

“We show that antisense oligonucleotide-mediated inhibition of 3'UTR splicing efficiently reduces oncogene expression and impedes tumor progression.”

Reviewer #2:

Remarks to the Author:

The manuscript by Chan et al deals with alternative 3'UTR splicing which is important in driving tumorigenesis. The paper is really well done and the initial part concerning the analysis of the 3'UTR splicing is full of control and details. I have few concerns in this manuscript:

1. it would be interesting to perform a multivariate analysis to understand if higher 3'UTR splicing correlates with any specific clinical parameter in HCC.

>>We thank the reviewer for the useful suggestion. To inspect potential factors that could impact or be associated with 3'UTR splicing, we correlated the 3'UTR splicing levels with multiple clinical parameters across the HCC patients, including age, gender, race, weight, tumor grade, stage, RS65 scores, CTNNB1 mutation, TERT promoter mutation, TP53 mutation, Hoshida subclass and HB16 clusters (**Rebuttal Table 2**). We observe a positive correlation between 3'UTR splicing levels and patient neoplasm histologic grade. In addition, 3'UTR splicing levels are significantly higher in patients from subclass 1 and 2, compared to subclass 3 based on a previously defined molecular subclass of HCC (Hoshida et al, Cancer Research, 2009). Interestingly, Hoshida et al also reported that subclass 1 was characterized by aberrant activation of the WNT signaling pathway, and subclass 2 by increased proliferation as well as activation of MYC and AKT. These results suggest that high neoplasm histologic grades, aberrant activation of the WNT pathway, MYC and AKT, may be the mechanisms underlying the poor prognosis of patients with high 3'UTR splicing levels (**Fig. 2b**).

Clinical/Molecular variable	Categories/Ranges	Correlation/Median	p-value
Age (Year)	16-90	-0.026	7.42E-01
Gender	Female, Male	0.608, 0.602	9.62E-01
Race	Hispanic/Latino, not Hispanic/Latino	0.593, 0.602	5.29E-01
Weight (kg)	40-172	-0.063	2.53E-01
Grade *	G1, G2, G3, G4	0.110	3.50E-02
Stage	Stage i, ii, iii, iv	0.001	9.85E-01
RS65 score	0-100	0.034	6.36E-01
CTNNB1 mutation	Mutant, wildtype	0.621, 0.608	4.14E-01
TERT promoter mutation	Mutation, no mutation	0.608, 0.611	7.96E-01
TP53 mutation	Mutant, wildtype	0.611, 0.607	2.20E-01
Hoshida subclass *	S1, S2, S3	-0.23	9.70E-04
HB16 cluster	C1, C2	0.607, 0.612	5.11E-01

Rebuttal Table 2: Summary of the correlation between 3'UTR splicing levels and various clinical and molecular parameters. The p-values were calculated using the Wilcoxon test for factors that only have two categories and correlation test for factors with multiple categories or continuous values,

respectively. The p-values in red and the asterisks (*) indicate the significant variables.

2. It's not clear to me how you ruled how potential NMD: you picked only two proteins (which is the rationale?). Also, the real time in fig Extended 6E reports all value to 1, how do I know it's not altering transcription? Moreover, where are the same data for CHEK1, mentioned in the manuscript? I would add them, at least in the extended data.

>>We note the reviewer's comments and have added the data for the remaining 3USP candidates (MAPK1, THUMPD1 and WDR55) following UPF1 knockdown (**Extended Data Fig. 7f,g, Rebuttal Fig. 5**). The data for CHEK1 are on the original figures (labeled "CHEK1 3'SP" on the x-axis of **Extended Data Fig. 7f**, and "CHEK1" in the western blot in **Extended Data Fig. 7g**)

The real time qPCR data in **Extended Data Fig. 7f** shows the fold change of each target gene normalized to the non-targeting control siRNA (as stated in the y-axis title, siNC is depicted by the dotted line). The normalized transcript levels of the various 3'SP are reported to 1 or values close to 1 as their transcription is unchanged, as opposed to the positive controls (ATF4, GAS5, RP9P and SMG5) upon UPF1 knockdown. These data support our hypothesis whereby transcripts undergoing 3'UTR splicing within a 50nt distance to the stop codon are not subjected to NMD regulation. Please see below for the new figures and addition to the main text.

"As the CTNNB1 3'UTR is spliced 11/12nt downstream of the stop codon, we ruled out splicing-induced NMD by knocking down a key NMD regulator, UPF1, which did not alter CTNNB1 transcript and protein expression (Extended Data Fig. 7e-g). This was also observed for other 3USP candidates: CHEK1, MAPK1, THUMPD1 and WDR55 (Extended Data Fig. 7f,g)."

Rebuttal Fig. 5: The effect of UPF1 knockdown on the 3' SP transcript (a) and protein (b) expression of CTNNB1, CHEK1, MAPK1, THUMPD1 and WDR55. 3' SP: spliced 3' UTR; siNC: non-targeting siRNA control.

3. Is the gene enrichment analysis done on tumor vs normal? How did you separate the over-spliced vs control counterpart? Because otherwise I won't be sure that this enrichment is specifically correlated to the oversplicing, rather other potential variables.

>>We thank the reviewer for raising this question. To address this, we have added a more detailed description in the main text and figure legend (please see below).

“As CTNNB1 plays critical roles in adherens junction formation and WNT signaling to regulate cell proliferation and migration, we first performed gene set enrichment analysis (GSEA) by comparing two groups of the tumors samples: (1) tumor samples with over-spliced CTNNB1 3' SP and (2) the rest of the tumor samples.”

Figure legend: “Gene set enrichment analysis (GSEA) by comparing two groups of tumors (as in Fig. 3a)”

This is consistent with our approach for the survival analysis as shown in Fig. 3a. Since the comparison is between two groups of tumor samples, it should exclude the compounding effect of other potential variables arising from the differences between the tumor and normal samples. Moreover, we could observe significant correlations for the same enriched gene sets in two independent datasets, TCGA (Fig. 3c) and PLANet (Extended Data Fig. 8a), suggesting that this enrichment is specifically correlated to CTNNB1 3' UTR over-splicing.

4. CTNNB1 protein levels in the human datasets? How are they? Are they correlated to the level of splicing?

>>The reverse phase protein array (RPPA) data available in TCGA profiled the expression of ~100 proteins, including CTNNB1. Among the 370 TCGA-LIHC tumor samples, 176 of them have RPPA data. We checked for correlation of CTNNB1 RPPA protein expression and total RNA expression ($r = 0.034$, p -value = 0.656), as well as the spliced 3'UTR variant expression ($r = -0.063$, p -value = 0.409), whereby both correlations were insignificant. This could be due to the lack of tumor samples (less than half) with protein expression data, therefore the trend may not hold true across all 370 samples. Due to the lack of protein expression datasets and the lack of depth in the mass spectrometry dataset, we could not perform further correlation analysis.

However, as we have in-house COAD patient samples and have shown that the CTNNB1 3'UTR splicing trend and effects are similar between LIHC and COAD, we performed both qPCR and western blot analysis of the CTNNB1 transcript and protein expression in these samples. We found that the CTNNB1 3'SP transcript and CTNNB1 protein expression are upregulated in the tumor samples and positively correlated in seven out of nine matched pairs (Rebuttal Fig. 6).

Rebuttal Fig. 6: (a) CTNNB1 transcript and protein expression in nine matched pairs of colon adenocarcinoma and adjacent normal patient samples. The tables consist of the relative CTNNB1 3'SP transcript expression (second row) and quantification of the CTNNB1 protein expression (last row). (b) Comparison of the average CTNNB1 3'SP transcript and protein levels between samples with no change in 3USP (3USP no change) and upregulated 3USP (3USP up). N: adjacent normal; T: tumor; 3'SP: spliced

3'UTR; 3USP: 3'UTR splicing.

5. Not sure whether switching to colon cancer to confirm your data on liver makes a lot of sense. You chose a model after the first screening, stick to it.

>>We note the reviewer's concern. We would like to clarify that liver cancer remains the focus of the study. The colon cancer data were added to support our observations in liver cancer and show that the dysregulated 3'UTR splicing event in CTNNB1 and its effects are not limited to liver cancer. The rationale for including the supporting data from colon cancer is mentioned in the main text (please see below).

"As global 3'UTR splicing is significantly increased in COAD and associated with poorer OS, while CTNNB1 3'UTR splicing is also upregulated in COAD (Extended Data Fig. 10a-e), we performed the same ASO and siRNA treatments in COAD cell lines, DLD-1 and HCT116, which resulted in similar phenotypic effects to that in the HCC cells (Extended Data Fig. 10f-m, data not shown for HCT116)."

6. Why the siRNA against the 3SP leaves a little amount of protein but is way more efficient in decreasing cell growth respect to the one targeting CDS? Also, these cells are not even forming tumors *In vivo*. This section needs further explanations.

>>The CDS-targeting siRNA (si-CDS) results in a greater or similar level of reduction in cell growth compared to the 3'SP-targeting siRNA (si-3'SP) (**Extended Data Fig. 8d**). This could be due to the similar decrease in the protein expression of cell cycle regulators such as CDK2, CDK4, CDK6 and CCNE1, as well as WNT target genes AXIN2, MYC and TCF7, following si-CDS and si-3'SP treatments (**Fig. 3i, Extended Data Fig. 8h**).

With regards to the *in vivo* data, the lack of tumors for the si-CDS and si-3'SP could be due a halt in cell proliferation, as a result of the significantly reduced levels of cell cycle and WNT target genes as mentioned above. To reinforce our initial observations, we repeated the xenograft experiment and confirmed that the trend is consistent whereby targeted knockdown of the CTNNB1 CDS and 3'SP significantly reduced xenograft tumor growth compared to the siNC control and 3'FL knockdown (**Rebuttal Fig. 7**).

Rebuttal Fig. 7: Effect of siRNA-mediated knockdown of CTNNB1 CDS, 3'FL and 3'SP on xenograft tumor growth (n=5). siNC: non-targeting siRNA control; CDS: coding sequence; 3'FL: full length 3'UTR; 3'SP: spliced 3'UTR.

7. In fig 4c looks like all the siRNA are decreasing CTNNB1 protein levels. Why the mRNA levels are not consistent with protein ones?

>>Based on the ENCODE eCLIP data and RNA-seq data from RBP knockdowns, we shortlisted the RBP candidates which had binding sites in the terminal CTNNB1 exon or had a significant effect on the CTNNB1 3'SP transcript expression upon their knockdown. Consistently, the knockdown of TARDBP did not have a significant effect on CTNNB1 transcript (including the overall CDS exon-exon junctions) or protein expression. The inconsistency between the CTNNB1 transcript and protein expression resulting from the HNRNPC knockdown could be due to effects from other genes whose splicing is regulated by HNRNPC, and thus affected by its knockdown. These genes could be direct or indirect regulators of CTNNB1 expression, leading to the decreased CTNNB1 protein expression observed.

Overall, the RBP knockdowns showing the most consistent trend at both the transcript and protein levels are that of SF3B1, SRSF1 and U2AF2. Their siRNA-mediated knockdowns simultaneously reduce 3'SP, increase 3'FL transcript expression and decrease CTNNB1 protein expression (**Fig. 4b,c, Extended Data Fig. 11d,e**), whilst the CDS exon-exon junctions are largely unaffected (**Extended Data Fig. 11c**). These data are consistent with our proposed model whereby CTNNB1 is predominantly translated from the 3'SP transcript isoform.

8. is protein stability somehow altered? Is the protein from the 3'UTR spliced more stable, other than more transcribed? It is discussed but no data are shown, it would be proper adding them.

>>We thank the reviewer for the comment and suggestion. We have added the data showing the effect of inhibiting transcription, translation or proteasomal degradation using actinomycin D, cycloheximide or MG132, respectively (**Extended Data Fig. 13a-c, Rebuttal Fig. 8**). Our data show that there is no significant difference in transcript and protein expression from the CTNNB1 3'FL and 3'SP variants upon

transcription or translation inhibition. We also do not observe increased stability for the CTNNB1 protein translated from the 3'SP transcript variant as there is no difference in protein expression from the 3'FL and 3'SP transcripts upon inhibiting proteasomal degradation using MG132. This is likely due to the identical protein sequence of CTNNB1 from both the 3'FL and 3'SP transcript variants. Therefore, we deduce that the increased protein expression from the CTNNB1 3'SP variant is most likely due to its cytoplasmic localization. We have rearranged the Extended Data Figure and moved these data to the results section "3'UTR splicing enhances CTNNB1 expression to promote tumorigenesis" (please see below).

"We first tested whether 3'UTR splicing regulated CTNNB1 expression at the transcript or protein level by inhibiting transcription or translation following the overexpression of HA-tagged CTNNB1 variants. The transcript and protein expression of both 3'FL and 3'SP variants were similarly changed (Extended Data Fig. 13a,b), contrary to a previous study that demonstrated a longer mRNA half-life for CTNNB1 3'SP in HeLa cells, which could be due to tissue-specific regulation¹⁹. We further inhibited proteasomal degradation and did not observe differential CTNNB1 protein stability (Extended Data Fig. 13c)."

Rebuttal Fig. 8: (a) Effect of actinomycin D (ActD) treatment on transcript levels of exogenously

expressed CTNNB1 CDS, CDS+3'FL and CDS+3'SP in Hep3B and SNU398. (b,c) Effect of cycloheximide (CHX) (b) or MG132 (c) treatment on exogenously expressed CTNNB1 protein levels in Hep3B and SNU398. CDS: coding sequence; 3'FL: full length 3'UTR; 3'SP: spliced 3'UTR.

Reviewer #3:

Remarks to the Author:

In this manuscript, the authors systematically analyzed the alternative splicing in 3'UTR, and identified lots of 3'UTR splicing events (3USPs) in tumor and normal samples which were presented in a database called SpUR. They found that 3'UTR splicing is widespread, upregulated in tumors, correlated with poor prognosis and occurred more frequently in oncogenes. The size reduction of 3'UTR in cancers was also reported previously, which is consistent with the findings in this work. Furthermore, they focused on the 3'UTR splicing of CTNNB1 that has an oncogenic function. They showed that this gene may be predominantly translated from the transcripts with spliced 3'UTR, and splicing in 3'UTR helps the transcripts to be exported into the cytoplasm. In summary, they discovered new regulation through splicing in the 3'UTR splicing, as well as the possible mechanisms by which the 3'UTR splicing of a specific gene promote carcinogenesis. This is a relatively comprehensive study but the clinical application is uncertain. In addition, their conclusion is a little overstated.

Comments:

1. In figure 1e, the overlap after clustering is confusing. For example, the overlap of AML and COAD is about 40% in the tenth row and first column while 80% in eleventh row and fifth column. It seems that these two number should be identical. In fact, I do not think the clustering is necessary here.

>>We thank the reviewer for the comments. Yes, the numbers of the overlapped events are identical when comparing two cancer types, e.g. there are 543 overlapped events between AML and COAD. However, the total numbers of the identified events in these two are different, e.g. AML has 1,430 events in total, while COAD only has 708 events. In Fig. 1e, we showed the proportion of events which is shared/overlapped across cancers. Thus, 540 of 708 events in COAD overlapping with AML equates to 76.7% of COAD events; whereas 540 of 1,430 events is equivalent to 38% of the total events in AML which are overlapped with COAD. The purpose of this figure is to show that despite the huge heterogeneities across different cancer types, the majority of the identified c3USPs are common across cancers, even between solid tumors and leukemia, suggesting the c3USPs are mainly from genes that are ubiquitously expressed across different tissues. As suggested by the reviewer, we have removed the clustering from **Fig. 1e**.

2. The figure legends should be more detailed. For example, the meaning of numbers in figure 1f should be mentioned in the legend, also the definition of favorable and unfavorable lacks details (e.g., how the correlation is defined as significant).

>>We thank the reviewer for pointing this out. Due to the word limit, the detailed methods about how to correlate 3'UTR splicing levels with overall survival rate can be found in the Supplementary Notes section. In brief, we performed univariate Cox proportional hazards regression analysis and Kaplan-Meier analysis to correlate the splicing levels of each c3USP with overall patient survival in each cancer type ($p < 0.05$). The favorable and unfavorable events are defined as a higher splicing level correlating with a better (hazard ratio < 1) and poorer survival rate (hazard ratio > 1), respectively. We have added this to the figure legend for **Fig. 1f** for better clarity (please see below).

“Bar plots showing the number of favorable (hazard ratio < 1 , p -value < 0.05) and unfavorable (hazard ratio > 1 , p -value < 0.05) c3USPs across 10 cancers. The hazard ratios and p -values were obtained from univariate Cox proportional hazards regression analysis.”

3. The figure 2f and extended figure 5d are two replicates, however, there are two bands in THUMPD1 3'FL in figure 2f, but only one band in Extended figure 5d. Please explain such data inconsistency.

>>We note the reviewer's concern and would like to clarify that the data in Fig. 2f and Extended Data Fig. 5d are from Hep3B and HepG2 cell lines, respectively. We have also compiled the THUMPD1 data for all the biological repeats we performed for this experiment in both cell lines (**Rebuttal Fig. 9**). The double bands appear in some sets in both cell lines and this is highly likely due to variations in electrophoresis run time as well as agarose gel percentage between sets, leading to different band resolution. We would like to emphasize the overall consistent changes in the 3'FL and 3'SP bands indicated by the arrowheads.

4. The fourth section of the result entitled 'Targeted inhibition of 3'UTR splicing suppresses HCC carcinogenesis', however, this evidence is not enough to support this conclusion.

>>To address this, we have amended the section title (please see below) to match the supporting evidence showing impairment of HCC cell proliferation upon 3'UTR splicing inhibition.

“Targeted inhibition of 3'UTR splicing impedes HCC carcinogenesis”

We also added additional data showing that the splice-inhibiting ASOs targeting 3'UTR splicing of CTNNB1, CHEK1, MAPK1, THUMPD1 and WDR55, reduce the protein expression of at least two of the following cell cycle regulators: CCNE1, CDK2, CDK4 and CDK6 (Extended Data Fig. 6a, Rebuttal Fig. 10), which results in the growth suppression observed (Fig. 2h, Extended Data Fig. 5f). Furthermore, this effect is consistent for the second ASO we designed for each gene. Collectively, these data indicate that the ASO-mediated decrease in cell proliferation is due to an inhibition of the cell cycle. Please see below for the amended text from the manuscript:

“These effects were accompanied by a decrease in the protein expression of the respective genes and the repression of tumor growth, likely due to cell cycle inhibition as evident from the downregulated expression of cell cycle genes, including CCNE1, CDK2, CDK4 and CDK6 (Fig. 2g,h and Extended Data Fig. 5e,f,6a).”

Rebuttal Fig. 10: Effect of ASO-SS targeting 3'UTR splicing of CTNNB1, CHEK1, MAPK1, THUMPD1 and WDR55 on cell cycle markers. ASO: antisense oligonucleotide; NC: non-targeting control; SS: splice site.

5. In figure 3a, they found that the splicing of CTNNB1 3'UTR was significantly correlated with patient survival, and focused on one of the variants, 3'SP, with fully spliced 3'UTR. However they did not define what is the “over-spliced” 3'SP, and did not show the direct evidence that the 3' SP isoform, not the FL isoform, is the one under active translation. Since this is one of the most important point of their paper, they need to use ribosome profile and translation reporter to validate such conclusion.

>>We apologize for not properly defining the over-spliced 3'SP variant that is the focus of this study. The two CTNNB1 3'SP variants termed 3'SP and 3'SP2 in this study are annotated in RefSeq and GENCODE/Ensembl as NM_001098210.2 (CTNNB1-202) and NM_001330729.2 (CTNNB1-203),

respectively (the RefSeq references have been added to the main text). As our computational analyses showed that unlike 3'SP, the splicing level of 3'SP2 (CTNNB1-203) is not upregulated in HCC compared to the adjacent normal, we focused only on the shorter spliced isoform, 3'SP (CTNNB1-202) (as explained in the Supplementary Notes referring to Fig. 2f and Extended Data Fig. 5d).

We would like to clarify that we did not claim that 3'SP is the isoform under active translation instead of 3'FL. Our proposed model is that 3'SP is exported to the cytoplasm after splicing takes place, therefore becomes accessible for translation, whereas 3'FL being predominantly nuclear due to the retained intron is less accessible for translation (shown in fractionation and RNA-FISH data in Fig. 5b,c). To verify this, we performed both translation reporter assay and polysome profiling as suggested by the reviewer. We observe a higher luciferase signal for the 3'SP reporter construct compared to that for 3'FL, suggesting that the 3'SP variant is more highly translated (Fig. 4h, Rebuttal Fig. 11a). However, this is in contrast with the polysome profiles of both Hep3B and SNU398 cell lines which show similar distributions of the 3'FL and 3'SP transcripts in the different monosome and polysome fractions, indicating both transcript variants in the cytoplasm are similarly translated (Fig. 4i, Rebuttal Fig. 11b). The discrepancy could be due to several factors: (1) the luciferase ORF (~1kb) is much smaller than that of CTNNB1 (~3kb), which could carry additional components that influence its splicing, folding and/or translation, and (2) the luciferase reporters are exogenously expressed, whereas polysome profiles measure endogenous levels of CTNNB1 and may be more representative of physiological conditions. Based on these, we propose that cellular localization of the 3'UTR variants is the predominant factor influencing CTNNB1 protein expression. The new text and data have been added to the manuscript under the results section "3'UTR splicing enhances CTNNB1 expression to promote tumorigenesis".

Rebuttal Fig. 11: (a) Translation reporter assay of CTNNB1 3'FL and 3'SP. (b) Polysome profiles for CTNNB1 3'FL and 3'SP in Hep3B and SNU398 cell lines in the non-translating, 80S, lowly and highly translating polysomes. HSP90 and TBP are housekeeping controls. 3'FL: full length 3'UTR; 3'SP: spliced 3'UTR.

6. In figure 4, the authors claimed that CTNNB1 has oncogenic functions, and 3'SP variant is the one to be exported to cytoplasm and translated. They also claimed that U1SnRNP play a role in retaining FL variant of CTNNB1 in the nucleus. Although this model makes sense, I think they still need to provide more direct evidence to support this. For example, if the mutation in the 5' splice site of

3'UTR can increase the transport and translation of FL variant of CTNNB1, if the manipulation of 3' UTR splicing by ASO or SR proteins can affect CTNNB1 translation. In addition, they should use cell fractionation experiments (in addition to the microscopy in fig 4h) to quantify the cytoplasm export, as the relative levels of two isoforms in nucleus vs cytoplasm seems to be critical.

>>We thank the reviewer for the comments. To address this, we mutated the 5' splice site (3'FL-5'SSmut) in the psiCHECK2 3'FL and pcDNA3.1 HA-tagged CDS+3'FL plasmids. The overexpression of the 5'SSmut results in higher CTNNB1 protein expression, similar to that of 3'SP, compared to the wild-type 3'FL (3'FL-WT) (**Fig. 5d, Rebuttal Fig. 12a**). Additionally, we also performed luciferase and translation reporter assays. The mutation increases luciferase signals compared to that for 3'FL-WT in the luciferase reporter assay but not the translational reporter assay, suggesting that the increased CTNNB1 protein expression from the mutant is not due to translational advantage (**Fig. 5e,f, Rebuttal Fig. 12b,c**). This is further supported by the polysome profile of Hep3B cells treated with the 3'UTR splice-inhibiting ASO, which does not affect the distribution of the CTNNB1 3'UTR variants across the polysome fractions (**Fig. 5g, Rebuttal Fig. 12d**). Please see below for the added text:

“To further interrogate the importance of 3'UTR splicing for CTNNB1 expression, we mutated the 5' splice site (5'SSmut) of the CTNNB1 3'FL plasmid constructs. Overexpression of CTNNB1 5'SSmut resulted in CTNNB1 protein levels higher than that of wild-type 3'FL (3'FL-WT) and comparable to 3'SP (Fig. 5d). It also significantly increased luciferase activity compared to 3'FL-WT in the luciferase reporter assay but not the translation reporter assay (Fig. 5e,f), suggesting that the 5'SSmut mutation does not confer translational advantage. This is supported by the polysome profile of cells treated with splice site-blocking ASOs showing comparable distributions of the 3'FL and 3'SP variants across the polysome fractions compared to the control (Fig. 5g). Taken together, these findings indicate that differential cellular localization of the CTNNB1 3'UTR variants could be the predominant factor impacting CTNNB1 protein expression.”

Rebuttal Fig. 12: (a) Effect of the overexpression of CTNNB1 CDS+3'FL-WT, CDS+3'FL-5'SSmut and CDS+3'SP on exogenous CTNNB1 protein expression. (b,c) Luciferase activity of plasmid-transfected (b) and RNA-transfected (c) reporter constructs with CTNNB1 3'FL-WT and 3'FL-5'SSmut. (d) Effect of ASO-mediated blocking of the CTNNB1 3'UTR splice site on the polysome profiles for CTNNB1 3'FL and 3'SP. HSP90 and TBP are housekeeping controls. CDS: coding sequence; 3'FL-WT: wild-type full length 3'UTR; 5'SSmut: 5' splice site mutant; 3'SP: spliced 3'UTR; ASO-NC: non-targeting control ASO; ASO-SS: splice site ASO.

In addition to the nuclear-cytoplasmic fractionation data of the CTNNB1 3'UTR transcript distribution shown in Fig. 5b, we have also added the 3'FL to 3'SP ratio in the cytoplasmic and nuclear compartments (Rebuttal Table 3).

3'FL:3'SP ratio	Hep3B	SNU398
Cytoplasmic	0.54	0.50
Nuclear	1.75	2.35

Rebuttal Table 3: Full length (3'FL) to spliced (3'SP) 3'UTR transcript ratio in the cytoplasmic and nuclear compartments in Hep3B and SNU398 cells following nuclear-cytoplasmic fractionation (values shown are averages of four independent sets).

7. The database of SpUR should be more user-friendly.

>>As requested by the reviewer, the website interface has been redesigned to be more user friendly.

Werevised the interface as follows: (1) an aesthetic overhaul and a new horizontal layout for better visibility of both the tabular data and graphs, (2) in the splicing data tab, the help captions on the data columns, which provide brief definitions for the corresponding data column, are now more clearly visible, (3) we added a search functionality to allow users to filter the splicing data by gene and by cancer, (4) the instructions on how to display the graphs are now more clearly displayed, and (5) more information have been added to the exported graphs, including gene name, coordinates, cancer type and sample sizes, and users can now also export the tabular data.

Decision Letter, first revision:

Subject: Decision on Nature Cell Biology submission NCB-T44994A

Message:

*Please delete the link to your author homepage if you wish to forward this email to co-authors.

Dear Dr Tay,

Your manuscript, "Pan-cancer, pervasive upregulation of 3'UTR splicing drives tumorigenesis", has now been seen by 3 of our original referees. As you will see from their comments (attached below), although referees #1 and #2 are satisfied with the revision, referee #3 continues to raise several concerns that should be addressed before we can consider publication in Nature Cell Biology.

Nature Cell Biology editors discuss the referee reports in detail within the editorial team, including the chief editor, to identify key referee points that should be addressed with priority, and requests that are overruled as being beyond the scope of the current study. To guide the scope of the revisions, I have listed these points below. We are committed to providing a fair and constructive peer-review process, so please feel free to contact me if you would like to discuss any of the referee comments further.

In particular, it would be essential to:

a) strengthen the claim that 3'UTR splicing enhances CTNNB1 expression:

Second, their results using luciferase reporter assay and polysome profile assay is not clear: They found the luciferase activity from 3'SP significantly higher than 3'FL, however the polysome profiling did not showed difference between translation efficiency from 3'SP and 3'FL isoform. They gave a handwaving explanation on this discrepancy (such as longer CTNNB1 UTR, and different localization of 3'FL and 3'SP). However, I think they need to use better reporters with longer UTR of CTNNB1, or mutations on/near splice sites, or the RBPs binding sites which they referred to in Fig4. In addition, the translation efficiency of the other isoforms (3'SP2) was not tested. Collectively, I think the conclusion '3'UTR splicing enhances CTNNB1 expression' may be over-stated.

b) examine the contribution of the 3'SP2 variant of CTNNB1 to tumorigenesis:

For my original point on CTNNB1, they have conducted additional analyses but the results are still a little confusing. First, CTNNB1 has two 3'UTR variants: 3'SP and 3'SP2. They showed that 3'SP significantly upregulated of splicing level in the tumor samples, and thus only focused on 3'SP. However, 3'SP2 was also significantly changed in tumor samples although it was reduced rather than increased. Since the splicing level of 3'SP2 is actually higher than 3'SP under normal conditions (extended data figure 7c), the role of 3'SP2 may also be important. They need to test if change of 3'SP2 by siRNA or overexpression can also affect tumorigenesis, because I am not sure whether tumorigenesis was resulted from 3'SP upregulation or 3'SP2 downregulation.

c) better link the splicing regulators to the cancer context:

In addition, the authors observed several RBPs like SRSF1 and U2AF2 which enriched for CTNNB1 3'UTR variants. However, they did not examine the change of their expression levels in tumor samples. Since SRSF1 is one of the general SR proteins and is usually upregulated in tumors, this may look into the effect of SRSF1 level on the CTNNB1 3'UTR splicing variants. In addition, U2AF2 is a critical subunit of core spliceosomal component that regulates splicing of almost all genes, but it is unclear how they explain its connection in tumorigenesis through 3'SP. In fact, I think the referring of RBP's role in this section is a little casual.

d) All other referee concerns pertaining to strengthening existing data should also be addressed.

e) Finally please pay close attention to our guidelines on statistical and methodological reporting (listed below) as failure to do so may delay the reconsideration of the revised manuscript. In particular please provide:

We therefore invite you to take these points into account when revising the manuscript. In addition, when preparing the revision please:

- ensure that it conforms to our format instructions and publication policies (see below and <https://www.nature.com/nature/for-authors>).
- provide a point-by-point rebuttal to the full referee reports verbatim, as provided at the end of this letter.
- provide the completed Reporting Summary (found here <https://www.nature.com/documents/nr-reporting-summary.pdf>). This is essential for reconsideration of the manuscript and will be available to editors and referees in the event of peer review. For more information see <http://www.nature.com/authors/policies/availability.html> or contact me.

When submitting the revised version of your manuscript, please pay close attention to our [href="https://www.nature.com/nature-research/editorial-policies/image-integrity">Digital Image Integrity Guidelines](https://www.nature.com/nature-research/editorial-policies/image-integrity). and to the following points below:

Nature Cell Biology is committed to improving transparency in authorship. As part of our efforts in this direction, we are now requesting that all authors identified as ‘corresponding author’ on published papers create and link their Open Researcher and Contributor Identifier (ORCID) with their account on the Manuscript Tracking System (MTS), prior to acceptance. ORCID helps the scientific community achieve unambiguous attribution of all scholarly contributions. You can create and link your ORCID from the home page of the MTS by clicking on ‘Modify my Springer Nature account’. For more information please visit www.springernature.com/orcid.

This journal strongly supports public availability of data. Please place the data used in your paper into a public data repository, or alternatively, present the data as Supplementary Information. If data can only

be shared on request, please explain why in your Data Availability Statement, and also in the correspondence with your editor. Please note that for some data types, deposition in a public repository is mandatory - more information on our data deposition policies and available repositories appears below.

[REDACTED]

We would like to receive the revision within 2 months. If submitted within this time period, reconsideration of the revised manuscript will not be affected by related studies published elsewhere, or accepted for publication in Nature Cell Biology in the meantime. We would be happy to consider a revision even after this timeframe, but in that case we will consider the published literature at the time of resubmission when assessing the file.

We hope that you will find our referees' comments, and editorial guidance helpful. Please do not hesitate to contact me if there is anything you would like to discuss.

Best wishes,

Jie Wang

Jie Wang, PhD
Senior Editor
Nature Cell Biology

Tel: +44 (0) 207 843 4924
email: jie.wang@nature.com

Reviewers' Comments:

Reviewer #1:

Remarks to the Author:

The authors have responded to my initial concerns and questions. I have no further issues with the manuscript.

Reviewer #2:

Remarks to the Author:

The revision of this interesting manuscript was carefully done and the authors added several important experiments.

The data are important and novel and the manuscript is of great interest for the readers of the journal.

Reviewer #3:

Remarks to the Author:

In this revised manuscript, the author did quite a few experiments and analysis to clarify some confused points, which significantly improved the manuscript. They sufficiently addressed some of my points, with the exception of my concerns on CTNNB1 3'SP. In addition, the mechanisms on how RBPs regulates the 3'SP and how 3'SP regulates downstream translation is unclear.

Specific Comments:

In Fig1e, it is still confusing why they compared solid tumors to AML. To evaluate the overall tissue similarities, I think they should use overlapped p-value by hypergeometric test rather than these percent of overlaps.

For my original point on CTNNB1, they have conducted additional analyses but the results are still a little confusing. First, CTNNB1 has two 3'UTR variants: 3'SP and 3'SP2. They showed that 3'SP significantly upregulated of splicing level in the tumor samples, and thus only focused on 3'SP. However, 3'SP2 was also significantly changed in tumor samples although it was reduced rather than increased. Since the splicing level of 3'SP2 is actually higher than 3'SP under normal conditions (extended data figure 7c), the role of 3'SP2 may also be important. They need to test if change of 3'SP2 by siRNA or overexpression can also affect tumorigenesis, because I am not sure whether tumorigenesis was resulted from 3'SP upregulation or 3'SP2 downregulation.

Second, their results using luciferase reporter assay and polysome profile assay is not clear: They found the luciferase activity from 3'SP significantly higher than 3'FL, however the polysome profiling did not showed difference between translation efficiency from 3'SP and 3'FL isoform. They gave a handwaving explanation on this discrepancy (such as longer CTNNB1 UTR, and different localization of 3'FL and 3'SP). However, I think they need to use better reporters with longer UTR of CTNNB1, or mutations on/near splice sites, or the RBPs binding sites which they referred to in Fig4. In addition, the translation efficiency of the other isoforms (3'SP2) was not tested. Collectively, I think the conclusion '3'UTR splicing enhances CTNNB1 expression' may be over-stated.

In addition, the authors observed several RBPs like SRSF1 and U2AF2 which enriched for CTNNB1 3'UTR variants. However, they did not examine the change of their expression levels in tumor samples. Since SRSF1 is one of the general SR proteins and is usually upregulated in tumors, this may look into the effect of SRSF1 level on the CTNNB1 3'UTR splicing variants. In addition, U2AF2 is a critical subunit of core spliceosomal component that regulates splicing of almost all genes, but it is unclear how they explain its connection in tumorigenesis through 3'SP. In fact, I think the referring of RBP's role in this section is a little casual.

GUIDELINES FOR SUBMISSION OF NATURE CELL BIOLOGY ARTICLES

ARTICLE FORMAT

ABSTRACT – should not exceed 150 words and should be unreferenced. This paragraph is the most visible part of the paper and should briefly outline the background and rationale for the work, and accurately summarize the main results and conclusions. Key genes, proteins and organisms should be specified to ensure discoverability of the paper in online searches.

TEXT – the main text consists of the Introduction, Results, and Discussion sections and must not exceed 3500 words including the abstract. The Introduction should expand on the background relating to the work. The Results should be divided in subsections with subheadings, and should provide a concise and accurate description of the experimental findings. The Discussion should expand on the findings and their implications. All relevant primary literature should be cited, in particular when discussing the background and specific findings.

REFERENCES – are limited to a total of 70 in the main text and Methods combined,. They must be numbered sequentially as they appear in the main text, tables and figure legends and Methods and must follow the precise style of Nature Cell Biology references. References only cited in the Methods

should be numbered consecutively following the last reference cited in the main text. References only associated with Supplementary Information (e.g. in supplementary legends) do not count toward the total reference limit and do not need to be cited in numerical continuity with references in the main text. Only published papers can be cited, and each publication cited should be included in the numbered reference list, which should include the manuscript titles. Footnotes are not permitted.

Methods should be written concisely, but should contain all elements necessary to allow interpretation and replication of the results. As a guideline, Methods sections typically do not exceed 3,000 words. The Methods should be divided into subsections listing reagents and techniques. When citing previous methods, accurate references should be provided and any alterations should be noted. Information must be provided about: antibody dilutions, company names, catalogue numbers and clone numbers for monoclonal antibodies; sequences of RNAi and cDNA probes/primers or company names and catalogue numbers if reagents are commercial; cell line names, sources and information on cell line identity and authentication. Animal studies and experiments involving human subjects must be reported in detail, identifying the committees approving the protocols. For studies involving human subjects/samples, a statement must be included confirming that informed consent was obtained. Statistical analyses and information on the reproducibility of experimental results should be provided in a section titled “Statistics and Reproducibility”.

All Nature Cell Biology manuscripts submitted on or after March 21 2016, must include a Data availability statement as a separate section after Methods but before references, under the heading “Data Availability”. For Springer Nature policies on data availability see <http://www.nature.com/authors/policies/availability.html>; for more information on this particular policy see <http://www.nature.com/authors/policies/data/data-availability-statements-data-citations.pdf>. The Data availability statement should include:

- Accession codes for primary datasets (generated during the study under consideration and designated as “primary accessions”) and secondary datasets (published datasets reanalysed during the study under consideration, designated as “referenced accessions”). For primary accessions data should be made public to coincide with publication of the manuscript. A list of data types for which submission to community-endorsed public repositories is mandated (including sequence, structure, microarray, deep sequencing data) can be found here <http://www.nature.com/authors/policies/availability.html#data>.

- Unique identifiers (accession codes, DOIs or other unique persistent identifier) and hyperlinks for datasets deposited in an approved repository, but for which data deposition is not mandated (see here for details <http://www.nature.com/sdata/data-policies/repositories>).
- At a minimum, please include a statement confirming that all relevant data are available from the authors, and/or are included with the manuscript (e.g. as source data or supplementary information), listing which data are included (e.g. by figure panels and data types) and mentioning any restrictions on availability.
- If a dataset has a Digital Object Identifier (DOI) as its unique identifier, we strongly encourage including this in the Reference list and citing the dataset in the Methods.

We recommend that you upload the step-by-step protocols used in this manuscript to the Protocol Exchange. More details can found at www.nature.com/protocolexchange/about.

DISPLAY ITEMS – main display items are limited to 6-8 main figures and/or main tables. For Supplementary Information see below.

FIGURES – Colour figure publication costs \$395 per colour figure. All panels of a multi-panel figure must be logically connected and arranged as they would appear in the final version. Unnecessary figures and figure panels should be avoided (e.g. data presented in small tables could be stated briefly in the text instead).

All imaging data should be accompanied by scale bars, which should be defined in the legend. Cropped images of gels/blots are acceptable, but need to be accompanied by size markers, and to retain visible background signal within the linear range (i.e. should not be saturated). The boundaries of panels with low background have to be demarked with black lines. Splicing of panels should only be considered if unavoidable, and must be clearly marked on the figure, and noted in the legend with a statement on whether the samples were obtained and processed simultaneously. Quantitative comparisons between samples on different gels/blots are discouraged; if this is unavoidable, it has to be performed for samples derived from the same experiment with gels/blots were processed in parallel, which needs to be stated in the legend.

Figures should be provided at approximately the size that they are to be printed at (single column is 86 mm, double column is 170 mm) and should not exceed an A4 page (8.5 x 11"). Reduction to the scale that will be used on the page is not necessary, but multi-panel figures should be sized so that the whole figure can be reduced by the same amount at the smallest size at which essential details in each panel are visible. In the interest of our colour-blind readers we ask that you avoid using red and green for

contrast in figures. Replacing red with magenta and green with turquoise are two possible colour-safe alternatives. Lines with widths of less than 1 point should be avoided. Sans serif typefaces, such as Helvetica (preferred) or Arial should be used. All text that forms part of a figure should be rewritable and removable.

Regardless of format, all figures must be vector graphic compatible files, not supplied in a flattened raster/bitmap graphics format, but should be fully editable, allowing us to highlight/copy/paste all text and move individual parts of the figures (i.e. arrows, lines, x and y axes, graphs, tick marks, scale bars etc). The only parts of the figure that should be in pixel raster/bitmap format are photographic images or 3D rendered graphics/complex technical illustrations.

Unprocessed scans of all key data generated through electrophoretic separation techniques need to be presented in a supplementary figure that should be labeled and numbered as the final supplementary figure, and should be mentioned in every relevant figure legend. This figure does not count towards the total number of figures and is the only figure that can be displayed over multiple pages, but should be provided as a single file, in PDF or TIFF format. Data in this figure can be displayed in a relatively informal style, but size markers and the figures panels corresponding to the presented data must be indicated.

The total number of Supplementary Figures (not including the “unprocessed scans” Supplementary Figure) should not exceed the number of main display items (figures and/or tables (see our Guide to Authors and March 2012 editorial <http://www.nature.com/ncb/authors/submit/index.html#suppinfo>; <http://www.nature.com/ncb/journal/v14/n3/index.html#ed>). No restrictions apply to Supplementary Tables or Videos, but we advise authors to be selective in including supplemental data.

Each Supplementary Figure should be provided as a single page and as an individual file in one of our accepted figure formats and should be presented according to our figure guidelines (see above). Supplementary Tables should be provided as individual Excel files. Supplementary Videos should be

provided as .avi or .mov files up to 50 MB in size. Supplementary Figures, Tables and Videos must be accompanied by a separate Word document including titles and legends.

GUIDELINES FOR EXPERIMENTAL AND STATISTICAL REPORTING

REPORTING REQUIREMENTS – We ask authors to complete a Reporting Summary that collects information on experimental design and reagents. We hope this will aid in your evaluation of the paper. The Reporting Summary can be found here <https://www.nature.com/documents/nr-reporting-summary.pdf>) Please note that these forms are dynamic ‘smart pdfs’ and must therefore be downloaded and completed in Adobe Reader. We will then flatten them for ease of use. If you would like to reference the guidance text as you complete the template, please access these flattened versions at <http://www.nature.com/authors/policies/availability.html>.

We strongly recommend the presentation of source data for graphical and statistical analyses as a separate Supplementary Table, and request that source data for all independent repeats are provided when representative experiments of multiple independent repeats, or averages of two independent experiments are presented. This supplementary table should be in Excel format, with data for different figures provided as different sheets within a single Excel file. It should be labelled and numbered as one of the supplementary tables, titled “Statistics Source Data”, and mentioned in all relevant figure legends.

Author Rebuttal, first revision:**Response to reviewers**

We thank the reviewers for their positive feedback.

Reviewer #1:

Remarks to the Author:

The authors have responded to my initial concerns and questions. I have no further issues with the manuscript.

Reviewer #2:

Remarks to the Author:

The revision of this interesting manuscript was carefully done and the authors added several important experiments.

The data are important and novel and the manuscript is of great interest for the readers of the journal.

Reviewer #3:

Remarks to the Author:

In this revised manuscript, the author did quite a few experiments and analysis to clarify some confused points, which significantly improved the manuscript. They sufficiently addressed some of my points, with the exception of my concerns on CTNNB1 3'SP. In addition, the mechanisms on how RBPs regulates the 3'SP and how 3'SP regulates downstream translation is unclear.

Specific Comments:

1. In Fig1e, it is still confusing why they compared solid tumors to AML. To evaluate the overall tissue similarities, I think they should use overlapped p-value by hypergeometric test rather than these percent of overlaps.

>> We thank the reviewer for the useful suggestions. We found that c3USPs identified in different cancer types are significantly overlapped. As splicing events have been reported to have high tissue specificity (Wang et al, Nature, 2008), we extended our analysis to a hematological malignancy, Acute Myeloid Leukemia (AML) (34 patients versus 21 healthy controls) to further examine common vs tissue-specific 3'UTR splicing events. (This explanation has been added to the main text, **page 7 lines 83-84**). As

expected, the overlap ratio between AML and solid tumors is only around 46%. As suggested, we performed hypergeometric test between each pair of cancer types and the results are consistent with the overlap ratio we presented. As shown in **Extended Data Fig. 3d (Rebuttal Fig. 1)**, the overlap between each pair of solid tumors is extremely significant ($p < 1e-100$), while the overlaps between AML and five types of solid tumors, including COAD, KIRC, LIHC, LUAD and STAD, are also significant ($p < 0.05$).

- For my original point on CTNNB1, they have conducted additional analyses but the results are still a little confusing. First, CTNNB1 has two 3'UTR variants: 3'SP and 3'SP2. They showed that 3'SP significantly upregulated of splicing level in the tumor samples, and thus only focused on 3'SP. However, 3'SP2 was also significantly changed in tumor samples although it was reduced rather than increased. Since the splicing level of 3'SP2 is actually higher than 3'SP under normal conditions (extended data figure 7c), the role of 3'SP2 may also be important. They need to test if change of 3'SP2 by siRNA or overexpression can also affect tumorigenesis, because I am not sure whether tumorigenesis was resulted from 3'SP upregulation or 3'SP2 downregulation.

>> To address the comments above, we have performed additional experiments to assess the effects of knocking down the 3'SP2 variant. Our qPCR analysis shows that the 3'SP2-targeting siRNA specifically reduces transcript expression of the 3'SP2 variant and has a small effect on the overall CTNNB1 CDS transcript expression (**Rebuttal Fig. 2a**). This is reflected at the protein level whereby si-3'SP2 has a much smaller effect on CTNNB1 protein expression compared to si-CDS and si-3'SP (**Rebuttal Fig. 2b**). This trend is consistent in the growth assay whereby 3'SP2 knockdown reduces growth at a similar order of magnitude to the 3'FL knockdown, but the effect is smaller compared to that of si-CDS and si-3'SP (**Rebuttal Fig. 2c**). Thus, the downregulation of 3'SP2 is unlikely to drive tumorigenesis. Furthermore, 3'SP2 only increases luciferase reporter signal to the same level as 3'FL, which is lower than that of 3'SP (**Rebuttal Fig. 2d**). Collectively, our data suggest that changes in CTNNB1 3'SP2 levels have minimal effect on CTNNB1 expression and tumorigenesis.

Rebuttal Fig. 2: Effect of siRNA-mediated knockdown of CTNNB1 3'SP2 on (a) the CTNNB1 CDS, 3'FL, 3'SP and 3'SP2 transcript expression, (b) CTNNB1 protein expression and (c) anchorage-independent

growth in Hep3B and SNU398. (d) Luciferase activity of reporter constructs with CTNNB1 3'FL, 3'SP and 3'SP2. CDS: coding sequence; 3'FL: full length 3'UTR; 3'SP: spliced 3'UTR (variant 1); 3'SP2: spliced 3'UTR (variant 2); siNC: non-targeting control siRNA; n.s.: not significant.

3. Second, their results using luciferase reporter assay and polysome profile assay is not clear: They found the luciferase activity from 3'SP significantly higher than 3'FL, however the polysome profiling did not show difference between translation efficiency from 3'SP and 3'FL isoform. They gave a handwaving explanation on this discrepancy (such as longer CTNNB1 UTR, and different localization of 3'FL and 3'SP). However, I think they need to use better reporters with longer UTR of CTNNB1, or mutations on/near splice sites, or the RBPs binding sites which they referred to in Fig4. In addition, the translation efficiency of the other isoforms (3'SP2) was not tested. Collectively, I think the conclusion '3'UTR splicing enhances CTNNB1 expression' may be over-stated.

>> We thank the reviewer for the comments and suggestions. We have performed luciferase reporter assays comparing the effect of the longer CTNNB1 3'UTR, 3'FL, to 3'SP (**Fig. 4g**), as well as 3'FL-WT (wild-type) to 3'FL-5'SSmut (5' splice site mutant) (**Fig. 5e**). For easier comparison, we have combined these into a single chart (**Rebuttal Fig. 3a**). With regards to the discrepancy between the luciferase data and polysome profiles for 3'FL and 3'SP, we would like to clarify that we were referring to the longer CTNNB1 CDS, not 3'UTR, compared to the luciferase CDS. The sheer difference in CDS lengths as well as the potential differences in regulatory components/elements between the luciferase and CTNNB1 CDS regions could be contributing factors to the variation observed. Furthermore, luciferase signals are dependent on expression levels and cellular distribution of transcripts, while polysome profiles are not. Although 3'FL maybe predominantly nuclear, the small proportion of 3'FL transcripts present in the cytoplasm (as shown in our cellular fractionation data in **Fig. 5b**) could be translated, hence the similar polysome profiles for CTNNB1 3'FL and 3'SP. This trend is also observed when manipulating the splice site, whereby the splice site mutant (5'SSmut) increases reporter signal but the ASO-mediated inhibition of 3'UTR splicing (mimicking the effect of 5'SS mut) did not alter the 3'FL and 3'SP polysome profiles (**Fig. 5g**). Based on these collective observations, we proposed a model whereby differential transcript localization, not translation efficiency, may be the predominant factor influencing CTNNB1 protein expression and tumorigenesis. Our model of a primarily nuclear intron-containing 3'FL variant is also supported by previous findings of U1 snRNP-mediated nuclear retention of transcripts with introns (Mount et al, Cell, 1983; Prasanth et al, Cell; 2005; Takemura et al, Genes Cell, 2011). To moderate our claims, we have amended our conclusion as follows:

"3'UTR splicing may enhance CTNNB1 expression to promote tumorigenesis."

Additionally, we performed polysome profiling for 3'SP2 (**Rebuttal Fig. 3b**) and found that there was no significant difference in translation efficiency between 3'SP and 3'SP2.

Rebuttal Fig. 3: (a) Translation reporter assay comparing the luciferase activity of reporter constructs with CTNNB1 3'FL-WT, 3'FL-5'SSmut and 3'SP. (b) Polysome profiles of CTNNB1 3'SP and 3'SP2 in Hep3B and SNU398. 3'FL: full length 3'UTR; 3'SP: spliced 3'UTR (variant 1); 3'SP2: spliced 3'UTR (variant 2); WT: wild-type; 5'SSmut: 5' splice site mutant; n.s.: not significant.

- In addition, the authors observed several RBPs like SRSF1 and U2AF2 which enriched for CTNNB1 3'UTR variants. However, they did not examine the change of their expression levels in tumor samples. Since SRSF1 is one of the general SR proteins and is usually upregulated in tumors, this may look into the effect of SRSF1 level on the CTNNB1 3'UTR splicing variants. In addition, U2AF2 is a critical subunit of core spliceosomal component that regulates splicing of almost all genes, but it is unclear how they explain its connection in tumorigenesis through 3'SP. In fact, I think the referring of RBP's role in this section is a little casual.

>> Thank you for the comments. To address these, we first checked the transcript expression levels of these RBPs using TCGA-LIHC data, which show that SF3B1, SRSF1 and U2AF2 are upregulated in the tumor samples compared to the normal tissues (**Rebuttal Fig. 4a**). We further verify the upregulation of these RBPs in HCC cell lines, Hep3B, HepG2 and SNU398, compared to normal liver hepatocytes, THLE-2 (**Rebuttal Fig. 4b**). We also find that the individual knockdown of these RBPs significantly reduce HCC cell proliferation (**Rebuttal Fig. 4c**), which is in line with their elevated expression and potential oncogenic role in HCC. We have previously shown that the knockdown of SRSF1 and U2AF2 reduced 3'SP while increasing 3'FL transcript expression (**Fig. 4b and Extended Data Fig. 11d**). At the same time, it also reduced CTNNB1 protein expression (**Fig. 4c and Extended Data Fig. 11e**). We agree with the Reviewer that SRSF1 is a general SR protein and U2AF2 may regulate splicing of many genes. However, different splicing events have distinct tolerance or sensitivity to the perturbation of the RBP expression. Furthermore, we have verified that in the context of CTNNB1, they appear to more specifically regulate 3'UTR splicing as the four CDS exon-exon junctions tested were largely unaffected by their knockdown (**Extended Data Fig. 11c**). Additionally, the knockdown of SRSF1 and U2AF2 phenocopies the siRNA-mediated 3'SP knockdown (**Extended Data Fig. 8d**), thus their effect on tumorigenesis could be in part due to changes in CTNNB1 3'SP levels, highlighting the potential role of these RBPs in the splicing of CTNNB1 3'UTR and tumorigenesis.

Rebuttal Fig. 4: (a) Transcript expression level of SF3B1, SRSF1 and U2AF2 in the TCGA-LIHC tumor (n = 374) compared to normal liver (n = 50) samples (extracted from ENCORI, <http://starbase.sysu.edu.cn/>). (b) Transcript expression of the above RNA-binding proteins in normal liver cells, THLE-2, and HCC cell lines, Hep3B, HepG2 and SNU398. (c) Effect of siRNA-mediated knockdown of SF3B1, SRSF1 and U2AF2 on anchorage-independent growth in Hep3B and SNU398. siNC: non-targeting control siRNA.

Decision Letter, second revision:

Subject: Your manuscript, NCB-T44994B
Message:

Our ref: NCB-T44994B

8th February 2022

Dear Dr. Tay,

Thank you for submitting your revised manuscript "Pan-cancer, pervasive upregulation of 3'UTR splicing drives tumorigenesis" (NCB-T44994B). It has now been seen by the original referee 3 and the comments are below. The reviewers find that the paper has improved in revision, and therefore we'll be happy in principle to publish it in Nature Cell Biology, pending minor revisions to satisfy the referee's final requests and to comply with our editorial and formatting guidelines.

The current version of your manuscript is in a PDF format. Please email us a copy of the file in an editable format (Microsoft Word or LaTeX)-- we can not proceed with PDFs at this stage.

Thank you again for your interest in Nature Cell Biology. Please do not hesitate to contact me if you have any questions.

Sincerely,

Jie Wang, PhD
Senior Editor
Nature Cell Biology

Tel: +44 (0) 207 843 4924
email: jie.wang@nature.com

Reviewer #3 (Remarks to the Author):

The authors conducted a series of experiments and analyses to address my concerns, and the revision is largely acceptable. I only have two minor comments that they can address during final stage of editing.

However, I still think they should comments on the surprisingly low overlap of c3USPs between the AML and the 10 TCGA tumors that have very high overlap within the group. Does this reflect a intrinsic difference between AML and solid tumor? Or is it because of the different data source (i.e. batch effect)?

In addition, I think they should include the data in rebuttal figure 2 into the revised version as a supplementary figure, as this reflect an important control.

Decision letter, final requests:

Subject: NCB: Your manuscript, NCB-T44994B
Message: Our ref: NCB-T44994B

16th February 2022

Dear Dr. Tay,

Thank you for your patience as we've prepared the guidelines for final submission of your Nature Cell Biology manuscript, "Pan-cancer, pervasive upregulation of 3'UTR splicing drives tumorigenesis" (NCB-T44994B). Please carefully follow the step-by-step instructions provided in the attached file, and add a response in each row of the table to indicate the changes that you have made. Ensuring that each point is addressed will help to ensure that your revised manuscript can be swiftly handed over to our production team.

We would like to start working on your revised paper, with all of the requested files and forms, as soon as possible (preferably within one week). Please get in contact with us if you anticipate delays.

In recognition of the time and expertise our reviewers provide to Nature Cell Biology's editorial process, we would like to formally acknowledge their contribution to the external peer review of your manuscript entitled "Pan-cancer, pervasive upregulation of 3'UTR splicing drives tumorigenesis". For those reviewers who give their assent, we will be publishing their names alongside the published article.

Nature Cell Biology offers a Transparent Peer Review option for new original research manuscripts submitted after December 1st, 2019. As part of this initiative, we encourage our authors to support increased transparency into the peer review process by agreeing to have the reviewer comments, author rebuttal letters, and editorial decision letters published as a Supplementary item. When you submit your final files please clearly state in your cover letter whether or not you would like to participate in this initiative. Please note that failure to state your preference will result in delays in accepting your manuscript for publication.

Cover suggestions

As you prepare your final files we encourage you to consider whether you have any images or illustrations that may be appropriate for use on the cover of Nature Cell Biology.

Nature Cell Biology has now transitioned to a unified Rights Collection system which will allow our Author Services team to quickly and easily collect the rights and permissions required to publish your

work. Approximately 10 days after your paper is formally accepted, you will receive an email in providing you with a link to complete the grant of rights. If your paper is eligible for Open Access, our Author Services team will also be in touch regarding any additional information that may be required to arrange payment for your article.

Please note that Nature Cell Biology is a Transformative Journal (TJ). Authors may publish their research with us through the traditional subscription access route or make their paper immediately open access through payment of an article-processing charge (APC). Authors will not be required to make a final decision about access to their article until it has been accepted. Find out more about Transformative Journals

Authors may need to take specific actions to achieve compliance with funder and institutional open access mandates. For submissions from January 2021, if your research is supported by a funder that requires immediate open access (e.g. according to Plan S principles) then you should select the gold OA route, and we will direct you to the compliant route where possible. For authors selecting the subscription publication route our standard licensing terms will need to be accepted, including our self-archiving policies. Those standard licensing terms will supersede any other terms that the author or any third party may assert apply to any version of the manuscript.

For information regarding our different publishing models please see our Transformative Journals page. If you have any questions about costs, Open Access requirements, or our legal forms, please contact ASJournals@springernature.com.

[REDACTED]

Best regards,

Ziqian Li
Editorial Assistant
Nature Cell Biology

On behalf of

Jie Wang, PhD
Senior Editor
Nature Cell Biology

Tel: +44 (0) 207 843 4924
email: jie.wang@nature.com

Reviewer #3:

Remarks to the Author:

The authors conducted a series of experiments and analyses to address my concerns, and the revision is largely acceptable. I only have two minor comments that they can address during final stage of editing.

However, I still think they should comments on the surprisingly low overlap of c3USPs between the AML and the 10 TCGA tumors that have very high overlap within the group. Does this reflect a intrinsic difference between AML and solid tumor? Or is it because of the different data source (i.e. batch effect)?

In addition, I think they should include the data in rebuttal figure 2 into the revised version as a supplementary figure, as this reflect an important control.

Author Rebuttal, second revision:

Response to reviewer

We thank the reviewer for the positive feedback.

Reviewer #3 (Remarks to the Author):

The authors conducted a series of experiments and analyses to address my concerns, and the revision is largely acceptable. I only have two minor comments that they can address during final stage of editing.

However, I still think they should comments on the surprisingly low overlap of c3USPs between the AML and the 10 TCGA tumors that have very high overlap within the group. Does this reflect a intrinsic difference between AML and solid tumor? Or is it because of the different data source (i.e. batch effect)?

Thank you for the comments. To address this, we have added the following to the discussion section:

“The low overlap of c3USPs between AML and the solid tumors is noteworthy and may reflect potential intrinsic differences between blood and solid tumors at the genomic level. Further work on other hematological malignancies will provide a better understanding of these variations.”

In addition, I think they should include the data in rebuttal figure 2 into the revised version as a supplementary figure, as this reflect an important control.

As suggested, we have incorporated rebuttal figure 2 and a short explanation into the revised manuscript (**Page 9, line 133-134; Extended Data Fig. 5a-d**).

Final Decision Letter:

Subject: Decision on Nature Cell Biology submission NCB-T44994C
Message:

Dear Dr Tay,

I am pleased to inform you that your manuscript, "Pan-cancer pervasive upregulation of 3'UTR splicing drives tumorigenesis", has now been accepted for publication in Nature Cell Biology.

Please note that Nature Cell Biology is a Transformative Journal (TJ). Authors may publish their research with us through the traditional subscription access route or make their paper immediately open access through payment of an article-processing charge (APC). Authors will not be required to make a final decision about access to their article until it has been accepted. Find out more about Transformative Journals

Authors may need to take specific actions to achieve compliance with funder and institutional open access mandates. If your research is supported by a funder that requires immediate open access (e.g. according to Plan S principles) then you should select the gold OA route, and we will direct you to the compliant route where possible. For authors selecting the subscription publication route, the journal's standard licensing terms will need to be accepted, including self-archiving policies. Those licensing terms will supersede any other terms that the author or any third party may assert apply to any version of the manuscript.

If you have not already done so, we strongly recommend that you upload the step-by-step protocols used in this manuscript to the Protocol Exchange (www.nature.com/protocolexchange), an open online resource established by Nature Protocols that allows researchers to share their detailed experimental know-how. All uploaded protocols are made freely available, assigned DOIs for ease of citation and are fully searchable through nature.com. Protocols and Nature Portfolio journal papers in which they are used can be linked to one another, and this link is clearly and prominently visible in the online versions of both papers. Authors who performed the specific experiments can act as primary authors for the Protocol as they will be best placed to share the methodology details, but the Corresponding Author of the present research paper should be included as one of the authors. By uploading your Protocols to Protocol Exchange, you are enabling researchers to more readily reproduce or adapt the methodology you use, as well as increasing the visibility of your protocols and papers. You can also establish a dedicated page to collect your lab Protocols. Further information can be found at www.nature.com/protocolexchange/about

With kind regards,

Jie Wang, PhD
Senior Editor
Nature Cell Biology

Tel: +44 (0) 207 843 4924
email: jie.wang@nature.com